# LOKI: A Comprehensive Synthetic Data Detection Benchmark using Large Multimodal Models

**Junyan Ye**[1,2]*, **Baichuan Zhou**[2]* , **Zilong Huang**[1]* , **Junan Zhang**[6,2]* , **Tianyi Bai**[2,5]* ,
**Hengrui Kang**[2] , **Jun He**[1] , **Honglin Lin**[2] , **Zihao Wang**[1] , **Tong Wu**[4] , **Zhizheng Wu**[6,2] ,
**Yiping Chen**[1] , **Dahua Lin**[2,4] , **Conghui He**[2,3]†, **Weijia Li**[1] †

[1] Sun Yat-sen University, [2] Shanghai AI Laboratory, [3] SenseTime Research,
[4] The Chinese University of Hong Kong, [5] The Hong Kong University of Science and Technology,
[6] SDS, SRIBD, The Chinese University of Hong Kong, Shenzhen

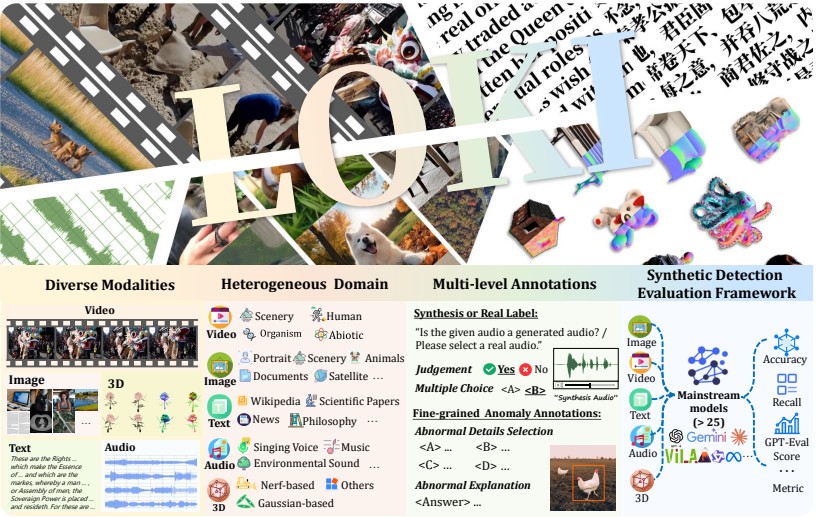

Figure 1: **Overview of LOKI benchmark.** LOKI possesses four key characteristics: 1) Diverse modalities (video, image, 3D, text and audio); 2) Heterogeneous categories (26 detailed subcategories); 3) Multi-level annotations; 4) Multimodal synthetic data evaluation framework.

## ABSTRACT

With the rapid development of AI-generated content, the future internet may be inundated with synthetic data, making the discrimination of authentic and credible multimodal data increasingly challenging. Synthetic data detection has thus garnered widespread attention, and the performance of large multimodal models (LMMs) in this task has attracted significant interest. LMMs can provide natural language explanations for their authenticity judgments, enhancing the explainability of synthetic content detection. Simultaneously, the task of distinguishing between real and synthetic data effectively tests the perception, knowledge, and reasoning capabilities of LMMs. In response, we introduce LOKI, a novel benchmark designed to evaluate the ability of LMMs to detect synthetic data across multiple modalities. LOKI encompasses video, image, 3D, text, and audio modalities, comprising 18K carefully curated questions across 26 subcategories with clear difficulty levels. The benchmark includes coarse-grained judgment and multiple-choice questions, as well as fine-grained anomaly selection and explanation tasks, allowing for a comprehensive analysis of LMMs. We evaluated 22 open-source LMMs and 6 closed-source models on LOKI, highlighting their potential as synthetic data detectors and also revealing some limitations in the development of LMM capabilities. More information about LOKI can be found at https://opendatalab.github.io/LOKI/.

---

*These authors contributed equally to this work.

†Corresponding author(s). E-mail(s): liweij29@mail.sysu.edu.cn, heconghui@pjlab.org.cn

# 1 INTRODUCTION

With the rapid development of diffusion models (Rombach et al., 2022; Dhariwal & Nichol, 2021b) and large language models (Abdullah et al., 2022; Brown, 2020), AI-generated content (AIGC) technology has increasingly integrated synthetic multimodal data into our daily lives. For instance, tools like SORA (Brooks et al., 2024) can produce highly realistic video, while Suno (Shulman et al., 2022) enables the creation of music at a level comparable to professional artists. However, synthetic multimodal data also brings significant risks, including potential misuse and societal disruption (Cooke et al., 2024; Ju et al., 2022). For example, the risks include generating fake news using large language models (LLMs), synthesizing fraudulent faces with diffusion models for scams, and potential contamination of internet training data. Due to the convenience of artificial intelligence synthesis, the future Internet may be saturated with AI-generated content, making the task of discerning the authenticity and trustworthiness of multimodal data increasingly challenging.

To address such threats, the field of synthetic data detection has garnered widespread attention in recent years (Barni et al., 2020; Frank et al., 2020; Gragnaniello et al., 2021; Shao et al., 2023; 2024).However, most current synthetic data detection methods are primarily focused on authenticity evaluation, with certain limitations regarding the human interpretability of the prediction results (Li et al., 2024b). The recent rapid advancement of large multimodal models (LMMs) has sparked curiosity about their performance in detecting synthetic multimodal data (Ku et al., 2023; Wu et al., 2024b). On one hand, for synthetic data detection tasks, LMMs can provide reasoning behind authenticity judgments in natural language, paving the way for enhanced explainability. On the other hand, the task of distinguishing between real and synthetic data involves the perception, knowledge, and reasoning abilities of multimodal data, serving as an excellent test of LMM capabilities. Therefore, the focus of this paper is to evaluate the performance of LMMs in synthetic data detection tasks.

However, traditional synthetic data detection benchmarks, such as Fake2M (Lu et al., 2023b) and ASVSpoof 2019 (Wang et al., 2020b), primarily assess conventional detection methods, and evaluations of LMMs in detecting multimodal synthetic data are still lacking. These benchmarks often miss fine-grained anomaly annotations represented in natural human language, making it difficult to transparently analyze the explainability capabilities of LMMs. FakeBench (Li et al., 2024a) aligns more closely with our objectives, but it only evaluates the performance of LMMs within a single standard image modality, lacking both breadth and depth. Specifically, FakeBench fails to explore other modalities such as audio and 3D data, focusing primarily on general image types and not conducting thorough tests on expert domain images like satellite imagery. To bridge this gap, we introduce LOKI, a comprehensive benchmark for evaluating the performance of LMMs on synthetic data detection. The key highlights of the LOKI benchmark include:

- *Diverse Modalities.* LOKI includes high-quality multimodal data generated by recent popular synthetic models, covering video, image, 3D data, text, and audio.

- *Heterogeneous Categories.* Our collected dataset includes 26 detailed categories across different modalities, such as specialized satellite and medical images; texts like philosophy and ancient Chinese; and audio data like singing voices, environmental sound and music.

- *Multi-level Annotations.* LOKI includes basic "Synthetic or Real" labels, suitable for fundamental question settings like true/false and multiple-choice questions. It also incorporates fine-grained anomalies for inferential explanations, enabling tasks like abnormal detail selection and abnormal explanation, to explore LMMs' capabilities in explainable synthetic data detection.

- *Multimodal Synthetic Evaluation Framework.* We propose a comprehensive evaluation framework that supports inputs of various data formats and over 25 mainstream multimodal models.

On the LOKI benchmark, we evaluated 22 open-source LMMs, 6 advanced proprietary LMMs, and several expert synthetic detection models. Our key findings are summarized as follows:

For *synthetic data detection tasks* we find: (1) LMMs exhibit moderate capabilities in synthetic data detection tasks, with certain levels of explainability and generalization, but there is still a gap compared to human performance; (2) Compared to expert synthetic detection models, LMMs exhibit greater explainability and, compared to humans, can detect features invisible to the naked eye, demonstrating promising developmental prospects.

For *LMMs capabilities* we find: (1) Most LMMs exhibit certain model biases, tending to favor synthetic or real data in their responses; (2) LMMs lack of expert domain knowledge, performing poorly on specialized image types like satellite and medical images; (3) Current LMMs show unbalanced multimodal capabilities, excelling in image and text tasks but underperforming in 3D and audio tasks; (4) Chain-of-thought prompting enhances LMMs' performance in synthetic data detection, whereas simple few-shot prompting falls short of providing the necessary reasoning support.

These findings highlight the challenging and comprehensive nature of the LOKI task and the promising future of LMMs in synthetic data detection tasks.

## 2 RELATED WORK

### 2.1 SYNTHETIC DATA DETECTION

Currently, synthetic data detection has garnered widespread attention to prevent the misuse of multimedia synthetic data (Gragnaniello et al., 2021; Hou et al., 2023). The detection of synthetic data in image and audio has long been a popular research (Barni et al., 2020; Frank et al., 2020), while methods for synthetic video detection have recently emerged, such as DuB3D(Ji et al., 2024) and AIGVDet(Bai et al., 2024a). However, most work primarily focuses on the binary distinction between authentic and synthetic data, resulting in poor interpretability. Some studies aim to enhance the interpretability of synthetic detection by providing latent representations(Dong et al., 2022), feature explanations(Chai et al., 2020), and artifact localization (Zhang et al., 2023a; Shao et al., 2023; 2024); however, most research remains limited to the interpretability of abstract symbols, leaving a significant gap in alignment with human understanding. In practice, current AI-generated synthetic data still exhibits noticeable flaws, such as discontinuities in synthetic videos and insufficient geometric accuracy in 3D data. These shortcomings can be effectively captured and perceived by human users(Tariang et al., 2024), who can provide reasonable explanations. However, existing expert synthetic data detection methods fail to provide human-interpretable bases for their judgments.

### 2.2 LARGE MULTIMODAL MODELS

Recently, the rapid development of multimodal large models (LMMs) has been notable, with models like GPT-4o (OpenAI, 2024) and Claude 3.5 (Anthropic, 2024) excelling in various tasks such as scientific questioning (Lu et al., 2022; Yue et al., 2024) and commonsense reasoning (Talmor et al., 2018), showcasing exceptional perceptual and reasoning abilities (Bai et al., 2024b). Research has also applied LMMs to evaluate AIGC synthetic results, utilizing GPT to assess the quality of generated images (Ku et al., 2023; Peng et al., 2024) and 3D models (Wu et al., 2024b), providing scores that align with human preferences along with interpretable justifications. Consequently, in synthetic data detection, LMMs can offer reasons for determining authenticity in natural language, paving the way for enhanced interpretability in synthetic detection. Moreover, LMMs can access features invisible to human users, such as deep image and spectral features, demonstrating their potential to exceed human detection capabilities. Furthermore, synthetic data detection involves multimodal data perception and complex logical reasoning, making it an excellent task to assess the capabilities of LMMs. This task also provides quantitative evaluation metrics like accuracy, allowing for a more direct assessment of model performance compared to more qualitative scoring tasks.

### 2.3 SYNTHETIC DATA DETECTION BENCHMARK

Currently, there are numerous datasets corresponding to synthetic data detection tasks, including those designed for traditional detection methods and those tailored for LMMs. For instance, traditional synthetic datasets such as Fake2M (Lu et al., 2023b), HC3 (Guo et al., 2023), and ASVSpoof 2019 (Wang et al., 2020b) have explored the performance of traditional deepfake detection methods across various modalities, but they lack assessments for LMMs models. VANE (Bharadwaj et al., 2024) evaluates the capability of LMMs in detecting video anomalies, including the detection of criminal activities in real videos and synthetic video detection, although it focuses more on video content understanding. Fakebench (Li et al., 2024b) assesses LMM performance in the image modality, yet it concentrates on a single modality and offers limited subcategories. In contrast, LOKI covers a broader range of data modalities, including video, image, 3D, text, and audio, as well as data from specialized fields such as remote sensing, medical imaging, and environmental sounds. In terms of problem design, LOKI encompasses tasks for authenticity judgment, as well as more complex challenges like Abnormal Details selection and Abnormal Explanation, which test the LMMs' ability to explain reasons in synthetic data detection.

## 3 DATASET

### 3.1 OVERVIEW OF LOKI

We introduce LOKI, a multimodal synthetic data detection benchmark, designed specifically to comprehensively assess the capabilities of LMMs in detecting synthetic data. As illustrated in Figure 2, LOKI encompasses a variety of modalities including video, image, 3D, text, and audio, with over 26 specific subcategories of data. The benchmark utilizes fine-grained anomaly annotations to construct a tiered variety of question types, including judgment questions, multiple-choice questions, abnomal detail selection and abnomal explanation questions, totaling over 18k questions.

Table 1 provides a detailed comparison of LOKI with existing datasets, including traditional synthetic detection benchmarks and those tailored for evaluating LMMs. In terms of breadth, LOKI covers a wider range of modalities and finer categories. In depth, it goes beyond binary judgment question designs to include questions that require a deep understanding and explanation of detailed anomalies. Additionally, LOKI classifies question difficulty based on human evaluation metrics.

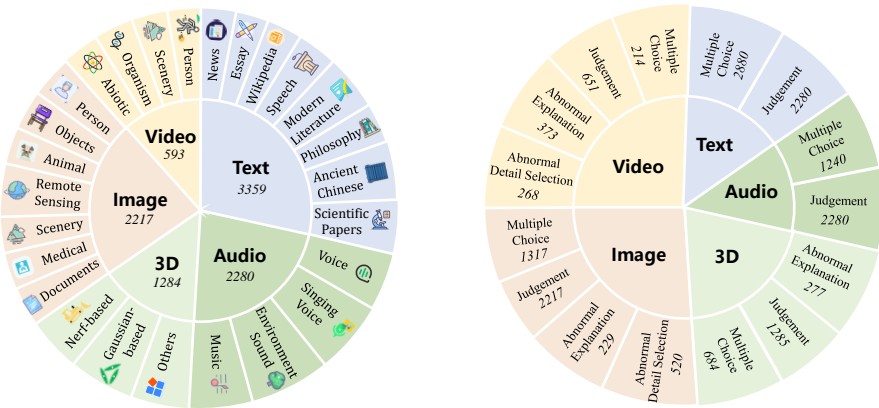

Figure 2: **Statistical information of LOKI.** The left side displays the detailed categories of each modality, while the right side presents the questions across different modalities. The inner circle numbers represent the data volume, and the outer circle numbers indicate the number of questions.

Table 1: The comparison between LOKI and other benchmarks. Answer types include JD (Judgment), MC (Multiple Choice), and OE (Open-ended). "Real paired" indicates whether real data is paired within the same domain, while "Difficulty Level" shows if questions are graded by difficulty.

| Dataset | Size | Category | Data Modality | | | | | Answer | | | Real Paired | Difficulty Level |
|---|---|---|---|---|---|---|---|---|---|---|---|---|
| | | | Img | Vid | Txt | Aud | 3D | JD | MC | OE | | |
| FFHQ | 70k | - | ✓ | | | | | ✓ | | | ✗ | ✗ |
| Fake2M | >1M | 8 types | ✓ | | | | | ✓ | | | ✓ | ✗ |
| HC3 | ~80K | 5 types | | | ✓ | | | ✓ | | ✓ | ✓ | ✗ |
| Mixset | 3.6 K | 5 types | | | ✓ | | | ✓ | | | ✓ | ✗ |
| ASVS2019 | 108K | - | | | | ✓ | | ✓ | | | ✓ | ✗ |
| Codecfake | ~1M | - | | | | ✓ | | ✓ | | | ✓ | ✗ |
| FakeBench | 6K | 6 types | ✓ | | | | | ✓ | | ✓ | ✗ | ✗ |
| VANE | 0.9K | - | | ✓ | | | | | ✓ | | ✗ | ✗ |
| LOKI | 18K | 26 types | ✓ | ✓ | ✓ | ✓ | ✓ | ✓ | ✓ | ✓ | ✓ | ✓ |

### 3.2 DATA COLLECTION AND ANNOTATION

**Video:** We collected 593 video clips by utilizing various closed-source and open-source models such as SORA (OpenAI, 2024), Keling, and Open_sora (Zheng et al., 2024), generating high-quality text-to-video synthesis data along with corresponding real domain sample data. For the AI-generated

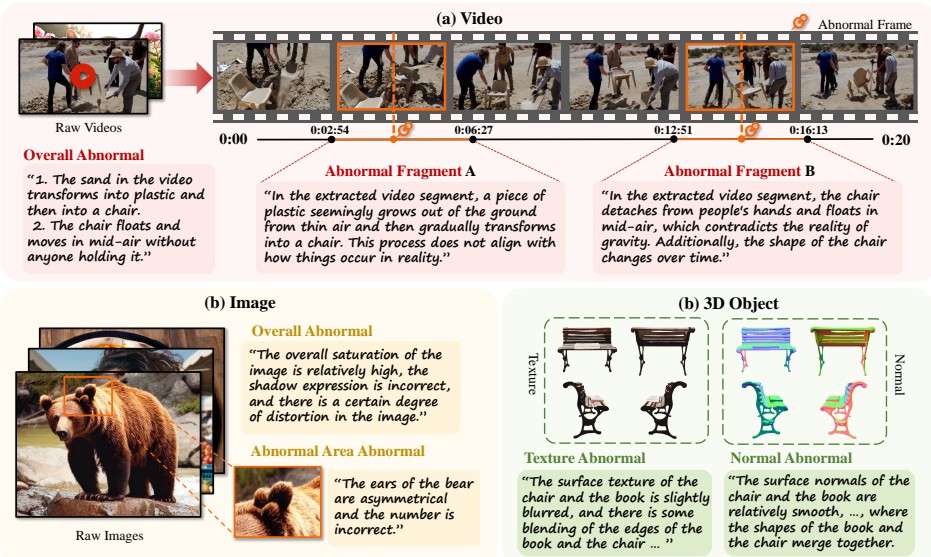

Figure 3: **Examples of Synthetic Data Annotations:** (a) Detailed annotations of video anomalies; (b) Detailed annotations of image anomalies; (c) Detailed annotations of 3D anomalies.

video clips, we employed the LabelU[1] tool to annotate anomaly details, including anomalous segments and their descriptions, anomalous key frames, and global anomaly descriptions. As shown in Figure 3 (a), anomalies in the videos, such as "violating natural physics" and "frame flickering," are also annotated globally. Additionally, the anomalous segment from 02:54 to 06:27 is highlighted, with the corresponding reasons for the anomalies explained by human annotators. Furthermore, each anomalous segment includes an anomalous key frame to facilitate subsequent LMMs in accurately reading the anomalous frames when processing video data.

**Image:** We have collected over 2,200 images from 7 subcategories through existing dataset extraction, internet collection, and new data synthesis. The image synthesis methods include FLUX, Midjourney (AI, 2023), Stable Diffusion (Blattmann et al., 2023), and ten other different methods to ensure high quality and diversity of the data. For the synthesized image data, in addition to overall annotations, we performed anomaly region bounding and explanations, as shown in Figure 3 (b). The region anomaly annotations allow for more fine-grained and specific labeling, which can be used for generating subsequent anomaly detail questions.

**3D data:** We conducted a comprehensive analysis of OmniObject3D (Wu et al., 2023), selecting scanned instances as ground truth within the same domain. By constructing prompt texts, we synthesized three Nerf models (Poole et al., 2022) and three 3D GS models (Tang et al., 2023), and supplemented them with results from the advanced commercial model Clay and some Nerf-based results from GPTEval3D (Wu et al., 2024b). We collected a total of over 1,200 3D models from ten different synthesis methods, including both synthesized and real scanned data. Additionally, as shown in Figure 3 (c), we performed texture anomaly description annotations corresponding to the RGB four views of the synthesized 3D data, as well as normal anomaly description annotations. Notably, besides the multi-view format, the 3D data also supports point clouds and panoramic videos.

**Audio:** We collected various categories of audio, including speech, singing voice, environmental sounds, and music. The speech and singing voice data ensured consistency in speaker timbre, sourced from the Logical Access part of ASVSpoof2019 (Wang et al., 2020b) and the CtrSVDD Benchmark, covering four generation paradigms: TTS, VC, SVS, and SVC. Environmental audio data came from DCASE 2023 Task 7, with real audio from the development set and synthetic audio generated using multiple methods from Track A. Music data were sourced from MusicCaps, with synthetic music generated based on descriptions using MusicGen (Copet et al., 2024), AudioLDM2-Music (Liu et al., 2024a), and Suno[2].

---

[1]LabelU: https://github.com/opendatalab/labelU

[2]Suno: https://suno.com/

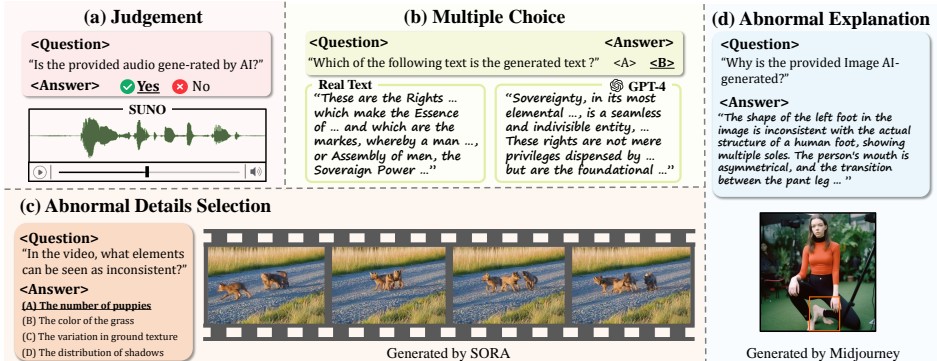

Figure 4: **Example Questions of LOKI.** LOKI includes four types of questions:(a) Judgment questions; (b) Multiple choice questions; (c) Abnormal detail selection; (d) Abnormal explanation.

**Text:** Based on summarization and regeneration methods, we generated counterfeit texts similar to the original texts using mainstream models such as GPT-4o, Qwen-Max, and Llama 3.1-405B (Bai et al., 2024c). We collected eight categories of text data, pairing each sample with a real text and a model-generated similar text, totaling 3,359 text entries. Our text data were categorized by length and language, including short texts (50-100 characters), medium texts (100-200 characters), and long texts (over 200 characters), with a 1:1 ratio of Chinese to English data. More information regarding the collection and statistics of each modality can be found in Appendix B.

## 3.3 QUESTION GENERATION

**Judgment Task:** This task requires large language models (LMMs) to determine whether the input data is synthetic or real. As shown in Figure 4 (a), LMMs need to answer the judgment question, "Is the provided audio generated by AI?" To minimize the influence of prompts on model judgments, questions are asked in two forms: whether the data is AI-synthesized or real, and identifying either the real or AI-synthesized data. Furthermore, we categorize the data into different difficulty levels based on human performance. If all tested human users (more than three) answer correctly, the task is classified as "easy"; if more than 50% answer incorrectly, it is classified as "hard"; all other cases fall into the "medium" category.

**Multiple Choice Task:** This task requires LMMs to correctly select AI-generated or real data from the provided synthetic and real data. As illustrated in Figure 4 (b), LMMs need to complete the multiple-choice question, "Which of the following texts is generated?" The design of this question benefits from our collection of real paired data within the same domain, effectively assessing LMMs' comparative analysis capabilities.

**Abnormal Detail Selection:** Based on fine-grained anomaly annotation data from modalities such as video, images, and 3D, we effectively design prompts and utilize GPT-4o to generate questions for Abnormal Detail Selection. As shown in Figure 4 (c), for video content's detail anomalies, we ask, "What elements can be seen as inconsistent?" By providing clear anomaly annotations, we can effectively reduce the hallucination phenomenon in GPT-4o, ensuring the quality of the questions. More details can be found in the supplementary materials.

**Abnormal Explanation:** Furthermore, we design open-ended abnormal explanation questions, requiring LMMs to independently identify anomalies and explain their reasons. As shown in Figure 4 (d), we ask, "Why is the provided image AI-generated?" It is worth to note that in real anomaly explanation tasks, the input does not include bounding boxes around anomalous areas. Tasks related to Abnormal Detail Selection and Abnormal Explanation can more precisely test whether LMMs genuinely perceive corresponding detail anomalies rather than guessing answers.

**Quality Control:** To mitigate the impact of hallucinations of GPT-4o during question generation in abnormal detail selection task, all samples in this task undergo manual reviews. Each question that involves GPT must pass through at least two rounds of verification by human users. A total of 20 users participated in the verification process, which took approximately 160 hours to complete.

## 4 EXPERIMENT

In this section, we evaluate various Language Model Multimodalities (LMMs) under our proposed LOKI evaluation framework, which includes both open-source and proprietary models, multimodal LMMs, Audio LMMs, and text-based LLMs. Our evaluations are conducted in a *zero-shot* setting. In the following subsections, we first introduce our evaluation models and the evaluation protocols. Next, we analyze the performance of existing LMMs in synthetic data detection tasks, comparing them with human users and expert models. We will then discuss the challenges and shortcomings faced by multimodal large models in the current task settings. Additionally, we explore the potential impact of few-shot or chain-of-thought prompting on this task.

### 4.1 BASELINES

**LMMs.** We evaluate 3 closed-source and 18 open-source LMMs across different model types and sizes. For closed-source models, we consider GPT-4o (OpenAI, 2024), Gemini-1.5-Pro (Team et al., 2023), Claude-3.5-Sonnet (Anthropic, 2024). Given that modality alignment in multimodal LMMs may lead to a decline in LLM performance on text-based tasks (Dai et al., 2024), we also selected pure text LLMs, such as LLaMA-3.1-405B (Team, 2024), Qwen-Max (Chu et al., 2023) and Mistral-Large (Mistral, 2024), to evaluate the text modality. In the evaluation of Audio LMMs, we selected high-performing open-source models such as Qwen-Audio (Chu et al., 2023) and SALMONN-7B (Sun et al., 2024). For proprietary models, we chose Gemini-Flash (Team et al., 2023), which supports audio input.

**Human Users.** We invited over 50 human users, including senior university students and regular users, to participate in the judgment and multiple-choice question tests for different modalities of synthetic data. Each question was tested by at least 3 users to ensure the robustness of the results. Additionally, we designed an online platform to distribute random questionnaires, and more than 200 users participated in the testing of 15 basic questions.

**Expert Models.** We selected recently open-sourced expert-level synthetic data detection methods and their corresponding weights for testing, including video detection (AIGVDet (Yang et al., 2024)), image detection (AIDE (Yan et al., 2024)), text detection (RADAR-Vicuna-7B (Hu et al., 2023)), and audio detection (AASIST (Jung et al., 2022)). Due to the limited availability of 3D synthetic data detection methods, 3D was not considered. Additionally, there is no overlap between the training sets of these methods and the LOKI test data, reducing the possibility of data contamination. We selected only a small number of expert models for evaluation, primarily to serve as references, similar to the role of human references.

**Evaluation Protocol.** *Data Input:* For the video modality, we utilize an 8-frame video clip along with corresponding questions as input. For 3D modal data, we employ the commonly used multi-view input method. Results based on surround video and point cloud inputs are also included in the supplementary materials. For other modalities, inputs are based on textual prompts combined with corresponding images, audio, and textual materials. During the evaluation, each model independently generates responses to questions without retaining any dialogue history.

*Evaluation Metric:* For judgement, multiple-choice and abnormal detail selection questions, we use the average accuracy rate as a metric. In addition to accuracy, we also calculate the Normalized Bias Index (NBI) based on recall rates to assess model bias. For open-ended questions regarding anomalous details, we use the GPT-4 model to assess the score of the responses. Further details on the calculation of evaluation metrics can be found in Appendix C.2.

*Evaluation Framework:* To standardize the evaluation of different LMMs and various input modalities for synthetic data detection, we propose a comprehensive multimodal evaluation framework. This framework provides support for various input modalities such as 3D point clouds, videos, images, audio, and text, while unifying APIs of over 25 mainstream LMMs, ensuring both model compatibility and consistency throughout the evaluation process.

### 4.2 SYNTHETIC DATA DETECTION RESULTS

In this section, we provide a comprehensive analysis of the performance of various LMMs and LLMs on synthetic data detection tasks using the LOKI dataset.

Table 2: Results of different models on the LOKI for Judgment and Multiple Choice questions. (a) Multimodal evaluation of LMMs; (b) Text evaluaion of LLMs; (c) Audio evaluation of Audio LMMs; * denotes the closed-source models.

(a) Multimodal evaluation of LMMs

| | Judgment | | | | | Multiple Choice | | | | |
|---|---|---|---|---|---|---|---|---|---|---|
| | Video | Image | 3D | Text | Overall | Video | Image | 3D | Text | Overall |
| Random Choice | 51.1 | 50.5 | 50.5 | 49.9 | 50.3 | 47.7 | 49.0 | 49.7 | 45.2 | 46.9 |
| Human (Medium) | 83.5 | 80.1 | 72.0 | 68.5 | 76.0 | 91.3 | 84.5 | 91,2 | 78.5 | 86.4 |
| Expert models | 53.1 | 63.1 | - | 72.1 | 62.8 | - | - | - | - | - |
| Phi-3.5-Vision | 56.8 | 52.5 | 50.0 | 49.4 | 52.2 | 58.2 | 44.0 | 59.6 | 42.0 | 50.9 |
| MiniCPM-V-2.6 | 57.2 | 44.8 | 56.4 | 49.4 | 52.0 | 52.8 | 49.8 | 50.7 | 48.9 | 50.6 |
| InternLM-XComposer2.5 | 58.4 | 46.4 | 43.9 | 52.6 | 50.3 | 56.3 | 51.0 | 48.0 | 40.5 | 49.0 |
| mPLUG-Owl3-7B | 55.3 | 45.9 | 49.9 | 53.6 | 51.1 | 60.3 | 52.5 | 49.9 | 50.0 | 53.1 |
| LongVA-7B | 60.4 | 46.2 | 49.9 | 48.6 | 51.7 | 57.5 | 51.6 | 61.4 | 48.9 | 52.6 |
| Qwen2-VL-7B | 59.5 | 47.8 | **72.3** | 48.9 | **57.1** | 64.0 | 65.1 | 55.5 | 46.4 | 57.7 |
| LLaVA-OV-7B | 56.8 | 49.8 | 68.4 | 53.0 | 57.0 | 59.8 | 51.7 | 53.8 | 48.4 | 53.4 |
| Llama-3-LongVILA-8B | 51.9 | 49.8 | 32.2 | 49.9 | 46.0 | 54.0 | 51.1 | 50.5 | 44.3 | 50.0 |
| Idefics2-8B | 54.8 | 45.0 | 38.4 | 47.2 | 46.3 | 55.6 | 51.3 | 54.2 | 37.0 | 49.5 |
| Mantis-8B | 55.4 | **54.6** | 50.0 | 52.0 | 53.0 | 47.9 | 61.5 | **62.5** | 48.4 | 55.1 |
| InternVL2-8B | 60.8 | 49.7 | 49.4 | 50.3 | 52.6 | 54.0 | 51.4 | 53.1 | 46.6 | 51.3 |
| InternVL2-26B | 55.0 | 44.3 | 50.4 | 51.1 | 49.9 | 62.4 | 48.5 | 48.8 | 50.3 | 53.2 |
| InternVL2-40B | **62.0** | 49.6 | 49.9 | 53.1 | 52.2 | **65.7** | 63.1 | 59.9 | 45.2 | 52.7 |
| VILA1.5-13B | 51.9 | 49.3 | 34.0 | 47.7 | 45.7 | 52.1 | 55.3 | 53.5 | 44.0 | 51.2 |
| VILA1.5-40B | 59.2 | 48.8 | 50.0 | 50.1 | 52.7 | 49.1 | 64.0 | 47.9 | 50.4 | 53.7 |
| Qwen2-VL-72B | 59.6 | 53.2 | 60.3 | 52.8 | 55.4 | **65.7** | 68.6 | 58.7 | **69.7** | **65.6** |
| LLaVA-OV-72B | 56.5 | 46.3 | 51.3 | **61.2** | 56.3 | 62.9 | **70.8** | 61.3 | 69.2 | 65.2 |
| Claude-3.5-Sonnet* | 61.7 | 53.6 | 58.0 | 61.5 | 61.6 | 60.5 | 65.5 | 51.9 | **89.2** | **74.8** |
| Gemini-1.5-Pro* | 58.5 | 43.5 | 55.4 | 55.7 | 53.2 | 66.1 | 67.3 | 60.2 | 57.3 | 62.7 |
| GPT-4o* | **71.3** | **63.4** | **65.2** | 55.9 | **63.9** | **77.3** | **80.8** | **70.2** | 66.6 | 73.7 |

(b) Text evaluation of LLMs

| Model | Judgment | Choice | Overall |
|---|---|---|---|
| Human (Medium) | 69.2 | 71.1 | 70.1 |
| Expert model | 69.4 | - | 69.4 |
| LLaMA-3.1-405B | 56.8 | 73.1 | 64.4 |
| Mistral-Large* | 52.2 | 69.1 | 57.8 |
| Qwen-Max* | 48.3 | 44.4 | 46.5 |
| Claude-3.5-Sonnet* | **61.5** | **89.2** | **70.7** |
| Gemini-1.5-Pro* | 55.7 | 57.3 | 56.2 |
| GPT-4* | 55.9 | 66.6 | 59.5 |

(c) Audio evaluation of Audio LMMs

| Model | Judgment | Choice | Overall |
|---|---|---|---|
| Human (Medium) | 69.2 | 71.1 | 70.1 |
| Expert model | 69.4 | - | 69.4 |
| Qwen-Audio | 49.8 | 50.1 | 49.9 |
| SALMONN-7B | **51.2** | - | **51.2** |
| AnyGPT | 49.8 | **50.3** | 50.1 |
| OneLLM | 49.9 | - | 49.9 |
| LUT | 44.4 | - | 44.4 |
| Gemini-1.5-Flash* | 49.4 | 49.2 | 49.3 |

**Judgment and Multiple Choice.** Table 2 illustrates the performance of various models on judgment and multiple-choice questions in LOKI. For the synthetic data judgment task, the closed-source model GPT-4o achieves the best results, with an overall accuracy (excluding audio) of 63.9%. When real paired data is included for comparison in the multiple-choice questions, accuracy further increases to 73.7%. In the text modality, Claude-3.5 outperform other LMMs and LLMs, achieving accuracies exceeding 70%. In the Audio LMMs category, both open-source and closed-source models show performances comparable to random selection, which is not satisfactory.

**Abnormal Detail Selection and Explanation.** We compared the performance of different models on the tasks of abnormal detail selection and abnormal reason explanation, as shown in Table 3. GPT-4o achieved an accuracy exceeding 75% in abnormal detail selection and a score over 70% in abnormal reason explanation. This indicates that advanced LMMs like GPT-4o has demonstrated strong detail understanding capabilities, effectively analyzing and interpreting "synthetic traces." Notably, we observe that Claude-3.5-Sonnet (Anthropic, 2024) tends to misclassify synthetic images as real, despite the primary goal of our tasks being to explain abnormalities in synthetic images. More examples of abnormal explanations can be found in Appendix F.

**Comparing Humans and Expert Models.** Humans exhibit an average performance of 76% in judgment tasks and 86.4% in multiple-choice questions, both 10% higher than the LMM method. No-

Table 3: Results of different models on the LOKI for Abormal Details Selection and Abnormal explanation questions.[*] denotes the closed-source models.

| | Abnormal Details Selection | | | Abnormal Explanation | | | |
|---|---|---|---|---|---|---|---|
| | Video | Image | Overall | Video | Image | 3D | Overall |
| LLaVA-OV-7B | 76.9 | 18.8 | 43.1 | 46.7 | 68.9 | 71.0 | 62.0 |
| Qwen2-VL-7B | **79.4** | 31.5 | 51.5 | 48.4 | 63.8 | 73.4 | 61.9 |
| InternVL2-8B | 66.8 | 70.2 | 68.8 | 46.5 | 72.2 | 71.3 | 63.0 |
| Gemini-1.5-Pro[*] | 58.7 | 40.0 | 47.8 | 57.6 | **77.1** | 70.8 | 68.1 |
| Claude-3.5-Sonnet[*] | 50.9 | 19.8 | 32.8 | 50.1 | 1.7 | **78.2** | 45.8 |
| GPT-4o[*] | 74.0 | **76.2** | **75.3** | **67.6** | 72.9 | 77.0 | **72.6** |

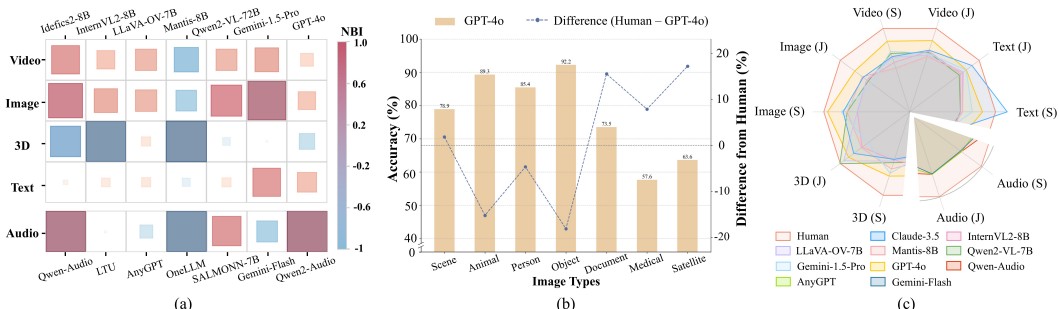

Figure 5: **The multimodal large model capability assessment analysis results.** (a) Model bias assessment, where the closer the color is to red, the more the model is biased towards classifying the data as real; the closer to blue, the more it leans towards synthetic data. The size of the square also represents the degree of bias. (b) The performance of GPT-4o across different image types and its difference from human users. (c) A relative radar chart of the model's performance across various modalities, with Human benchmarks for comparison.

tably, if LMM tools are to be applied in production, their decision-making performance in judgment tasks must exceed 90% to be convincing. As synthesis technologies advance, the distinct "traces" of synthetic data are becoming increasingly subtle. However, LMMs capture minute details, such as image features imperceptible to the human eye, demonstrating their potential to surpass human.

LMMs demonstrate superior performance in most tasks compared to expert models. This is primarily due to the rich and diverse sources of synthetic data collected by LOKI, which significantly differ from existing data domains, resulting in suboptimal generalization performance of expert models. The accuracy of synthetic detection by expert models trained on similar data should significantly improve. Currently, LMMs perform at a moderate level in synthetic data detection but surpass expert models in generalization ability. Unlike traditional expert models, LMMs possess the capability to explain the reasons behind anomalies, highlighting their unique advantage as synthetic detectors.

## 4.3 LARGE MULTIMODAL MODELS CAPABILITIES

**Model Bias.** The heatmap of the Normalized Bias Index calculated based on recall rates, as shown in Figure 5 (a), is utilized for analyzing model biases. The results indicate that most models exhibit significant biases in synthetic data detection tasks, with a tendency to incorrectly categorize data as either real or synthetic. For instance, GPT-4o tends to classify textual data as real, whereas it is biased towards judging 3D data as AI-generated. Despite diverse questioning techniques implemented to minimize cueing effects, a pronounced bias is still evident across most models.

**Lack of Expert Domain Knowledge.** In Figure 5 (b), we present the varying performance of GPT-4o across different image subcategories. The experimental results clearly indicate that GPT exhibits strong recognition abilities on common image types such as objects and landscapes, even surpassing human users. However, GPT-4o's performance significantly deteriorates in specialized fields such as satellite and medical imaging, and in less-trained image types like documents. This suggests that current LMMs still lack certain expert domain knowledge.

Table 4: Result decomposition across questions difficulty levels.

| Difficulty Levels (Video & Image & 3D & Text) | | | |
| | Easy (2470) | Medium (1104) | Hard (3938) | Overall (7512) |
| --- | --- | --- | --- | --- |
| LLaVA-OV-7B | 60.4 | 47.6 | 39.1 | 47.3 |
| InternVL2-8B | 64.5 | 47.8 | 33.5 | 45.7 |
| Qwen2-VL-7B | 67.7 | 45.6 | 35.2 | 47.4 |
| Gemini-1.5-pro | 70.8 | 42.4 | 32.4 | 46.4 |
| Claude-3.5-Sonnet | 76.0 | 44.7 | 29.8 | 47.1 |
| GPT-4o | 78.8 | 52.3 | 44.4 | 56.8 |

Table 5: LMMs' performances under different prompting strategies for judgement tasks.

| Prompting Strategies Performances (Image & 3D) | | |
| | Baseline | FS Few-shot | CoT Chain-of-Thought |
| --- | --- | --- | --- |
| LLaVA-OV-7B | 56.6 | 46.4 | 18.8 |
| InternVL2-8B | 49.6 | 46.1 | 50.4 |
| Qwen2-VL-7B | 56.8 | 52.6 | 59.5 |
| Gemini-1.5-pro | 47.9 | 41.2 | 51.0 |
| Claude-3.5-Sonnet | 55.2 | 53.7 | 56.4 |
| GPT-4o | 64.1 | 75.1 | 74.2 |

**Unbalanced Multimodal Capabilities.** In Figure 5 (c), we compare the performance of various LMMs across different modalities. Results indicate that current models excel in frequently trained modalities such as images and text, even surpassing human performance in some tests. However, their performance declines significantly on audio tasks, with most open-source models lacking corresponding capabilities. For future AGI to develop into a versatile assistant, it needs to possess more balanced multimodal abilities.

**Model Performance across Different Levels.** Based on human user performance, we categorized the difficulty levels of the questions, as shown in Table 4, which presents the performance of selected models across different difficulty levels. As the difficulty increases, the performance of LMMs gradually declines, consistent with human user performance. Under challenging conditions, GPT-4o's accuracy drops to only 44.4%, which is lower than that achieved by random selection. This indicates that LMMs have certain limitations in handling complex synthetic data detection tasks.

**Prompting Strategies Impact LMMs Capabilities.**

In Table 5, we demonstrate the effects of different prompting strategies in LOKI's image and 3D judgement tasks, where CoT refers to the Chain of Thought prompting (Wei et al., 2022b) and FS refers to the few-shot prompting (Alayrac et al., 2022). During inference, models are prompted with two random examples that are in the same domain as the questions by different strategies. In CoT prompting, we manually craft "thought chains" with our human annotations to elicit reasoning steps out of LMMs, while in FS prompting, we simply prepend examples with answers to the questions. Interestingly, GPT-4o shows strong reasoning ability without chain-of-thought prompting, while other models rely on it for improved performance. Few-shot learning fails to support the necessary step-by-step reasoning for synthetic data detection, but GPT-4o performs well regardless, suggesting its inherent ability to reason effectively without additional reasoning guidance. However, LLaVA-OV-7B experienced significant performance drop when prompted with CoT. We conjecture that this degradation may result from a decline in its ability to understand long contexts after fine-tuning (Zhai et al., 2023). More CoT experimental results are available in Appendix E.3.

## 5 CONCLUSION

In this paper, we introduced LOKI, a multimodal benchmark designed to evaluate the performance of large multimodal models in detecting synthetic data across various modalities. We conducted a comprehensive study of LMMs' performance on video, image, 3D, audio, text, and specialized subdomains, and we also analyzed LMMs' ability to explain detailed anomalies in synthetic data. The experimental results indicate that LMMs have a certain level of competence in detecting synthetic data and a preliminary ability to explain anomalies. Synthetic data detection tasks also effectively evaluate the various capabilities of LMMs during their development. These findings highlight the challenging and comprehensive nature of the LOKI task, as well as the potential of LMMs in future synthetic data detection tasks. We aim to inspire more powerful and interpretable synthetic data detection methods through LOKI to address the potential risks posed by rapidly advancing AI synthesis technologies. Furthermore, while the relationship between synthesis and detection is adversarial, they are mutually beneficial; better and more explainable synthetic detectors will further advance AI synthesis technologies.

ACKNOWLEDGMENTS

This project was funded by National Natural Science Foundation of China (Grant No. 42201358) and Shanghai AI Laboratory. Additionally, this work is partially supported by the NSFC under Grant 62376237, Shenzhen Science and Technology Program ZDSYS20230626091302006, and Internal Project Fund from Shenzhen Research Institute of Big Data (Grant No. T00120230002).

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

# LOKI: A Comprehensive Synthetic Data Detection Benchmark using Large Multimodal Models

## Supplementary Material

## Table of Contents in Appendix

## A    SYNTHETIC DATA DETECTION

In this appendix, we introduce and discuss the social impacts of synthetic data, such as deepfakes, as well as data contamination introduced by synthetic data. Finally we discuss the increasing attention on synthetic data detection.

### A.1    SOCIAL IMPACT OF SYNTHETIC DATA

While synthetic data generated by AIGC technology has offered numerous benefits to various aspects of society, it has also introduced significant challenges and risks. One of the most concerning risks is the potential to use synthetic data to create deepfakes. All forms of synthetic data can be leveraged to generate deepfakes, which can then be used to deceive, manipulate, or defraud individuals or organizations (see Fig.6). For instance, synthetic text data can be exploited to create fake news Papageorgiou et al. (2024), phishing emails, or manipulative advertisements. Similarly, synthetic image and 3D data can be used to generate realistic fake facesXu et al. (2023), scenes, or even content that leads to copyright violations Jiang et al. (2023b); Somepalli et al. (2023). Synthetic video data poses a threat by enabling the production of fake videos or fake news (e.g., political propaganda Pawelec (2022)), as well as deepfake video fraud calls Mustak et al. (2023). Likewise, synthetic audio data can be used for fake calls, voice cand even fake broadcasts. Furthermore, the advancements in synthetic data technologies are also impacting employment in creative industries, exemplified by the months-long strikes in the film industry Bohacek & Farid (2024).

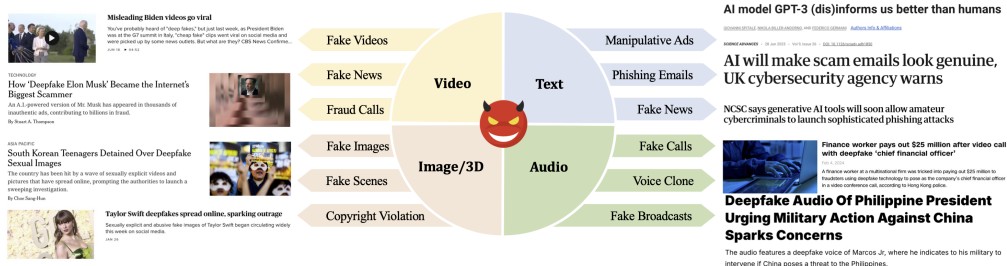

Figure 6: Soical impact of synthetic data across different modalities

### A.2    SYNTHETIC DATA CONTAMINATION

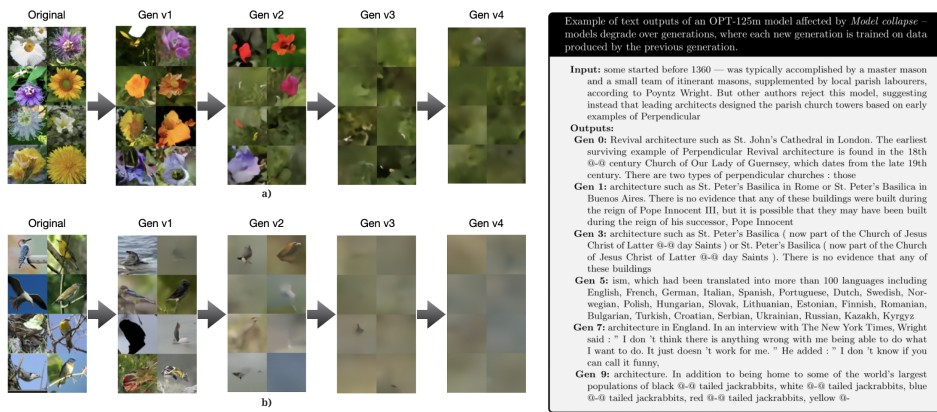

Figure 7: Model Performance Collapse Trained On Synthetic Data (Image from Martínez et al. (2023), Text from Shumailov et al. (2024) )

In today's LLM era, the internet is flooded with a substantial amount of synthetic data, even existing web-scale datasets are known to contain synthetic content Schuhmann et al. (2022). According to openai Altman (2024), they now generate about 100 billion words per day, while all people on

earth generate about 100 trillion words per day. All of this points to the fact that synthetic data will dominate the internet data side.

The use of synthetic data has been shown to significantly degrade the performance of deep learning models (see Fig.7), both for generation tasks and classification tasks Hataya et al. (2023); Ravuri & Vinyals (2019); Martínez et al. (2023); Shumailov et al. (2024); Bohacek & Farid (2023). Addressing the impact of synthetic data is crucial for the development of the next generation of models. There are two primary approaches to mitigating the negative effects of synthetic data. The first is exploring ways to better utilize synthetic data, proposing strategies for optimizing the integration of synthetic data into training pipelines Dohmatob et al. (2024a;b); Feng et al. (2024); Bertrand et al. (2024); He et al. (2023). The second approach involves developing methods to accurately detect synthetic data, allowing models to distinguish between real and synthetic inputs.

## A.3 INCREASING ATTENTION ON SYNTHETIC DATA DETECTION

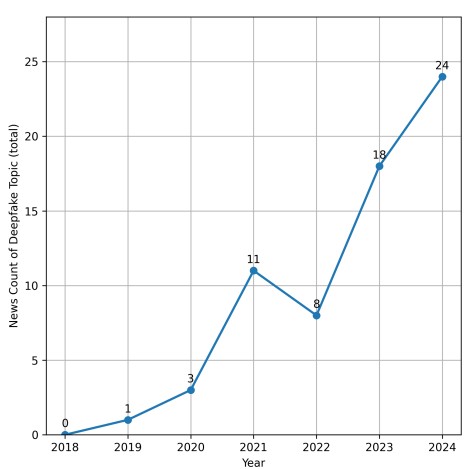

(a) The number of BBC official news reports on deepfake topic over the years.

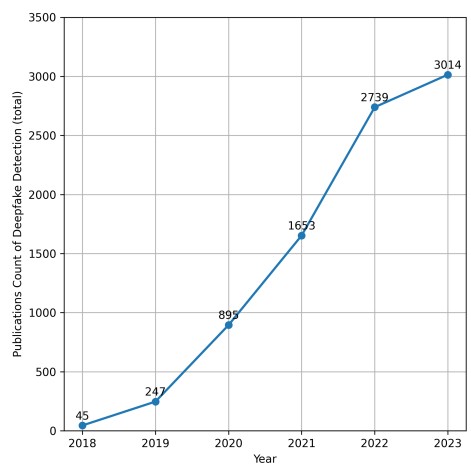

(b) The number of publications on deepfake detection over the years.(From Gong & Li (2024))

Figure 8: The rising concern of deepfakes in both media and academic research.

The growing prevalence of synthetic data has garnered increasing attention from society, including news reports, academic research, and government policies. The number of papers on deepfake detection has been steadily increasing, with the BBC reporting on deepfakes more frequently each year (see Fig.8). In response to the rise of synthetic data, several governments and global conferences have also introduced policies aimed at regulating the use of deepfakes and synthetic data AIS (2023); Michelle (2023); Biden (2023).

# B  DATASET DESCRIPTION

## B.1  DATA COLLECTION

Our data primarily originates from online internet collections, reused from public datasets, and self-synthesized into new composite data, as detailed in Table 6. To ensure diversity in synthetic data, each modality incorporates more than five different synthesis methods (Figure 9 & 10). To guarantee the quality of synthetic data, we also collected samples synthesized by mature proprietary models such as Sora, Midjourney, CLAY, Suno, and GPT-4. The far-right column of the table displays the public datasets that underpin our collected synthetic or authentically paired data.

Table 6: Synthetic Methods and Public Datasets Across Modalities

| Modality | Synthesis Methods | Public Datasets |
|---|---|---|
| Video | Sora (OpenAI, 2024), Keling, CoNo, Lumiere(Bar-Tal et al., 2024), Open-sora (Zheng et al., 2024), Runway, W.A.L.T (Gupta et al., 2023) | - |
| Image | FLUX, DDIM (Song et al., 2020a), Midjourney (AI, 2023), Stable Diffusion (V1.4,V1.5,V2.1) (Blattmann et al., 2023), DPM+ (Lu et al., 2023a), ADM(Dhariwal & Nichol, 2021a), Stylegan (Karras et al., 2019a), Skydiffusion (Ye et al., 2024), pix2pix (Isola et al., 2017), CUT (Park et al., 2020) | I2IQA(Yuan et al., 2023), Sentry(Lu et al., 2023c), GenImage(Zhu et al., 2023), FFHQ(Karras et al., 2019b), Stylegan3(Karras et al., 2021), CVUSA(Workman et al., 2015), ISBI 2016(Gutman et al., 2016), M3DSynth(Zingarini et al., 2024), M6Doc(Cheng et al., 2023), Deepfakeface(Song et al., 2023), VIGOR(Zhu et al., 2021) |
| 3D | CLAY (Zhang et al., 2024a), SyncDreamer (Liu et al., 2023c), Magic3D (Lin et al., 2023), DreamFusion (Poole et al., 2022), Fantasia3D (Chen et al., 2023a), DreamGaussian (Tang et al., 2023), Wonder3D (Long et al., 2024), GaussianDreamer (Yi et al., 2023), GradeADreamer (Ukarapol & Pruvost, 2024) | OmniObject3D(Wu et al., 2023), GPTEval3D(Wu et al., 2024a) |
| Audio | Suno, WaveNet (Rethage et al., 2018), WaveRNN (Kalchbrenner et al., 2018), Tacotron2 (Shen et al., 2018), Hifi-GAN (Kong et al., 2020), AceSinger (Timedomain, 2023), Soft-VITS-SVC (svc-develop team, 2024), DiffSinger (Dhariwal & Nichol, 2021b), VQ-VAE (Van Den Oord et al., 2017), AudioLDM (Liu et al., 2023a), VITS (Kim et al., 2021), AudioLDM2 (Liu et al., 2024a), MusicGen (Copet et al., 2024) | ASVSpoof2019(Wang et al., 2020b), CtrSVDD(Zang et al., 2024), DCASE2023 Track 7(Choi et al., 2023), MusicCaps(Agostinelli et al., 2023) |
| Text | llama3.1-405B (Team, 2024) , GPT-4o (OpenAI, 2024), Qwen-Max (Bai et al., 2023), Mistral-Large (Jiang et al., 2023a), Claude-3.5-Sonnet (Anthropic, 2024), Gemini-1.5-Flash (Team et al., 2023) | TheDataBeast (TheDataBeast, 2021),Mixset(Zhang et al., 2024b), NLP_chinese_corpus(Xu, 2019), ghostbuster-data(Verma et al., 2024) |

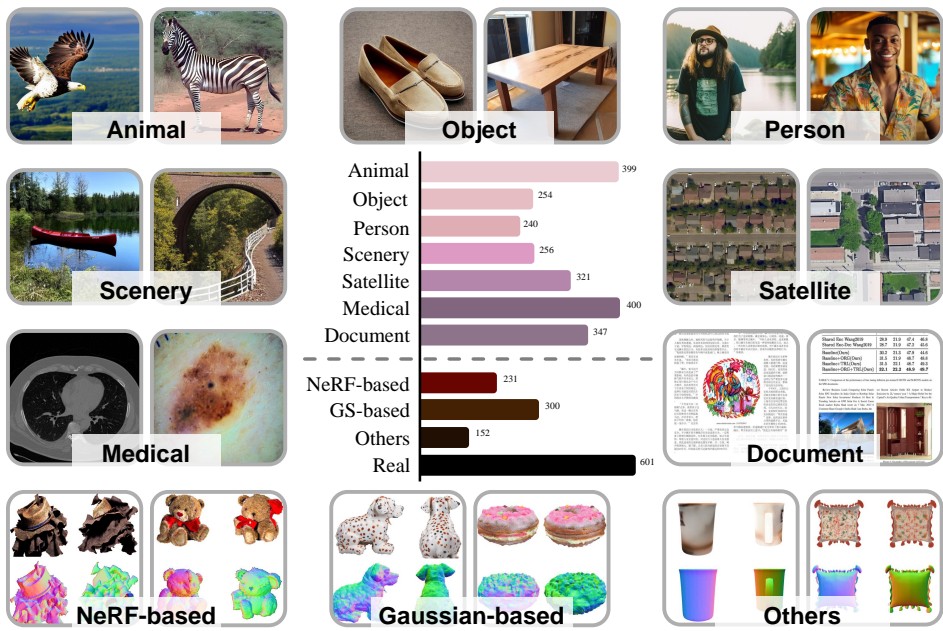

Figure 9: Examples of some 3D and image datasets, with the bar chart showing the quantity of data in different categories.

**Authentic pairing data:** We have collected a significant amount of authentic paired data from the internet, including sources such as arXiv, Wikipedia, Gutenberg, YouTube, TikTok, and Civitai. For data sourced from the internet, we will rigorously verify that it consists of authentic recordings or text authored by human users, rather than content synthesized using AI technology. It is important to note that our current research primarily focuses on multimedia data directly synthesized by AI, with limited consideration of methods like deepfake involving manual editing; we will continue to update our approach in future studies.

**Data Availability and Social Impact:** In collecting data, we strictly adhere to copyright and licensing regulations of the source websites, avoiding data acquisition from resources that prohibit copying or redistribution. For the LOKI dataset, which is open-sourced, users must submit a download request to the authors to prevent misuse of the data.

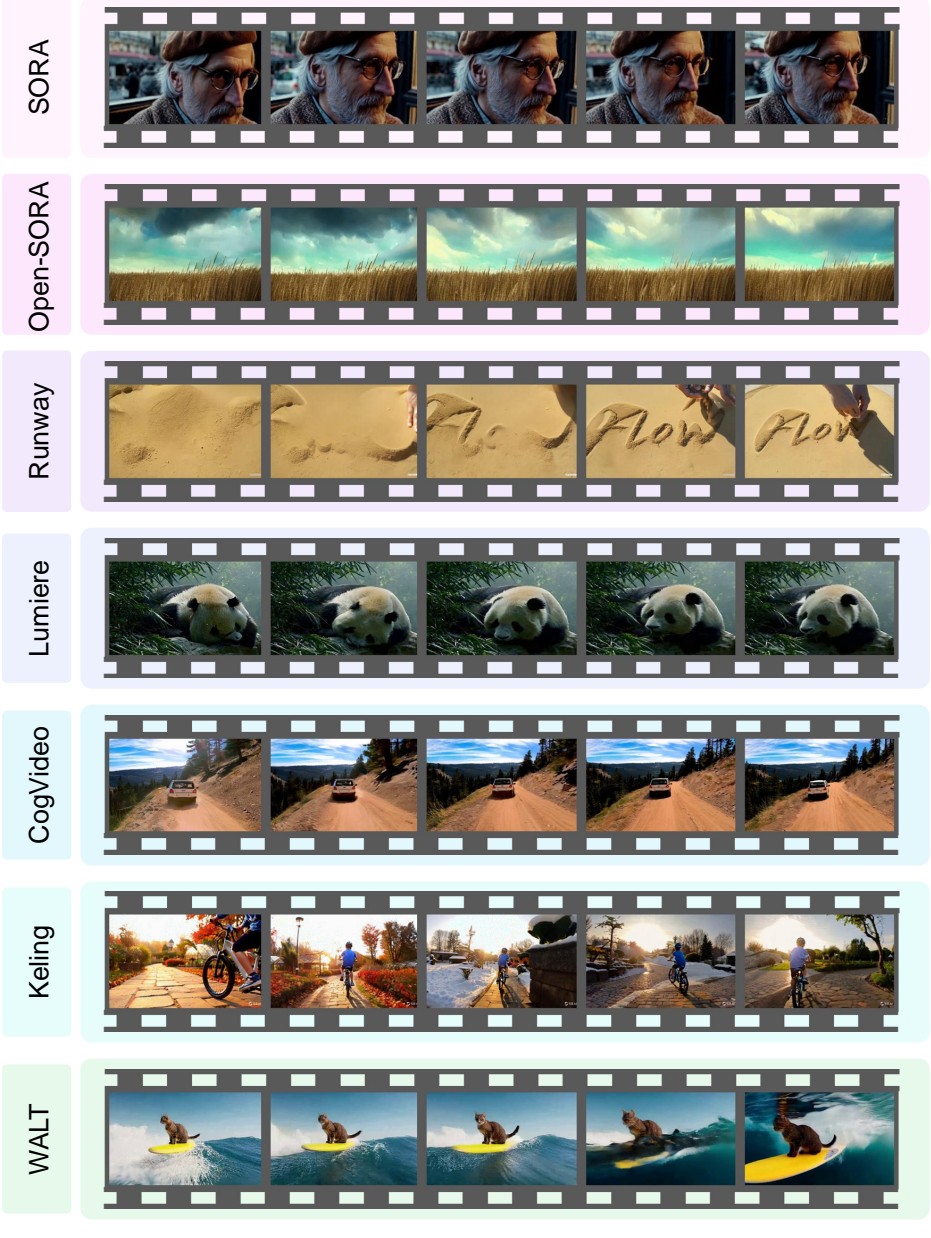

Figure 10: Examples of video data. We used 7 video generation models to obtain corresponding data for LMMs evaluation.

## B.2 DATASET ANNOTATION

### B.2.1 ANNOTATION GUIDELINES

**Video:** During the annotation of synthetic videos, we categorize the identified anomalies into two types: global anomalies and segment anomalies. Global anomalies refer to errors that persist for more than 80% of the video's duration, while segment anomalies are issues that occur for a limited portion of the video. For example, as shown in Fig. 11 (a), the anomaly of "flickering textures and distorted geometries of fences and utility poles," which is present throughout the video, is labeled as a global anomaly. In contrast, the "abnormal flames" and "basketball penetrating the hoop" in the video are classified as segment anomalies. Additionally, each identified anomaly, including both global and segment anomalies, is associated with a key frame that represents the anomaly, facilitating subsequent processing of video data by large multimodal models (LMMs).

**Image:** For the synthetic image data, we provide global anomaly annotations for overall image issues, as well as bounding box selections and textual descriptions for abnormal regions. The bounding boxes indicate the location and extent of the abnormal areas within the image, while the textual descriptions detail the specific anomalies present in those regions. As shown in Fig. 11 (b), the "texture quality issues" and "color distortion" in the image are annotated as global anomalies, whereas area errors such as the "texture errors" of the duck and "reflection anomalies" on the water surface are classified as region anomalies. Annotators mark these areas by drawing bounding boxes and provide textual explanations for the reasons behind the anomalies.

**3D Data:** Unlike video and image annotations, the annotation of 3D data involves a global-scale analysis of textures and normals. In terms of texture anomalies, we focus on assessing the authenticity, smoothness, and edge clarity of the textures. For normals, we analyze whether the model's geometric fluidity, surface smoothness, physical stability, and topological coherence are accurately represented. For instance, as shown in Fig. 11 (c), we conduct a detailed analysis of the "multiview discrepancies" and "texture blurriness" in the model's textures, while labeling issues such as "abnormal protrusions" and "asymmetrical structures" as related to normals, accompanied by appropriate textual explanations.

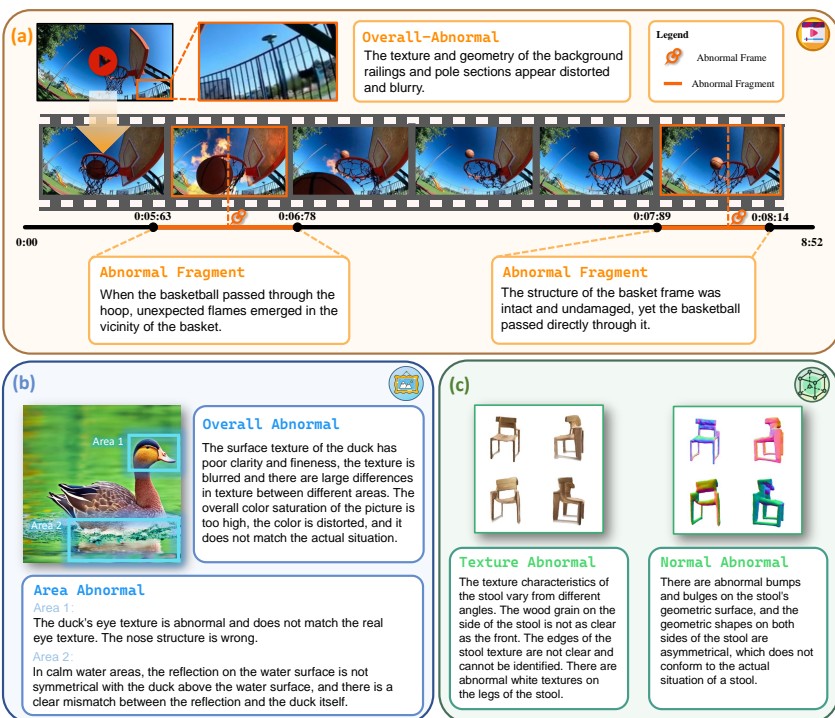

Figure 11: Examples of the synthetic data annotation process under different modalities, including (a) Video, (b) Image, (c) 3D Data.

### B.2.2 Annotator Informed Consent

Before commencing the annotation process, we ensure that all participating annotators are fully informed and provide their explicit agreement to the following terms and conditions. This comprehensive informed consent is designed to promote transparency, respect their autonomy, and align with ethical standards in research. It is imperative that each annotator has a thorough understanding of the nature, purpose, and potential implications of their contributions to the labeling process. The terms are outlined as follows:

**Data Usage.** Annotators acknowledge and consent to the possibility that the labeled data they generate may be used in various academic and scientific contexts, including the development of research papers, presentations at conferences, and other related scholarly activities. They understand that their work may significantly contribute to advancements in research fields such as natural language processing, machine learning, and artificial intelligence. Furthermore, annotators recognize that their contributions may be referenced or cited in scientific publications, thereby playing a role in shaping future research directions and applications.

**AI-Generated Content.** Annotators are informed that some of the content they will be labeling may have been produced by artificial intelligence models. This includes text, images, or other data types generated by algorithms designed to simulate human-like outputs. Annotators understand that this knowledge is crucial, as it may influence their perception, judgment, and approach to the labeling task. They agree to remain mindful of the potential biases or preconceived notions that may arise from this awareness and commit to maintaining objectivity and accuracy in their work.

**Potential Implications.** Annotators are aware of the broader implications of their labeling activities, which extend beyond the immediate scope of data annotation. They recognize the ethical considerations inherent in AI research, particularly concerning issues such as bias, fairness, and the societal impact of deploying AI technologies. Annotators agree to reflect on these ethical dimensions and to engage in the labeling process with a conscientious approach, acknowledging that their work may contribute to both the positive advancements and challenges associated with AI development and implementation.

**Commitment to Ethical Standards.** By agreeing to these terms, annotators affirm their commitment to upholding high ethical standards throughout the annotation process. They understand that their participation is voluntary and that they have the right to withdraw from the project at any time, without penalty. Annotators also acknowledge their responsibility to report any concerns or issues that may arise during the labeling process, ensuring the integrity and reliability of the data they provide.

This informed consent process ensures that all annotators are equipped with a comprehensive understanding of their role and its significance. It aims to foster an environment of mutual respect and collaboration, where the contributions of annotators are valued and their rights as participants in research are protected. By clarifying the expectations and responsibilities involved, we seek to create a foundation for ethical and impactful research that benefits both the scientific community and society at large.

### B.3 Quality Control and Validation

In annotating videos, images, and 3D synthetic data for anomaly details, we maintain high standards and accuracy. All annotators possess at least a university degree and demonstrate strong decision-making and judgment skills. Before annotation, human annotators receive extensive training with numerous examples of common errors to ensure a comprehensive understanding of the synthetic data detection task. Each data instance is annotated for detailed anomalies by at least two human annotators to ensure quality. Ambiguous or unclear instances are marked for further study and annotation during team meetings. Furthermore, to avoid the impact of hallucinations from Large Language Models (LMMs) on tasks involving LMMs, all anomaly detail explanation tasks undergo manual review based on LabelLLM [3].

---

[3]https://github.com/opendatalab/LabelLLM

### B.4 SPECIAL DATA DESCRIPTION

In the field of **document images**, we have collected synthesized images in four categories: newspapers, academic papers, magazines, and reconstructed documents. The corresponding real data for these categories come from the M6Doc dataset. Currently, document synthesis primarily follows a layout-first, content-rendering-later approach. For the first three document types, we design specific empirical rules for layout generation during the layout phase; during the content rendering phase, elements are selected from a constructed corpus to fill the structure. For the reconstructed type, we employ a restructuring algorithm on the M6Doc dataset to rearrange the content of the documents.

In the field of **remote sensing** imagery, synthetic data is primarily generated based on street-to-satellite datasets such as CVUSA and VIGOR, utilizing methods like CUT and Skydiffusion. The synthetic imagery encompasses major natural scenes such as urban and suburban environments, with the satellite remote sensing images being of high resolution.

In the field of **medical** imaging, we primarily collected two types of data: the ISIC 2016 skin dataset and the M3DSynth CT dataset. For the ISIC 2016 dataset, we utilized GAN methods for direct data synthesis. The M3DSynth dataset comprises synthetic images generated from the real-world LIDC dataset using Diffusion and GAN models. The images are categorized into two types: those with real tumors removed and those with synthetic tumors artificially inserted by the model. Each synthetic image is paired with its corresponding original image, complete with precise annotations of the tumor insertion or removal locations. Considering that most users are not medically trained, we deliberately selected more evident abnormal images to reduce the need for specialized knowledge in making decisions about synthetic data.

# C EVALUATION

## C.1 EVALUATION MODEL

We compared various models on the LOKI benchmark to understand their capabilities across multiple tasks. We support over ten open-source models, including InternVL2 (Chen et al., 2024), LLaVA (Liu et al., 2023b), Phi (Gunasekar et al., 2023), XComposer (Zhang et al., 2023b), Qwen2-VL (Wang et al., 2024), MiniCPM (Hu et al., 2024), and Idefics2 (Laurençon et al., 2024), as well as proprietary models such as GPT-4 (OpenAI, 2024), Gemini (Team et al., 2023), Qwen-VL-Max (Bai et al., 2023), and Claude (Anthropic, 2024). The following list details these models.

| Model Family | Model Version | Parameters | Links |
|---|---|---|---|
| **Close-sourced, API** | | | |
| GPT4 | GPT-4o | N/A | https://platform.openai.com/docs/models/gpt-4o |
| | GPT-4 | N/A | https://platform.openai.com/docs/models/gpt-4-turbo-and-gpt-4 |
| Gemini | Gemini-1.5-Pro | N/A | https://ai.google.dev/gemini-api/docs/models/gemini#gemini-1.5-pro |
| | Gemini-1.5-Flash | N/A | https://ai.google.dev/gemini-api/docs/models/gemini#gemini-1.5-flash |
| Claude | Claude-3.5-Sonnet | N/A | https://docs.anthropic.com/en/docs/about-claude/models |
| Mistral | Mistral-Large | N/A | https://docs.mistral.ai/getting-started/models/ |
| Qwen | Qwen-Max | N/A | https://www.alibabacloud.com/help/en/model-studio/developer-reference/use-qwen-by-calling-api |
| **Open-sourced** | | | |
| LLaMA | LLaMA-3.1-405B | 405B | https://huggingface.co/meta-llama/Llama-3.1-405B |
| InternVL2 | InternVL2-8B | 8B | https://huggingface.co/OpenGVLab/InternVL2-8B |
| | InternVL2-26B | 26B | https://huggingface.co/OpenGVLab/InternVL2-26B |
| | InternVL2-40B | 40B | https://huggingface.co/OpenGVLab/InternVL2-40B |
| | InternVL2-Llama3-76B | 76B | https://huggingface.co/OpenGVLab/InternVL2-Llama3-76B |
| LLaVA-OneVision | LLaVA-OneVision-7B | 7B | https://huggingface.co/lmms-lab/llava-onevision-qwen2-7b-ov |
| | LLaVA-OneVision-72B | 72B | https://huggingface.co/lmms-lab/llava-onevision-qwen2-72b-ov-sft |
| VILA | VILA-1.5-13B | 13B | https://huggingface.co/Efficient-Large-Model/VILA1.5-13b |
| | VILA-1.5-40B | 40B | https://huggingface.co/Efficient-Large-Model/VILA1.5-40b |
| Phi | Phi-3.5-Vision | 3.5B | https://huggingface.co/microsoft/Phi-3.5-vision-instruct |
| Idefics2 | idefics2-8b | 8B | https://huggingface.co/HuggingFaceM4/idefics2-8b |
| Qwen2-VL | Qwen2-VL-7B | 7B | https://huggingface.co/Qwen/Qwen2-VL-7B-Instruct |
| | Qwen2-VL-72B | 72B | https://huggingface.co/Qwen/Qwen2-VL-72B-Instruct |
| InternLM-XComposer | InternLM-XComposer-2d5 | 7B | https://huggingface.co/internlm/internlm-xcomposer2d5-7b |
| mPLUG-Owl3 | mplug-owl3 | 7B | https://huggingface.co/mPLUG/mPLUG-Owl3-7B-240728 |
| MiniCPM | MiniCPM-V2.6 | 8.1B | https://huggingface.co/openbmb/MiniCPM-V-2_6 |
| LongVILA | LongVILA | 8B | https://huggingface.co/Efficient-Large-Model/Llama-3-LongVILA-8B-128Frames |
| LongVA | LongVA-7B | 7B | https://huggingface.co/lmms-lab/LongVA-7B-DPO |
| Qwen-Audio | Qwen-Audio-Chat | 7B | https://huggingface.co/Qwen/Qwen-Audio-Chat |
| SALMONN | SALMONN-7B | 7B | https://huggingface.co/tsinghua-ee/SALMONN-7B |
| AnyGPT | AnyGPT-Chat | 7B | https://huggingface.co/fnlp/AnyGPT-chat |
| OneLLM | OneLLM-7B | 7B | https://huggingface.co/csuhan/OneLLM-7B |
| LTU | LTU-AS-7B | 7B | https://github.com/YuanGongND/ltu#pretrained-models |

## C.2 EVALUATION METRIC

**Average accuracy:** For judgment, multiple-choice, and detailed selection questions, we use the *average accuracy* as the primary metric. The accuracy rate is calculated using the following formula:

$$\text{Accuracy} = \frac{N_{\text{correct}}}{N_{\text{total}}} \times 100\%$$

In this context, $N_{\text{correct}}$ is the number of correctly answered questions, and $N_{\text{total}}$ is the total number of questions. To minimize the influence of prompts on model judgments, each question is presented in two forms: one asks whether the data is AI-synthesized or real, and the other asks the model to identify either the real or AI-synthesized data. By averaging the accuracy rates across different forms of the questions, we aim to reduce the potential bias introduced by the phrasing of prompts and ensure a fair evaluation of the model's performance.

**Normalized Bias Index (NBI):** To evaluate whether there is potential bias in existing models when determining authenticity on the LOKI benchmark, we introduce a metric termed the Normalized Bias Index (NBI) to quantify the performance differences of the model on natural and AI-generated data across different modalities, which is defined as follows:

$$\text{NBI} = \frac{R_{\text{natural}} - R_{\text{generated}}}{R_{\text{natural}} + R_{\text{generated}}} \in [-1, 1]$$

In this context, $R_{\text{natural}}$ and $R_{\text{generated}}$ represent the recall rates for natural and AI-generated samples, respectively, under the corresponding modality. By normalizing the difference between the two, the model's unexpected preference in making predictions can be quantified. Specifically, a positive and larger NBI indicates that the model is more biased toward predicting samples as natural, whereas a negative and smaller NBI suggests a bias toward predicting samples as AI-generated.

**GPT-Score:** For open-ended questions regarding anomalous details, we use the GPT-4 model to assess the score of the responses. We adopted a rating scheme, establishing a 5-level rating system with scores ranging from 1 (poor) to 5 (excellent). The final scores are normalized to a scale of 0 to 100. We adhere to the following scoring criteria:

1) Identification: Accurately detect the globally annotated anomalies and their corresponding detailed anomalous regions specified by human annotators.

2) Explanation: Provide accurate explanations for the causes of the anomalies, ensuring consistency with the reasons outlined in the human annotations.

3) Plausibility: Avoid misclassifying authentic regions as anomalous while encouraging other reasonable explanations for anomalies.

While the scoring criteria are similar across different modalities, they are slightly adjusted according to their content characteristics; for example, the image modality is subdivided into global score and regional score, whereas 3D data is subdivided into texture score and normal score.

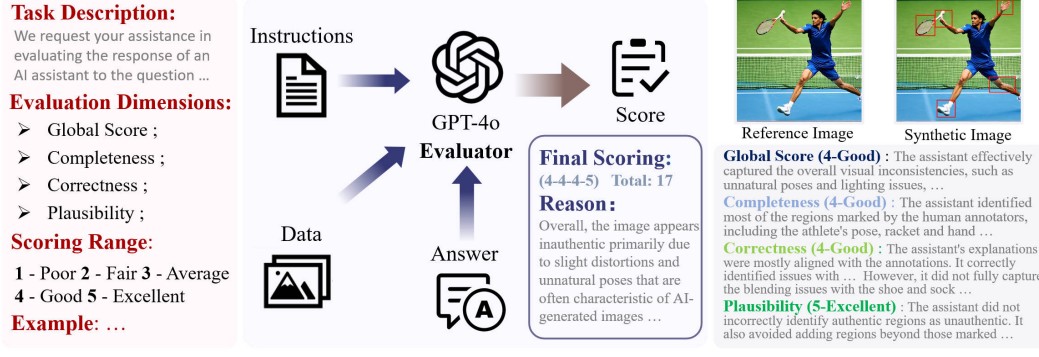

Figure 12: The overall process for automated evaluation of Abnormal explanation questions using GPT-4o.

## D   BREAKDOWN RESULTS ON DIFFERENT MODALITIES

In this appendix, we present detailed results of various synthesis methods or classification themes across different modalities of data.

### D.1   VIDEO

Tables 7 and 8 display the evaluation results of LMMs in video modality for judgment and multiple-choice questions, respectively. In the evaluation of video modality, the proprietary model GPT-4o achieved an accuracy of 71.3% for judgment questions and 77.3% for multiple-choice questions. The open-source model InternVL2-40B recorded accuracies of 62.0% for judgment questions and 65.7% for multiple-choice questions. Table 9 presents the evaluation results of different models in the video modality for open-ended questions.

Additionally, using GPT-4's detection outcomes as a benchmark, we assessed the quality of videos synthesized by different models. It was found that videos generated by Sora had a judgment accuracy of 67.2% and a multiple-choice accuracy of 64.5%, indicating a high quality of video synthesis. Conversely, videos synthesized by Runway and W.A.L.T exhibited more noticeable synthetic traces.

### D.2   IMAGE

Tables 10 and 11 present the evaluation results of LMMs in the image modality for judgment and multiple-choice questions, broken down by specific categories. The proprietary model GPT-4o scored a judgment accuracy of 63.4%, primarily impacted by specialized image types such as Satellite, Doc, and Medicine, while achieving a multiple-choice accuracy of 80.8%. Among open-source models, Qwen2-VL-72B performed exceptionally well, even surpassing GPT-4o in specialized image categories. Table 12 presents the evaluation results of different models in the Image modality for open-ended questions.

### D.3   3D

Table 13 presents the evaluation results for different 3D synthesis methods. Tests on judgment questions for 3D models reveal that most open-source models perform close to a 50% random chance. In contrast, the proprietary model GPT-4o demonstrates better decision-making and judgment capabilities, particularly in identifying low-quality Nerf models, where it achieves an accuracy rate exceeding 75Assuming the performance of GPT-4o as a benchmark, we can assess the quality of 3D data synthesized using different methods. Among three synthesis methods, Gaussian synthesis exhibits higher quality compared to Nerf synthesis, while commercial models like Clay offer the best quality. Table 14 presents the evaluation results of different models in the Image modality for open-ended questions.

As shown in Table 15, we extracted 200 3D objects, with an equal ratio of real to synthetic data, and evaluated the performance of different 3D input formats, including point clouds, surround videos with eight input frames, and multi-view inputs. In comparison, for current LMMs, the performance between surround video and multi-view inputs is similar. PointLLM, which partially supports point cloud inputs, performs particularly well, indicating that the point cloud format still holds certain advantages. Additionally, our results indicate that compared to the 7B PointLLM, the 13B PointLLM is more inclined to respond with "I cannot determine whether this is a real object or one generated by AI" rather than a straightforward "Yes" or "No." We speculate that as the size of PointLLM increases, it incorporates more comprehensive measures to correct illusions, making it less likely to provide definitive answers when faced with uncertainties (Wei et al., 2022a).

### D.4   AUDIO

Table 16 presents the evaluation results of various audio LMMs on the LOKI dataset. Currently, there are few open-source and proprietary LMMs that support the audio modality, and most models perform poorly on synthetic data detection tasks, showing little difference from random selection. However, the AASIST method, trained specifically for speech tasks, demonstrates relatively strong performance, although it still underperforms in other modalities. Overall, human performance in

the audio modality is comparatively weak relative to other modalities, yet it still exceeds the best-performing LMMs by 20%.

It has been shown that acoustic features, such as pauses between words and silent segments within audio, play a more critical role in detecting deepfake audio than the actual content of the audio(Liu et al., 2024b). However, current audio language models are primarily focused on content comprehension rather than acoustic characteristics. From the model architecture perspective, audio language models typically initialize their audio encoders with pre-trained audio models such as Whisper(Radford et al., 2023), AST(Gong et al., 2021), or BEATs(Chen et al., 2023b). These pre-trained models are trained with various data augmentation techniques, including SpecAugment(Park et al., 2019) and SpecSwap(Song et al., 2020b). The target of the augmented data remains constant, often focused on tasks like transcribing textual content, which shifts the model's attention towards understanding content rather than being sensitive to acoustic features. On the training data side, most audio-text datasets to date emphasize content comprehension in tasks like speech recognition, emotion/age recognition, audio/music captioning, sound event classification and music genre recognition. As a result, these models lack the inherent capability to distinguish between real and fake audio based on acoustic cues.

## D.5 TEXT

Tables 17 and 18 present the test results across different genres in the text modality. Most models underperform in this modality, primarily due to the low distinctiveness of text data itself, coupled with the maturity of current text synthesis technologies. Notably, genres such as modern literature and philosophy may benefit from the extensive training and memory capabilities of GPT series models, showing relatively better performance compared to other categories.

In Table 19, we evaluated the detection results for texts of varying lengths. The findings indicate that longer texts are detected more frequently and are more likely to reveal flaws. Additionally, we assessed the performance across different languages and found that, except for GPT-4o, most proprietary models perform significantly better in English than in Chinese. This indicates that the multilingual capabilities of large multimodal models require further development.

Table 7: **Judgement** questions results of different models on the LOKI **Video** modality. [*] denotes the closed-source models.

| | Overall | Sora | Keling | CogVideoX | Lumiere | Open-sora | Runway | W.A.L.T |
|---|---|---|---|---|---|---|---|---|
| Expert (AIGVDet) | 52.1 | 39.5 | 61.5 | 65.0 | 54.2 | 56.0 | 37.1 | 61.0 |
| MiniCPM-V-2.6 | 57.2 | 57.9 | 56.5 | 54.0 | 66.4 | 55.2 | 45.8 | 62.6 |
| Phi-3.5-Vision | 56.8 | 52.7 | 52.0 | 53.0 | 57.1 | 57.5 | 55.1 | 66.5 |
| LLaVA-OneVision-7B | 56.8 | 57.6 | 53.7 | 61.0 | 57.1 | 56.6 | 53.7 | 59.6 |
| InternLM-XComposer2.5 | 58.4 | 55.2 | 61.1 | 63.0 | 60.0 | 57.1 | 49.5 | 62.6 |
| mPLUG-Owl3-7B | 55.3 | 56.7 | 54.3 | 54.0 | 56.4 | 54.2 | 45.4 | 61.3 |
| Qwen2-VL-7B | 59.5 | 60.1 | 58.5 | 54.0 | 59.3 | 51.4 | 59.7 | 65.9 |
| LongVA-7B | 60.4 | 66.2 | 51.4 | 65.0 | 60.0 | 55.7 | 73.6 | 57.7 |
| Mantis-8B | 55.4 | 48.8 | 50.0 | 57.0 | 52.9 | 51.9 | 66.7 | 62.4 |
| Idefics2-8B | 54.8 | 56.1 | 54.0 | 51.0 | 57.9 | 51.4 | 41.2 | 64.3 |
| InternVL2-8B | 60.8 | 61.9 | 56.3 | 69.0 | 59.3 | 61.3 | 56.5 | 64.8 |
| Llama-3-LongVILA-8B | 51.9 | 54.0 | 49.1 | 53.0 | 52.1 | 48.1 | 52.8 | 54.1 |
| VILA1.5-13B | 51.9 | 51.2 | 50.9 | 51.0 | 54.3 | 52.8 | 52.3 | 51.9 |
| InternVL2-26B | 55.0 | 55.2 | 53.4 | 61.0 | 57.9 | 52.4 | 50.0 | 58.2 |
| VILA1.5-40B | 59.2 | 54.9 | 57.4 | 66.0 | 61.4 | 59.9 | 54.2 | 64.8 |
| InternVL2-40B | 62.0 | 61.3 | 57.7 | 68.0 | 60.7 | 61.8 | 62.0 | 65.9 |
| Qwen2-VL-72B | 59.6 | 58.2 | 61.4 | 59.0 | 58.6 | 54.7 | 58.8 | 63.2 |
| LLaVA-OneVision-72B | 56.5 | 57.0 | 55.4 | 62.0 | 55.7 | 51.9 | 46.8 | 64.6 |
| Gemini-1.5-Pro[*] | 58.5 | 53.4 | 60.5 | 58.0 | 63.6 | 62.3 | 45.4 | 64.8 |
| Claude-3.5-Sonnet[*] | 61.7 | 60.1 | 57.4 | 59.0 | 68.6 | 64.2 | 49.5 | 71.2 |
| GPT-4o[*] | 71.3 | 66.8 | 67.9 | 70.0 | 68.6 | 72.6 | 76.9 | 75.8 |

Table 8: **Multiple Choice** questions results of different models on the LOKI **Video** modality. [*] denotes the closed-source models.

| | Overall | Sora | Keling | CogVideoX | Lumiere | Open-sora | Runway | W.A.L.T |
|---|---|---|---|---|---|---|---|---|
| MiniCPM-V-2.6 | 52.8 | 39.5 | 62.1 | 55.6 | 58.8 | 54.0 | 48.0 | 53.7 |
| Phi-3.5-Vision | 58.2 | 57.9 | 42.4 | 50.0 | 70.6 | 58.0 | 68.0 | 58.5 |
| LLaVA-OneVision-7B | 59.8 | 55.3 | 62.1 | 63.9 | 66.2 | 56.0 | 64.0 | 54.9 |
| InternLM-XComposer2.5 | 56.3 | 43.4 | 54.5 | 66.7 | 67.6 | 62.0 | 54.0 | 53.7 |
| mPLUG-Owl3-7B | 60.3 | 53.9 | 53.0 | 61.1 | 66.2 | 62.0 | 62.0 | 64.6 |
| Qwen2-VL-7B | 64.0 | 63.2 | 62.1 | 52.8 | 69.1 | 66.0 | 70.0 | 62.2 |
| LongVA-7B | 57.5 | 59.2 | 50.0 | 58.3 | 63.2 | 52.0 | 58.0 | 59.8 |
| Mantis-8B | 47.9 | 56.6 | 43.9 | 50.0 | 52.9 | 52.0 | 42.0 | 39.0 |
| Idefics2-8B | 55.6 | 56.6 | 54.5 | 55.6 | 57.4 | 58.0 | 52.0 | 54.9 |
| InternVL2-8B | 54.0 | 53.9 | 57.6 | 55.6 | 61.8 | 48.0 | 52.0 | 48.8 |
| Llama-3-LongVILA-8B | 54.0 | 50.0 | 47.0 | 55.6 | 61.8 | 54.0 | 52.0 | 57.3 |
| VILA1.5-13B | 52.1 | 51.3 | 53.0 | 55.6 | 60.3 | 54.0 | 50.0 | 43.9 |
| InternVL2-26B | 62.4 | 61.8 | 57.6 | 66.7 | 63.2 | 56.0 | 68.0 | 64.6 |
| VILA1.5-40B | 49.1 | 50.0 | 48.5 | 47.2 | 54.4 | 50.0 | 42.0 | 48.8 |
| InternVL2-40B | 65.7 | 53.9 | 62.1 | 77.8 | 76.5 | 56.0 | 66.0 | 70.7 |
| Qwen2-VL-72B | 65.7 | 57.9 | 62.1 | 72.2 | 72.1 | 60.0 | 80.0 | 62.2 |
| LLaVA-OneVision-72B | 62.9 | 52.6 | 48.5 | 66.7 | 69.1 | 74.0 | 78.0 | 61.0 |
| Gemini-1.5-Pro[*] | 66.1 | 46.1 | 60.6 | 66.7 | 75.0 | 64.0 | 72.0 | 79.3 |
| Claude-3.5-Sonnet[*] | 60.5 | 53.9 | 54.5 | 58.3 | 72.1 | 72.0 | 58.0 | 57.3 |
| GPT-4o[*] | 77.3 | 61.8 | 71.2 | 75.0 | 94.1 | 78.0 | 82.0 | 80.5 |

Table 9: **Abnormal Explanation** questions results of different models on the LOKI **Video** modality. [*] denotes the closed-source models.

| Model | Overall | Correctness | Explanation | Plausibility |
|---|---|---|---|---|
| LLaVA-OV-7B | 46.7 | 41.0 | 48.4 | 50.8 |
| InternVL2-8B | 46.5 | 38.0 | 48.4 | 53.2 |
| Qwen2-VL-7B-Instruct | 48.4 | 41.2 | 50.4 | 53.6 |
| Claude-3-5-sonnet[*] | 50.1 | 44.2 | 48.8 | 57.4 |
| GPT-4o[*] | 67.6 | 60.6 | 69.2 | 73.0 |

Table 10: **Judgment** questions results of different models on the LOKI **Image** modality. [*] denotes the closed-source models.

| | Overall | Scene | Animal | Person | Object | Medicine | Doc | Satellite |
|---|---|---|---|---|---|---|---|---|
| Random Choice | 18.0 | 21.6 | 18.3 | 18.6 | 26.0 | 22.2 | 22.1 | |
| Human | 27.3 | 24.0 | 25.8 | 19.9 | 26.9 | 26.1 | 22.1 | |
| Expert (AIDE) | 63.1 | - | 89.9 | 62.5 | 96.5 | 53.4 | 49.7 | 39.3 |
| MiniCPM-V-2.6 | 44.8 | 52.0 | 34.4 | 53.1 | 31.5 | 53.8 | 51.5 | 38.3 |
| Phi-3.5-Vision | 52.5 | 50.8 | 41.7 | 71.5 | 34.1 | 57.3 | 54.3 | 60.5 |
| LLaVA-OneVision-7B | 49.8 | 59.2 | 41.9 | 58.1 | 37.3 | 52.3 | 53.0 | 50.1 |
| InternLM-XComposer2.5 | 46.4 | 52.7 | 40.0 | 56.7 | 32.5 | 56.1 | 49.8 | 38.2 |
| mPLUG-Owl3-7B | 45.9 | 52.1 | 37.3 | 52.9 | 31.4 | 55.3 | 53.8 | 38.1 |
| Qwen2-VL-7B | 47.8 | 54.7 | 38.9 | 57.9 | 30.3 | 56.0 | 59.6 | 36.9 |
| LongVA-7B | 46.2 | 57.6 | 37.4 | 52.5 | 34.1 | 54.4 | 49.8 | 39.7 |
| Mantis-8B | 54.6 | 54.9 | 52.2 | 54.8 | 53.5 | 53.1 | 51.9 | 63.3 |
| Idefics2-8B | 45.0 | 51.8 | 35.3 | 52.3 | 29.2 | 52.3 | 53.9 | 40.6 |
| InternVL2-8B | 49.7 | 58.8 | 39.4 | 54.4 | 37.8 | 53.9 | 60.2 | 44.2 |
| Llama-3-LongVILA-8B | 49.8 | 49.8 | 50.5 | 50.6 | 47.2 | 50.0 | 49.9 | 50.0 |
| VILA1.5-13B | 49.3 | 52.0 | 38.6 | 54.2 | 31.0 | 50.1 | 56.6 | 62.4 |
| InternVL2-26B | 44.3 | 51.6 | 35.4 | 50.8 | 28.2 | 51.3 | 54.4 | 37.6 |
| VILA1.5-40B | 48.8 | 53.7 | 39.3 | 50.0 | 33.4 | 52.5 | 59.9 | 50.6 |
| InternVL2-40B | 49.6 | 55.7 | 37.3 | 59.2 | 34.8 | 55.5 | 64.8 | 40.8 |
| Qwen2-VL-72B | 53.2 | 55.9 | 43.4 | 66.9 | 38.0 | 55.9 | 73.7 | 38.2 |
| LLaVA-OneVision-72b | 46.3 | 54.7 | 31.6 | 53.1 | 27.8 | 52.1 | 67.9 | 36.6 |
| Claude-3.5-Sonnet[*] | 53.6 | 51.6 | 51.6 | 55.2 | 51.4 | 51.9 | 59.1 | 50.9 |
| Gemini-1.5-Pro[*] | 43.5 | 53.7 | 35.7 | 51.5 | 30.3 | 50.0 | 47.2 | 38.1 |
| GPT-4o[*] | 63.4 | 70.1 | 69.7 | 84.4 | 70.3 | 54.3 | 60.1 | 45.0 |

Table 11: **Multiple Choice** questions results of different models on the LOKI **Image** modality. [*] denotes the closed-source models.

| | Overall | Scene | Animal | Person | Object | Medicine | Doc | Satellite |
|---|---|---|---|---|---|---|---|---|
| MiniCPM-V-2.6 | 49.8 | 49.2 | 52.3 | 49.2 | 48.5 | 50.5 | 45.1 | 51.6 |
| Phi-3.5-Vision | 44.0 | 31.3 | 22.9 | 67.9 | 20.7 | 69.8 | 58.3 | 52.1 |
| LLaVA-OneVision-7B | 51.7 | 55.9 | 51.8 | 63.3 | 46.9 | 52.8 | 51.4 | 46.3 |
| InternLM-XComposer2.5 | 51.0 | 48.4 | 49.5 | 46.3 | 50.5 | 57.0 | 52.0 | 51.2 |
| mPLUG-Owl3-7B | 52.5 | 53.1 | 53.5 | 53.8 | 49.5 | 55.5 | 49.4 | 52.3 |
| Qwen2-VL-7B | 65.1 | 60.9 | 71.3 | 70.4 | 60.5 | 68.0 | 63.1 | 59.6 |
| LongVA-7B | 51.6 | 58.2 | 54.9 | 42.9 | 48.2 | 55.3 | 46.6 | 51.6 |
| Mantis-8B | 61.5 | 66.0 | 58.1 | 57.1 | 56.9 | 62.3 | 68.3 | 63.6 |
| Idefics2-8B | 51.3 | 47.3 | 55.3 | 56.3 | 55.4 | 46.5 | 48.0 | 49.3 |
| InternVL2-8B | 51.4 | 56.6 | 51.2 | 54.2 | 48.5 | 48.8 | 50.6 | 52.6 |
| Llama-3-LongVILA-8B | 51.1 | 48.4 | 49.8 | 47.5 | 51.0 | 54.3 | 53.1 | 52.1 |
| VILA1.5-13B | 55.3 | 55.5 | 57.4 | 58.3 | 56.9 | 57.5 | 54.9 | 47.7 |
| InternVL2-26B | 48.5 | 50.4 | 49.1 | 50.8 | 40.3 | 50.8 | 53.1 | 46.7 |
| VILA1.5-40B | 64.0 | 59.8 | 71.0 | 58.8 | 53.8 | 71.5 | 62.9 | 63.8 |
| InternVL2-40B | 63.1 | 60.9 | 72.0 | 62.5 | 62.5 | 62.3 | 61.6 | 56.8 |
| Qwen2-VL-72B | 68.6 | 60.9 | 71.1 | 60.8 | 68.9 | 74.3 | 77.7 | 61.0 |
| LLaVA-OneVision-72b | 70.8 | 65.6 | 62.3 | 76.7 | 67.1 | 73.0 | 77.6 | 72.4 |
| Claude-3.5-Sonnet | 65.5 | 58.6 | 53.3 | 75.8 | 77.8 | 70.5 | 68.6 | 59.3 |
| Gemini-1.5-Pro[*] | 67.3 | 72.7 | 70.1 | 74.2 | 69.4 | 55.8 | 64.0 | 68.0 |
| GPT-4o[*] | 80.8 | 79.7 | 92.3 | 91.3 | 94.9 | 65.3 | 81.7 | 61.0 |

Table 12: **Abnormal Explanation** questions results of different models on the LOKI **Image** modality. [*] denotes the closed-source models.

|  | Overall | Global Score | Completeness | Correctness | Plausibility |
|---|---|---|---|---|---|
| Qwen2-VL-7B-Instruct | 63.8 | 66.6 | 60.8 | 53.2 | 74.4 |
| Llava-OV-7B | 68.9 | 72.4 | 63.0 | 60.6 | 79.8 |
| InternVL2-8B | 72.2 | 70.0 | 67.6 | 61.2 | 85.2 |
| Claude-3-5[*] | 1.7 | 2.0 | 1.6 | 1.4 | 1.8 |
| Gemini-1.5-pro[*] | 77.1 | 77.6 | 70.8 | 70.0 | 90.2 |
| GPT-4o[*] | 72.9 | 69.8 | 68.6 | 63.4 | 90.0 |

Table 13: **Judgment & Multiple Choice** questions results of different models on the LOKI **3D** modality. [*] denotes the closed-source models.

|  | Judgment | | | | Multiple Choice | | | |
|---|---|---|---|---|---|---|---|---|
|  | Nerf | Gaussian | Other | Overall | Nerf | Gaussian | Other | Overall |
| MiniCPM-V-2.6 | 56.7 | 56.1 | 56.7 | 56.4 | 50.6 | 50.0 | 52.9 | 50.7 |
| Phi-3.5-Vision | 50.0 | 50.0 | 50.0 | 50.0 | 62.0 | 56.1 | 62.9 | 59.6 |
| LLaVA-OneVision-7B | 69.3 | 67.4 | 69.0 | 68.4 | 52.5 | 54.6 | 54.6 | 53.8 |
| InternLM-XComposer2.5 | 42.8 | 44.6 | 44.4 | 43.9 | 41.2 | 51.0 | 55.4 | 48.0 |
| mPLUG-Owl3-7B | 49.1 | 50.5 | 49.8 | 49.9 | 46.6 | 51.7 | 52.5 | 49.9 |
| Qwen2-VL-7B | **72.1** | **73.2** | **70.4** | **72.3** | 53.2 | 57.6 | 55.0 | 55.5 |
| LongVA-7B | 49.6 | 50.0 | 50.0 | 49.9 | **63.2** | 59.9 | 61.3 | 61.4 |
| Mantis-8B | 50.0 | 50.0 | 50.0 | 50.0 | 51.3 | **73.0** | 60.4 | **62.5** |
| Idefics2-8B | 38.5 | 38.1 | 39.0 | 38.4 | 51.7 | 56.5 | 54.2 | 54.2 |
| InternVL2-8B | 48.8 | 49.8 | 49.6 | 49.4 | 52.7 | 54.3 | 51.3 | 53.1 |
| Llama-3-LongVILA-8B | 32.3 | 32.1 | 32.5 | 32.2 | 51.5 | 49.8 | 50.0 | 50.5 |
| VILA1.5-13B | 32.3 | 35.5 | 33.8 | 34.0 | 49.0 | 57.3 | 53.8 | 53.5 |
| InternVL2-26B | 50.3 | 50.4 | 50.4 | 50.4 | 44.8 | 51.2 | 51.7 | 48.8 |
| VILA1.5-40B | 50.0 | 50.0 | 50.0 | 50.0 | 44.1 | 50.2 | 50.4 | 47.9 |
| InternVL2-40B | 49.8 | 50.0 | 50.0 | 49.9 | 54.0 | 63.6 | 63.8 | 59.9 |
| Qwen2-VL-72B | 60.9 | 60.3 | 59.0 | 60.3 | 56.5 | 61.1 | 57.5 | 58.7 |
| LLaVA-OneVision-72B | 53.1 | 49.6 | 51.9 | 51.3 | 63.0 | 58.3 | **65.0** | 61.3 |
| Claude-3.5-Sonnet[*] | 59.5 | 57.1 | 57.1 | 58.0 | 52.3 | 52.0 | 50.8 | 51.9 |
| Gemini-1.5-Pro[*] | 60.5 | 50.7 | 56.0 | 55.4 | 64.3 | 56.0 | 61.7 | 60.2 |
| GPT-4o[*] | **65.9** | **64.4** | **65.4** | **65.2** | **78.8** | **66.4** | 60.8 | **70.2** |
| Human | 87.3 | 78.4 | 71.8 | 82.0 | 94.5 | 90.2 | 78.6 | 91.3 |

Table 14: **Abnormal Explanation** questions results of different models on the LOKI **3D** modality. [*] denotes the closed-source models.

|  | Overall | Texture Correctness | Normal Correctness | Texture Plausibility | Normal Plausibility |
|---|---|---|---|---|---|
| Llava-OV-7B | 71.0 | 63.2 | 70.8 | 73.8 | 76.2 |
| InternVL2-8B | 71.3 | 62.4 | 69.4 | 76.4 | 76.8 |
| Qwen2-VL-7B-Instruct | 73.4 | 64.8 | 72.2 | 77.8 | 78.6 |
| Claude-3-5-sonnet[*] | 78.2 | 70.0 | 80.0 | 78.8 | 83.8 |
| Gemini-1.5-pro[*] | 70.8 | 62.4 | 67.2 | 77.2 | 76.4 |
| GPT-4o[*] | 77.0 | 69.8 | 64.2 | 82.4 | 81.4 |

Table 15: **Judgment & Multiple Choice** questions results of different models on the LOKI **3D** modality. S-Video denotes Surround Video, MV-Image denotes Multi-Views Image.

|  | Judgment | | Multiple Choice | |  | Judgment |
|---|---|---|---|---|---|---|
|  | S-Video | MV-Image | S-Video | MV-Image |  | PointCloud |
| LLaVA-OneVision-7B | 67.4 | 69.0 | 52.5 | 54.6 | PointLLM-7B | 61.9 |
| Idefics2-8B | 38.1 | 39.0 | 51.7 | 56.5 | PointLLM-13B | 16.9 |
| LLaVA-OneVision-72B | 49.6 | 51.9 | 63.0 | 58.3 | OneLLM | 55.6 |

Table 16: **Overall** results of different models on the LOKI **Audio** modality.

| | Judgment | | | | | Multiple Choice | | | | |
|---|---|---|---|---|---|---|---|---|---|---|
| | Speech (800) | Singing (800) | Audio (280) | Music (400) | Overall (2280) | Speech (400) | Singing (400) | Audio (140) | Music (300) | Overall (1240) |
| AASIST Jung et al. (2022) | 96.4 | 54.4 | 57.1 | 54.3 | 69.4 | - | - | - | - | - |
| Qwen-Audio | 50.0 | 49.8 | 49.8 | 49.7 | 49.8 | 50.8 | 48.0 | 58.9 | 48.0 | 50.1 |
| SALMONN-7B | 52.8 | 52.4 | 49.3 | 46.8 | 51.2 | - | - | - | - | - |
| AnyGPT | 48.5 | 52.4 | 51.4 | 45.9 | 49.8 | 50.8 | 48.0 | 58.2 | 49.0 | 50.3 |
| OneLLM | 50.3 | 50.2 | 50.9 | 47.9 | 49.9 | - | - | - | - | - |
| LTU | 40.2 | 45.3 | 50.5 | 46.9 | 44.4 | - | - | - | - | - |
| Qwen-Audio2 | 49.4 | 40.0 | 49.6 | 27.5 | 42.3 | 49.3 | 49.3 | 48.6 | 51.3 | 49.7 |
| Gemini-1.5-Flash Team et al. (2023) | 50.1 | 48.3 | 47.3 | 49.8 | 49.4 | 48.6 | 49.5 | 48.6 | 49.7 | 49.2 |

Table 17: **Judgment** questions results of different models on the LOKI **text** modality. [*] denotes the closed-source models.

| | Scientific Papers | News | Essay | Wikipedia | Speech | Modern Literature | Phiosophy | Ancient Chinese |
|---|---|---|---|---|---|---|---|---|
| Expert (RADAR-Vicuna-7B) | 76.2 | 81.4 | 90.0 | 67.2 | 73.9 | 63.3 | 79.45 | 1.7 |
| MiniCPM-V-2.6 | 51.4 | 47.7 | 49.8 | 50.8 | 49.4 | 50.5 | 49.4 | 43.5 |
| Phi-3.5-Vision | 48.1 | 50.3 | 47.9 | 47.5 | 48.2 | 53.1 | 50.7 | 50.0 |
| LLaVA-OneVision-7B | 54.5 | 50.3 | 55.1 | 55.8 | 52.7 | 57.0 | 51.2 | 42.1 |
| InternLM-XComposer2.5 | 51.9 | 52.5 | 55.8 | 48.2 | 49.7 | 55.7 | 54.9 | 52.2 |
| mPLUG-Owl3-7B | 51.3 | 50.5 | 53.4 | 52.3 | 53.2 | 60.6 | 59.5 | 41.9 |
| Qwen2-VL-7B | 49.9 | 48.3 | 49.6 | 51.4 | 48.1 | 50.5 | 48.0 | 42.1 |
| LongVA-7B | 51.0 | 48.8 | 48.0 | 49.8 | 47.2 | 50.2 | 44.8 | 49.0 |
| Mantis-8B | 51.0 | 51.8 | 51.5 | 52.8 | 53.5 | 54.9 | 52.7 | 43.2 |
| Idefics2-8b | 49.2 | 46.4 | 45.5 | 45.4 | 44.0 | 52.4 | 49.4 | 43.1 |
| InternVL2-8B | 51.6 | 49.4 | 45.6 | 49.1 | 47.1 | 52.7 | 54.7 | 54.4 |
| Llama-3-LongVILA-8B | 49.4 | 50.6 | 49.1 | 48.9 | 51.2 | 50.0 | 49.8 | 50.0 |
| VILA1.5-13B | 50.5 | 48.0 | 44.0 | 47.2 | 42.6 | 55.1 | 45.8 | 48.5 |
| InternVL2-26B | 51.6 | 49.7 | 50.4 | 48.4 | 47.2 | 56.2 | 53.3 | 53.6 |
| VILA1.5-40B | 50.7 | 50.3 | 50.0 | 50.0 | 49.7 | 50.0 | 49.8 | 50.7 |
| Qwen2-VL-72B | 53.6 | 49.9 | 50.6 | 50.6 | 52.0 | 53.1 | 59.8 | 52.5 |
| LLaVA-OneVision-72B | 56.0 | 61.7 | 50.7 | 57.0 | 54.7 | 73.4 | 71.5 | 68.6 |
| Llama-3.1-405B | 54.5 | 53.4 | 49.2 | 57.8 | 56.7 | 57.7 | 67.2 | 58.8 |
| Qwen-max | 48.0 | 51.5 | 47.6 | 48.3 | 50.1 | 49.1 | 47.4 | 41.1 |
| Mistral Large[*] | 49.4 | 48.9 | 45.1 | 49.4 | 50.9 | 60.6 | 61.3 | 50.8 |
| Claude-3.5-Sonnet[*] | 53.2 | 56.0 | 60.2 | 58.9 | 66.1 | 68.3 | 68.2 | 60.1 |
| Gemini-1.5-Pro[*] | 55.0 | 52.2 | 49.9 | 52.9 | 56.5 | 62.5 | 59.0 | 58.6 |
| GPT-4o[*] | 50.1 | 54.0 | 48.4 | 52.1 | 55.3 | 64.2 | 64.7 | 61.5 |

Table 18: **Multiple Choice** questions results of different models on the LOKI text modality. * denotes the closed-source models.

| | Scientific Papers | News | Essay | Wikipedia | Speech | Modern Literature | Phiosophy | Ancient Chinese |
|---|---|---|---|---|---|---|---|---|
| MiniCPM-V-2.6 | 48.6 | 42.6 | 47.8 | 48.2 | 48.5 | 50.1 | 56.3 | 50.0 |
| Phi-3.5-Vision | 45.8 | 39.9 | 41.0 | 41.1 | 36.3 | 42.6 | 44.3 | 47.5 |
| LLaVA-OneVision-7B | 51.5 | 48.2 | 48.6 | 44.9 | 44.9 | 49.6 | 51.1 | 48.3 |
| InternLM-XComposer2.5 | 44.0 | 42.8 | 33.9 | 41.0 | 34.3 | 46.3 | 40.8 | 41.9 |
| mPLUG-Owl3-7B | 51.8 | 49.2 | 45.1 | 48.3 | 44.9 | 54.0 | 62.1 | 38.6 |
| Qwen2-VL-7B | 49.6 | 46.3 | 42.6 | 44.2 | 45.8 | 45.4 | 50.7 | 46.1 |
| LongVA-7B | 50.1 | 49.0 | 48.1 | 47.5 | 46.1 | 51.4 | 49.3 | 50.3 |
| Mantis-8B | 58.9 | 54.4 | 36.3 | 51.5 | 35.3 | 49.7 | 59.6 | 34.7 |
| Idefics2-8B | 50.1 | 40.6 | 31.5 | 40.1 | 35.8 | 35.0 | 33.6 | 20.8 |
| InternVL2-8B | 53.9 | 52.8 | 38.9 | 47.2 | 42.5 | 43.8 | 50.7 | 40.0 |
| Llama-3-LongVILA-8B | 46.4 | 43.5 | 40.1 | 42.2 | 37.4 | 51.4 | 46.7 | 49.2 |
| VILA1.5-13b | 50.3 | 40.8 | 39.9 | 44.6 | 40.8 | 43.5 | 48.1 | 44.2 |
| InternVL2-26B | 52.6 | 54.9 | 43.2 | 49.3 | 47.6 | 48.9 | 56.3 | 48.9 |
| VILA1.5-40B | 49.9 | 52.5 | 39.4 | 48.9 | 46.3 | 57.6 | 58.2 | 50.8 |
| Qwen2-VL-72B | 67.2 | 79.2 | 53.1 | 72.9 | 62.5 | 76.7 | 76.1 | 69.7 |
| LLaVA-OneVision-72B | 65.1 | 79.9 | 50.6 | 63.1 | 61.3 | 82.6 | 78.3 | 76.9 |
| Llama-3.1-405B | 56.8 | 61.7 | 74.7 | 54.4 | 69.6 | 74.2 | 86.5 | 83.1 |
| Qwen-max | 48.3 | 48.3 | 47.4 | 33.8 | 43.3 | 43.8 | 48.3 | 48.1 |
| Mistral Large* | 55.7 | 75.1 | 55.4 | 62.8 | 61.8 | 86.7 | 86.1 | 69.4 |
| Claude-3.5-Sonnet* | 83.9 | 88.3 | 76.4 | 86.9 | 92.8 | 96.9 | 97.5 | 92.2 |
| Gemini-1.5-Pro* | 51.3 | 56.7 | 33.5 | 46.8 | 60.7 | 70.4 | 71.5 | 77.2 |
| GPT-4o* | 62.2 | 62.9 | 49.4 | 59.0 | 66.9 | 81.5 | 79.3 | 75.6 |

Table 19: Performance Comparison of Models across Text Lengths and Languages. * denotes the closed-source models.

| | Short text | Medium text | Long text | English | Chinese |
|---|---|---|---|---|---|
| MiniCPM-V-2.6 | 49.1 | 48.9 | 46.6 | 49.7 | 48.9 |
| Phi-3.5-Vision | 63.2 | 59.2 | 56.5 | 48.6 | 45.5 |
| LLaVA-OneVision-7B | 45.5 | 45.7 | 46.8 | 53.1 | 50.1 |
| InternLM-XComposer2.5 | 46.4 | 46.7 | 47.2 | 48.2 | 49.0 |
| mPLUG-Owl3-7B | 48.8 | 47.6 | 49.0 | 56.3 | 48.9 |
| Qwen2-VL-7B | 43.8 | 42.4 | 43.5 | 47.6 | 48.4 |
| LongVA-7B | 37.7 | 37.0 | 37.5 | 49.5 | 48.0 |
| Mantis-8B | 54.7 | 55.5 | 55.6 | 53.7 | 48.2 |
| Idefics2-8B | 46.9 | 41.8 | 33.9 | 46.7 | 41.2 |
| InternVL2-8B | 45.6 | 44.4 | 42.1 | 49.7 | 48.6 |
| Llama-3-LongVILA-8B | 47.1 | 46.8 | 47.0 | 47.7 | 48.3 |
| VILA1.5-13B | 49.4 | 48.0 | 44.4 | 47.0 | 46.0 |
| InternVL2-26B | 34.9 | 33.9 | 35.4 | 51.0 | 50.7 |
| VILA1.5-40B | 50.5 | 50.9 | 50.7 | 49.8 | 50.5 |
| Qwen2-VL-72B | 40.4 | 41.6 | 47.4 | 60.2 | 56.9 |
| LLaVA-OneVision-72B | 53.5 | 56.6 | 63.1 | 63.8 | 64.0 |
| Qwen-max | 35.8 | 34.0 | 32.8 | 48.3 | 45.9 |
| Llama-3.1-405B | 49.0 | 49.6 | 56.5 | 63.4 | 61.3 |
| Mistral-Large* | 44.6 | 46.3 | 51.3 | 60.5 | 55.5 |
| Claude-3.5-Sonnet* | 56.1 | 60.3 | 67.6 | 73.5 | 68.3 |
| Gemini-1.5-Pro* | 39.7 | 40.3 | 47.0 | 58.4 | 54.3 |
| GPT-4o* | 46.9 | 49.4 | 58.2 | 60.0 | 59.0 |

# E   MORE EXPERIMENTAL RESULTS AND DISCUSSIONS

In this appendix, we provide a detailed discussion of additional experimental results, including compression artifacts tests, analyses of Deepfake category experiments, and further results from CoT experiments.

## E.1   COMPRESSION ARTIFACT TESTS

To better evaluate the applicability of large multimodal models (LMMs) in real-world scenarios, we conducted additional compression artifact tests. Following previous studies(Wang et al., 2020a; Ojha et al., 2023), we conducted JPEG compression tests on judgment tasks under the image modality, based on three representative LMMs and an expert model AIDE, as previously introduced in the main text.

Table 20: **Compression artifact test**. The test results of different models under various compression ratios.

|  | 100% | 95% | 90% | 80% | 60% |
|---|---|---|---|---|---|
| InternVL2-8B | 56.6% | 56.1% | 56.1% | 56.4% | 56.6% |
| Qwen2-VL-7B | 54.5% | 54.0% | 54.2% | 54.6% | 54.1% |
| GPT-4o | 63.4% | 63.2% | 63.1% | 62.8% | 63.1% |
| AIDE | 62.8% | 46.4% | 45.0% | 44.2% | 45.2% |

From these results, we observe that expert models AIDE(Yan et al., 2024) are more significantly impacted by compression artifacts. In contrast, LMMs demonstrate remarkable robustness, with compression levels having little to no effect on their detection capabilities. This distinction highlights the inherent advantages of LMMs over traditional expert models. There are several possible factors that contribute to the robustness of LMMs to compressed images.

Extensive Pretraining on Diverse Data: The vast amount of data available on the internet includes images with varying levels of compression. During pretraining, LMMs inherently learn features that are robust to such artifacts by including the internet-scale data, as demonstrated in works like UniversalFakeDetect(Ojha et al., 2023) and CLIPping(Khan & Dang-Nguyen, 2024). Model Scale Advantage: Unlike traditional expert modules, which typically operate with parameter counts in the millions, LMMs are equipped with parameters in the billions. This scale significantly enhances their generalization and robustness, consistent with the emergent properties observed in large-scale models. This finding further underscores the potential of LMMs for real-world applications, particularly in scenarios involving compression artifacts. We will include the above experimental results and analysis of compression robustness in the supplementary material to provide a comprehensive view of LMM reliability and applicability. If you have additional questions, we welcome further discussion.

## E.2   DEEPFAKE DETECTION

Deepfake detection is one of the most prominent synthetic data detection categories today. LOKI already includes partial Deepfake datasets from audio and image modalities. The existing data for the Deepfake category includes:

**Audio Modality:** In audio tasks, some samples of speech and singing are sourced from their respective audio deepfake datasets ASVSpoof2019 and DCASE2023 Track 7, as mentioned in Table 6. These datasets are built on state-of-the-art voice cloning techniques, such as Text-to-Speech(Ren et al., 2019), Voice Conversion(Sisman et al., 2020), and Singing Voice Conversion(Nachmani & Wolf, 2019). These technologies address major challenges in deepfake detection, including vulnerabilities in voice authentication systems and the misuse of synthetic voice technology to generate counterfeit content, such as fake artists.

**Image Modality:** As shown in Table 6, our image data includes a portion of facial deepfake images, comprising partial datasets from FFHQ and Deepfakeface. These data are directly aligned with the

core challenges in image-based deepfake detection, ensuring that our benchmark comprehensively evaluates models on this critical aspect of synthetic data detection.

Table 21: The accuracy of different models on the deepfake dataset and their overall accuracy.

| Model | Claude-3.5-Sonnet | InternVL2-8B | VILA1.5-40B | Llava-OV-7B | Mantis-8B | Qwen2-VL-72B | Llama-3-8B | GPT-4o |
|---|---|---|---|---|---|---|---|---|
| Deepfake Accuracy | 0.14 | 0.13 | 0.29 | 0.13 | 0.53 | 0.34 | 0.47 | 0.59 |
| Overall Accuracy | 0.54 | 0.50 | 0.49 | 0.50 | 0.55 | 0.53 | 0.50 | 0.63 |

We have independently evaluated the deepfake-related components of our current dataset and presented the results below. From the results, it is evident that most models perform significantly worse on deepfake data compared to their overall accuracy. This discrepancy likely arises because the overall evaluation task includes simpler synthetic samples (e.g., basic synthetic data), which are easier for the models to distinguish, thereby inflating their overall accuracy. In contrast, deepfake data often involves more nuanced and fine-grained forgery features that are particularly challenging for general-purpose multimodal pretrained models to capture effectively. Interestingly, GPT-4o achieves the best performance on deepfake data among all evaluated models, demonstrating its relatively stronger ability to handle such complex forgeries.

### E.3 MORE CoT EXPERIMENTS RESULTS

To further validate the effectiveness of CoT across various question types and modalities, we conducted additional experiments. These experiments included testing multiple question types in the image modality (e.g., multiple-choice and anomaly detection) and extending the evaluation to video and text modalities (e.g., binary classification tasks). The results are shown in Table 22.

From the results, we observe that for most models, CoT significantly improves performance across different question types and modalities. Models such as InternVL2-8B, Qwen2-VL and GPT-4o consistently benefit from CoT prompting, with noticeable performance gains across both tasks (e.g., anomaly detection, binary classification) and modalities (e.g., image, video, text), which is consistent with findings in other CoT prompting methods(Wei et al., 2022b; Kojima et al., 2022). However, not all models benefit from CoT. For example, LLaVA-OV-7B shows a significant performance drop $(56.6 \rightarrow 18.8)$. Upon closer examination, we found that LLaVA-OV-7B struggles with long-context reasoning, as evidenced by irregular outputs such as incomplete sentences, irrelevant responses, or prematurely ending sentences with eos token. This suggests limitations in its training for handling extended contexts, which may hinder its ability to utilize CoT effectively.

In conjunction with the results in Table 4, we observe that the negative impact of FS (Few-shot) prompting on LMMs may be attributed to the inherent difficulty of synthetic detection tasks, which requires fine-grained perception and complex reasoning to recognize artifacts, which might be out of the training scope of current LMMs. Therefore, in FS prompting, simply providing in-context examples without reasoning steps might be insufficient for most models to learn how to detect artifacts. Furthermore, adding them to their context window might interfere with their original reasoning paths, resulting in degraded performances. Therefore, in the difficult setup of synthetic detection, the effects of FS prompting on model performance are limited. We note that the strong performance of GPT-4o models after FS prompting might stem from its inherently strong reasoning abilities.

Table 22: More Chain of Thought prompting experiments results.

| Model | Image Multi-choice | | Image Abnormal Selection | | Video Judgement | | Text Judgement | |
|---|---|---|---|---|---|---|---|---|
| | w/o CoT | w/ CoT | w/o CoT | w/ CoT | w/o CoT | w/ CoT | w/o CoT | w/ CoT |
| LLaVA-OV-7B | 51.7 | 33.7 | 18.8 | 14.4 | 65.0 | 50.5 | 51.6 | 43.1 |
| InternVL2-8B | 51.4 | 53.2 | 70.2 | 75.3 | 51.2 | 55.1 | 41.6 | 52.3 |
| Qwen2-VL | 65.1 | 67.6 | 31.5 | 36.6 | 62.3 | 69.4 | 45.3 | 54.3 |
| GPT-4o | 80.8 | 88.9 | 76.2 | 84.3 | 71.2 | 77.8 | 53.5 | 59.6 |

E.4    PERFORMANCE ACROSS DIFFERENT LEVELS AND MODALITIES

Table 23 presents the performance of different models across various modalities and difficulty levels. Based on human user performance, we categorized the difficulty levels of the questions. As the difficulty of the questions increases, most models show a gradual decline in performance. Notably, GPT-4o maintains relatively strong performance in both the Video and Image modalities. Given the diverse tasks and modalities involved in LOKI, considering the scores across different question difficulties is beneficial for assessing the overall performance of the models.

Table 23: The performance of different models across various modalities and task difficulty levels.

| Model | Video | | | Image | | | Text | | | 3D | | |
|---|---|---|---|---|---|---|---|---|---|---|---|---|
| | Easy | Medium | Hard | Easy | Medium | Hard | Easy | Medium | Hard | Easy | Medium | Hard |
| LLaVA-OV-7B | 0.58 | 0.55 | 0.52 | 0.55 | 0.59 | 0.56 | 0.62 | 0.46 | 0.41 | 0.61 | 0.58 | 0.55 |
| InternVL2-8B | 0.64 | 0.56 | 0.49 | 0.54 | 0.58 | 0.53 | 0.56 | 0.50 | 0.39 | 0.42 | 0.32 | 0.25 |
| Qwen2-VL-7B | 0.63 | 0.52 | 0.52 | 0.55 | 0.55 | 0.54 | 0.62 | 0.46 | 0.37 | 0.67 | 0.55 | 0.45 |
| Gemini-1.5-pro* | 0.63 | 0.55 | 0.48 | 0.52 | 0.51 | 0.51 | 0.64 | 0.14 | 0.17 | 0.60 | 0.64 | 0.68 |
| Claude-3.5-Sonnet* | 0.65 | 0.61 | 0.54 | 0.54 | 0.53 | 0.51 | 0.72 | 0.15 | 0.27 | 0.55 | 0.53 | 0.58 |
| GPT-4o* | 0.74 | 0.65 | 0.63 | 0.70 | 0.69 | 0.66 | 0.64 | 0.18 | 0.29 | 0.58 | 0.51 | 0.58 |

# F  CASE STUDY

In this section, we delve into an analysis of GPT-4o's performance on synthetic data detection tasks, with a particular focus on the questions it answered incorrectly. This examination is crucial for understanding the model's operational capabilities and limitations. The analysis not only identifies the model's current shortcomings but also provides guidance for future design and training improvements. We randomly selected 50 erroneous instances from GPT-4o's predictions across various modalities and difficulty levels for detailed inspection. These instances included multiple-choice, true/false, and open-ended responses, and even for the structured questions, GPT-4o was required to provide the reasoning behind its decisions. These instances were analyzed by specialized annotators who identified the fundamental causes of the erroneous predictions based on their expertise and annotations.

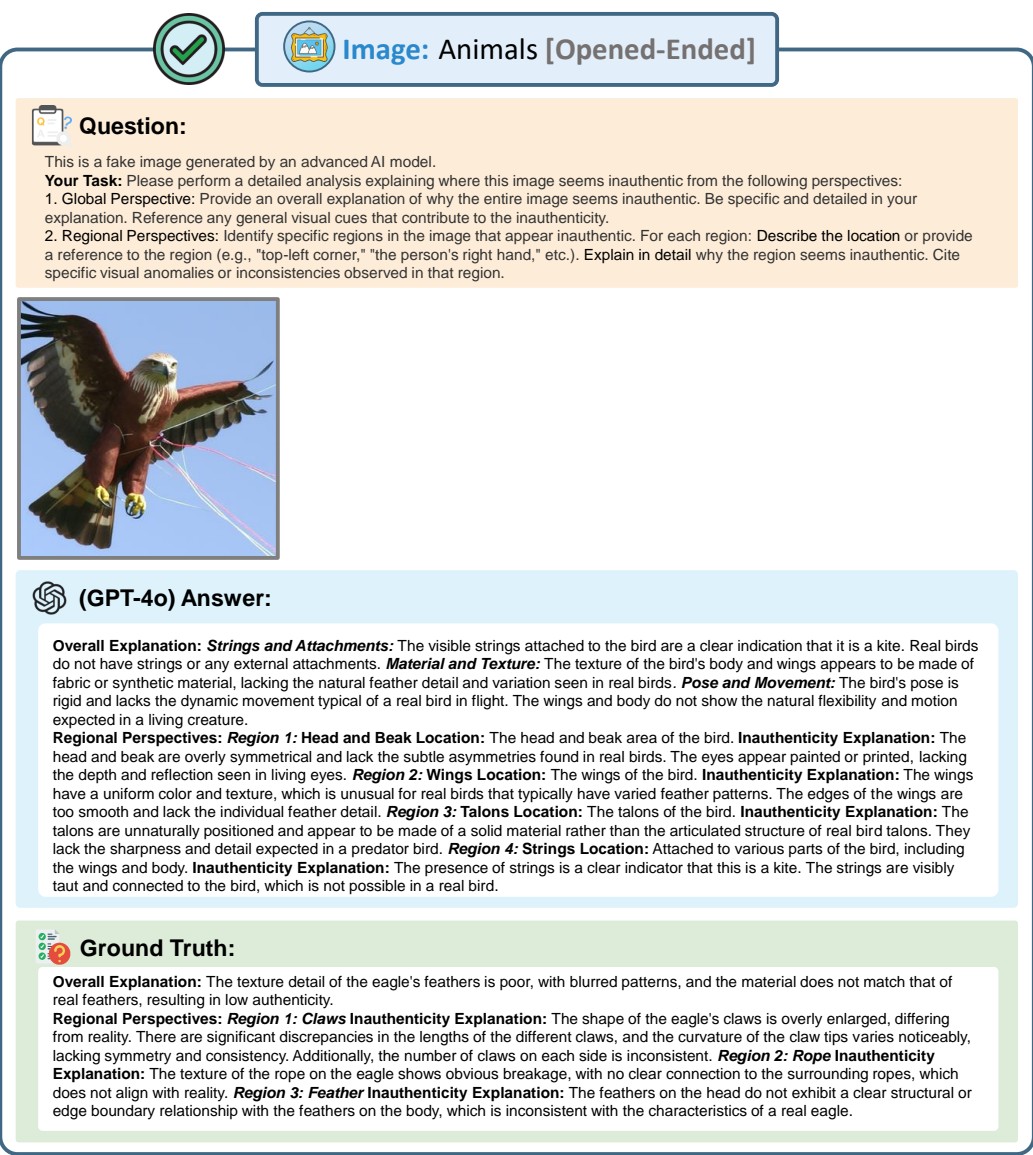

Figure 13: Abnormal detail explanation example of category **animals** in image modality, with GPT-4o's answers

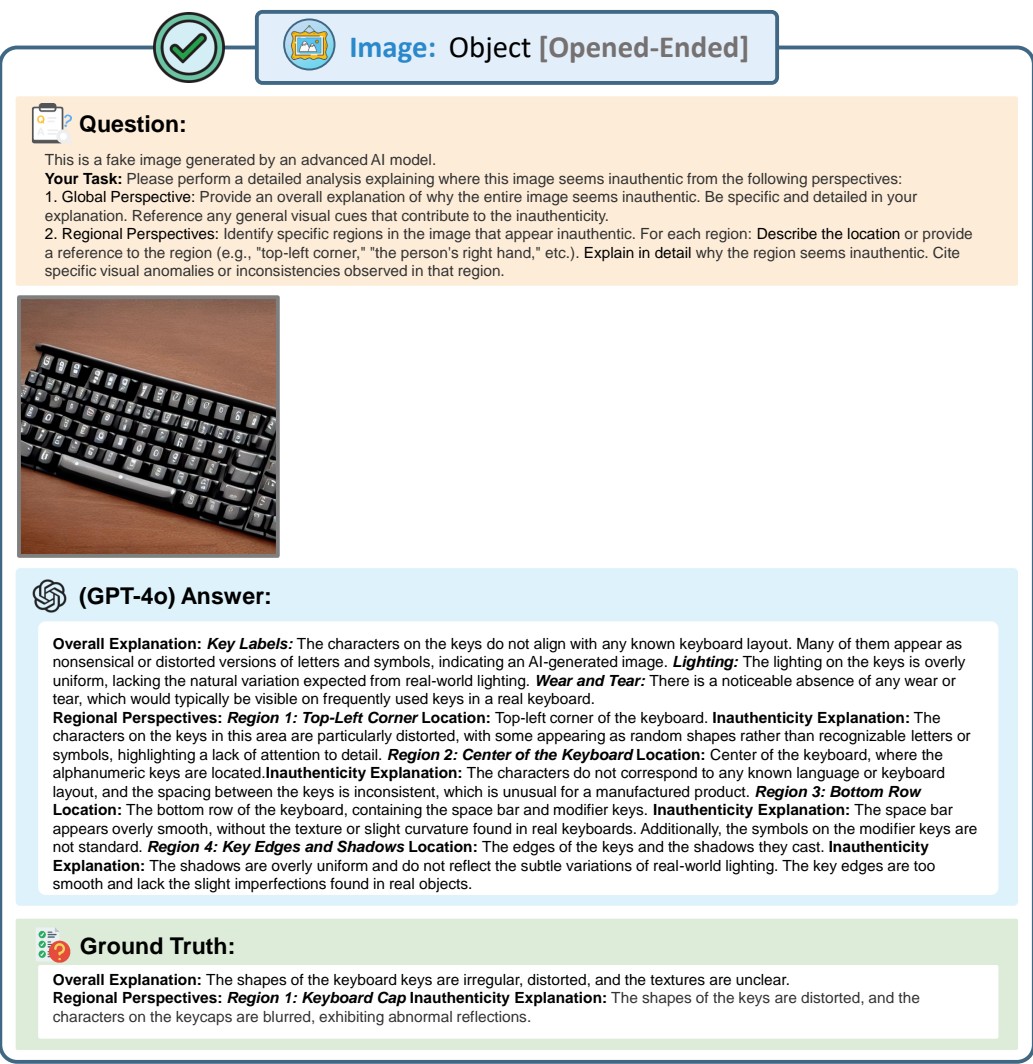

Figure 14: Abnormal detail explanation example of category **object** in image modality, with GPT-4o's answers

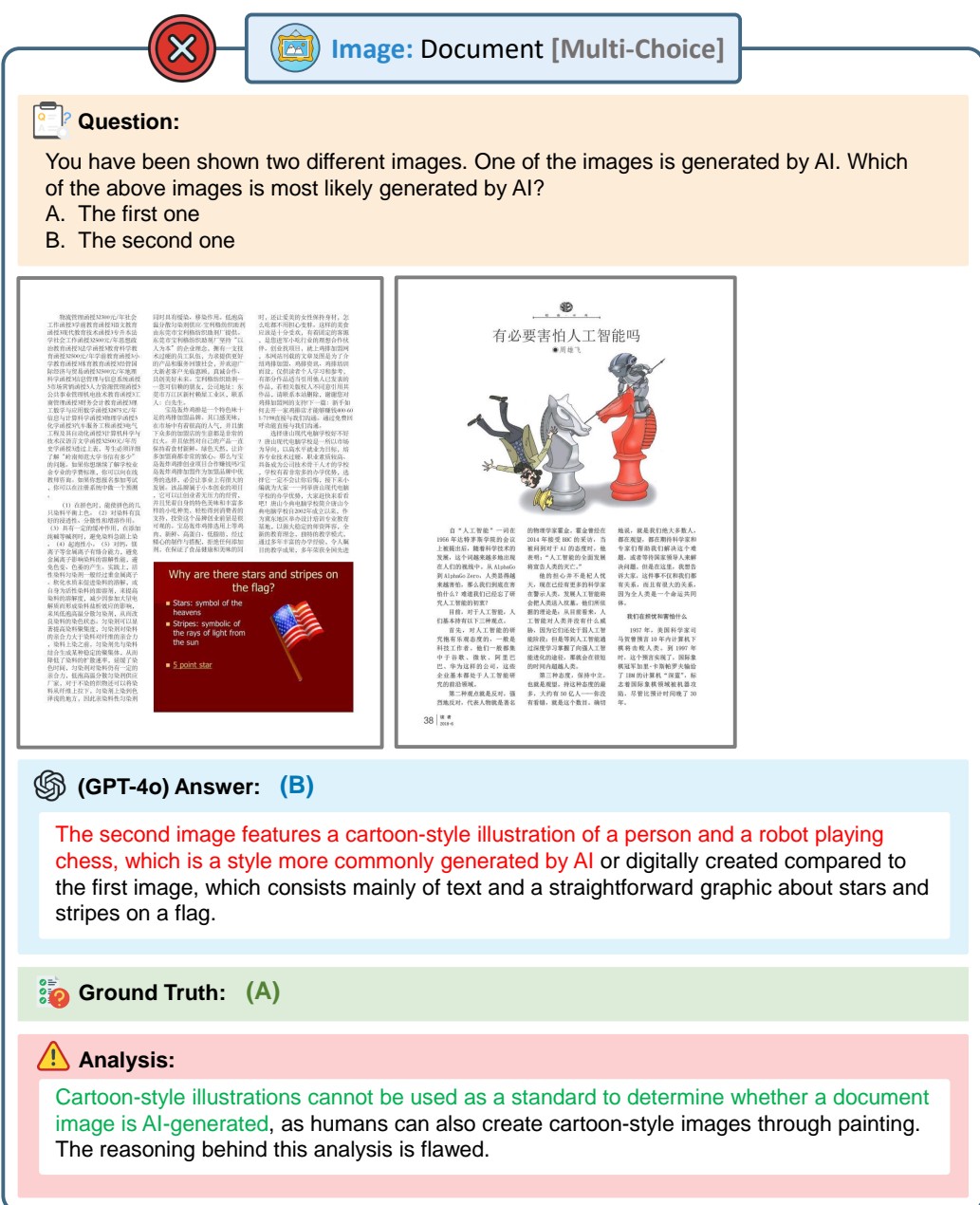

Figure 15: Multi-choice example of category **document** in image modality, with GPT-4o's answers

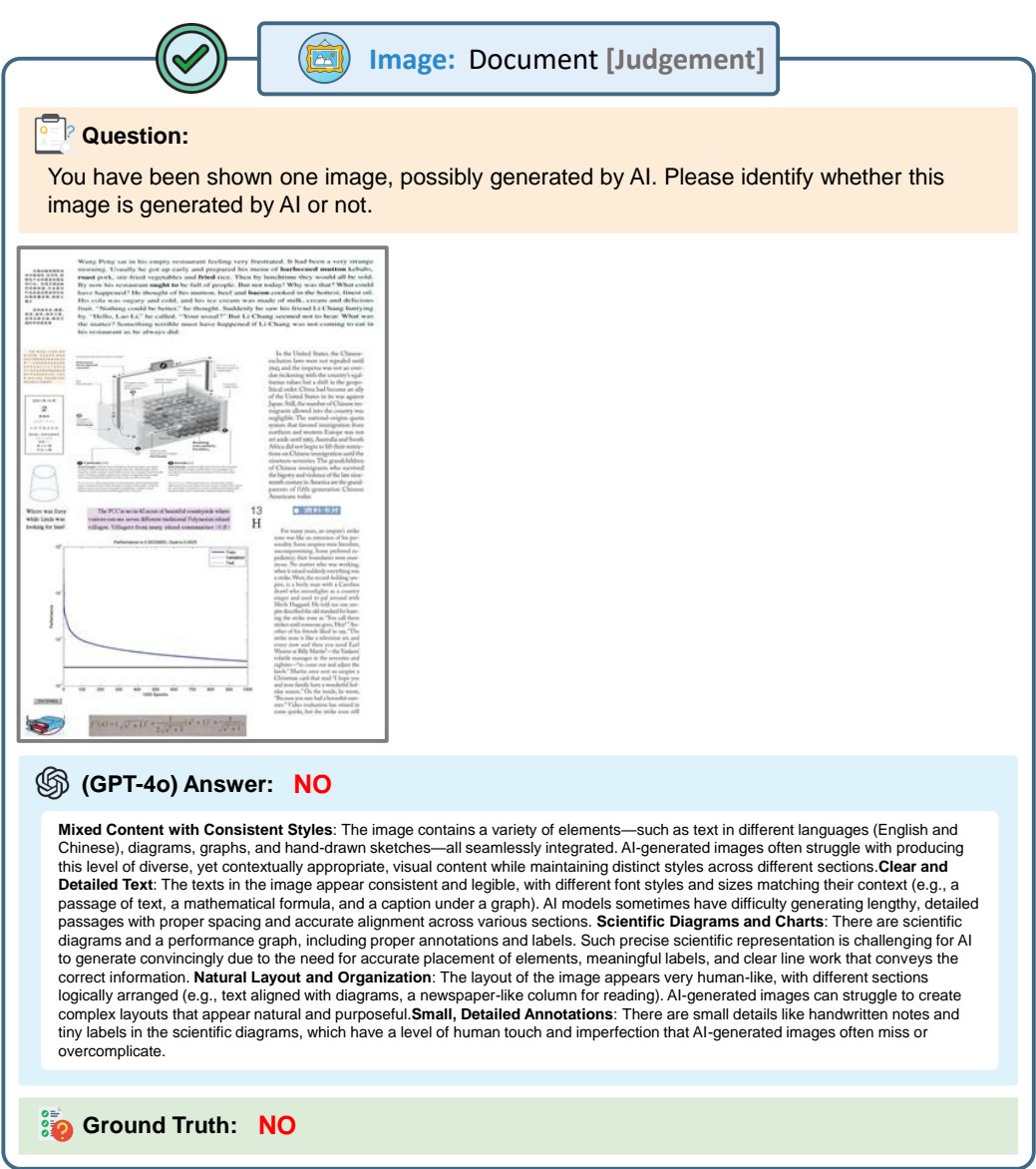

Figure 16: Judgement example of category **document** in image modality, with GPT-4o's answers

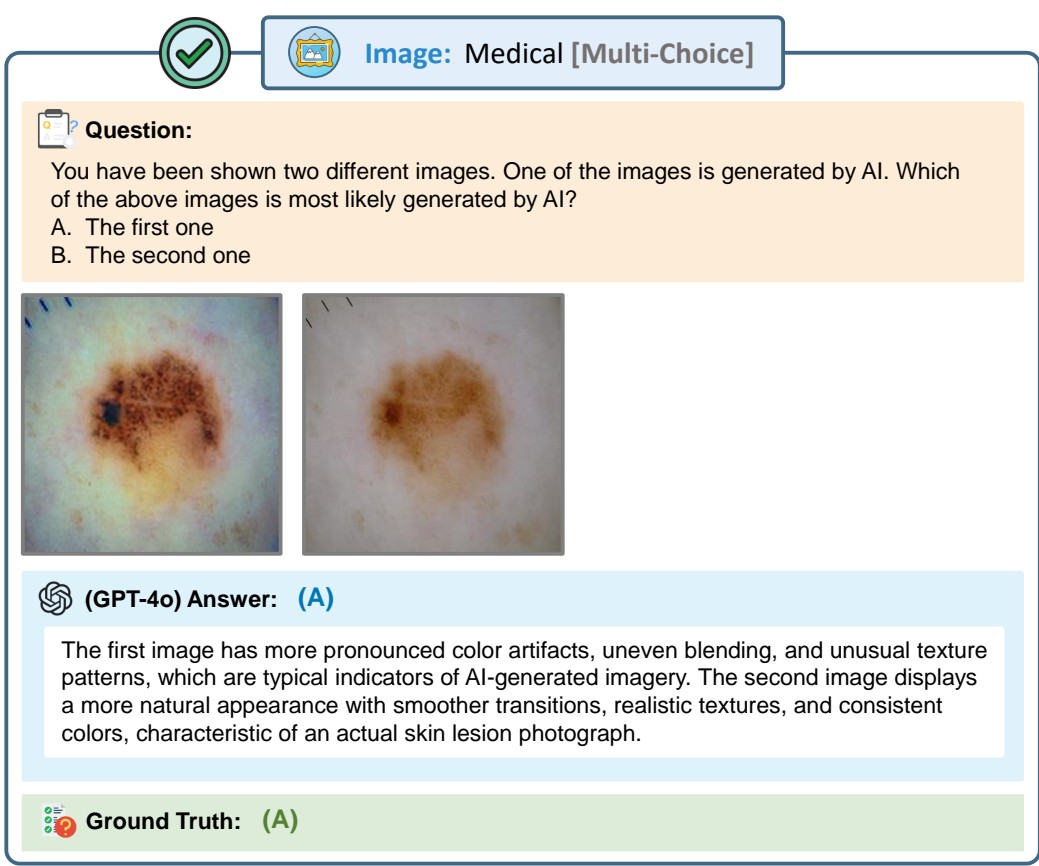

Figure 17: Multi-choice example of category **medical** in image modality, with GPT-4o's answers

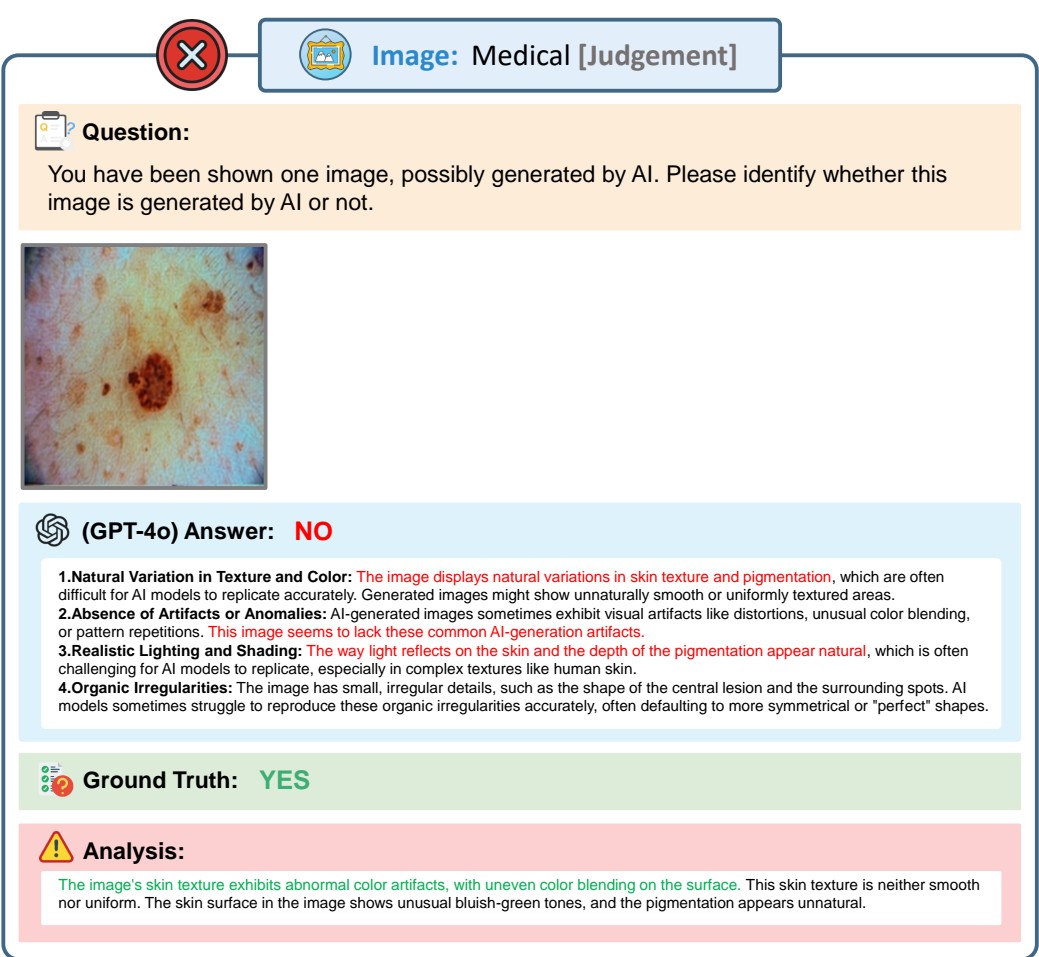

Figure 18: Judgement example of category **medical** in image modality, with GPT-4o's answers

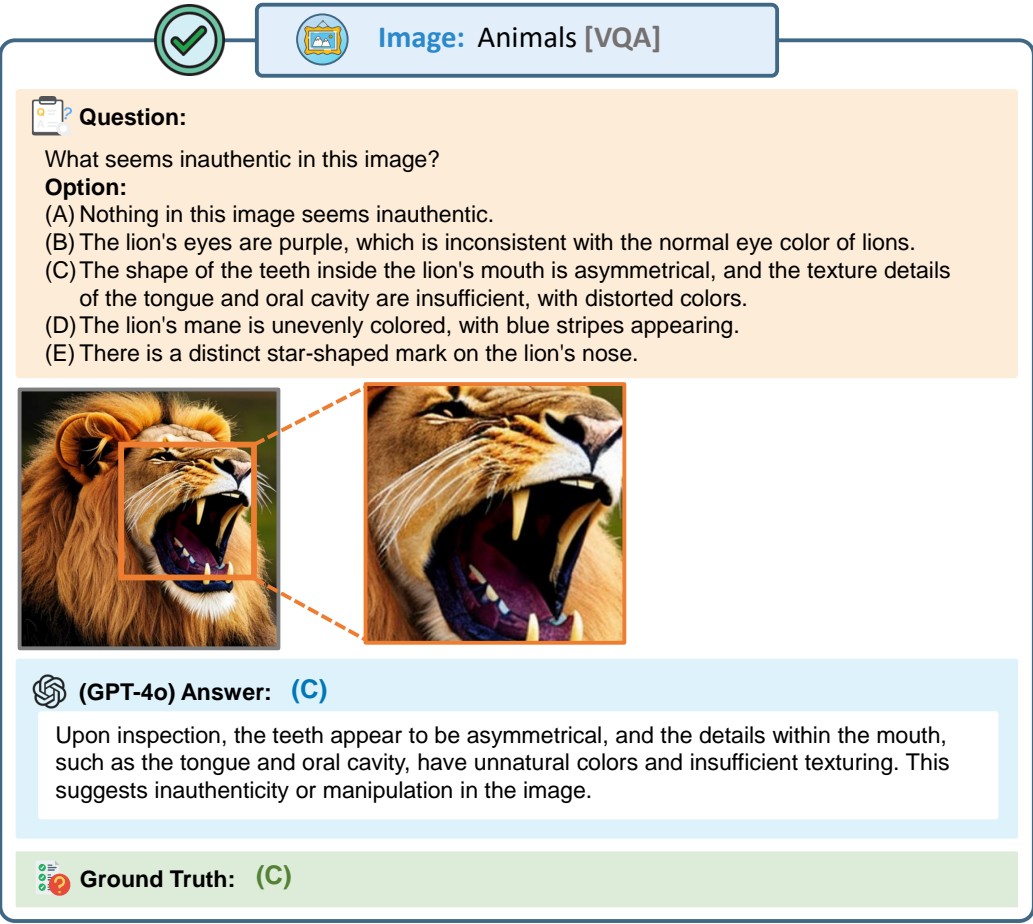

Figure 19: Abnormal selection example of category **animals** in image modality, with GPT-4o's answers

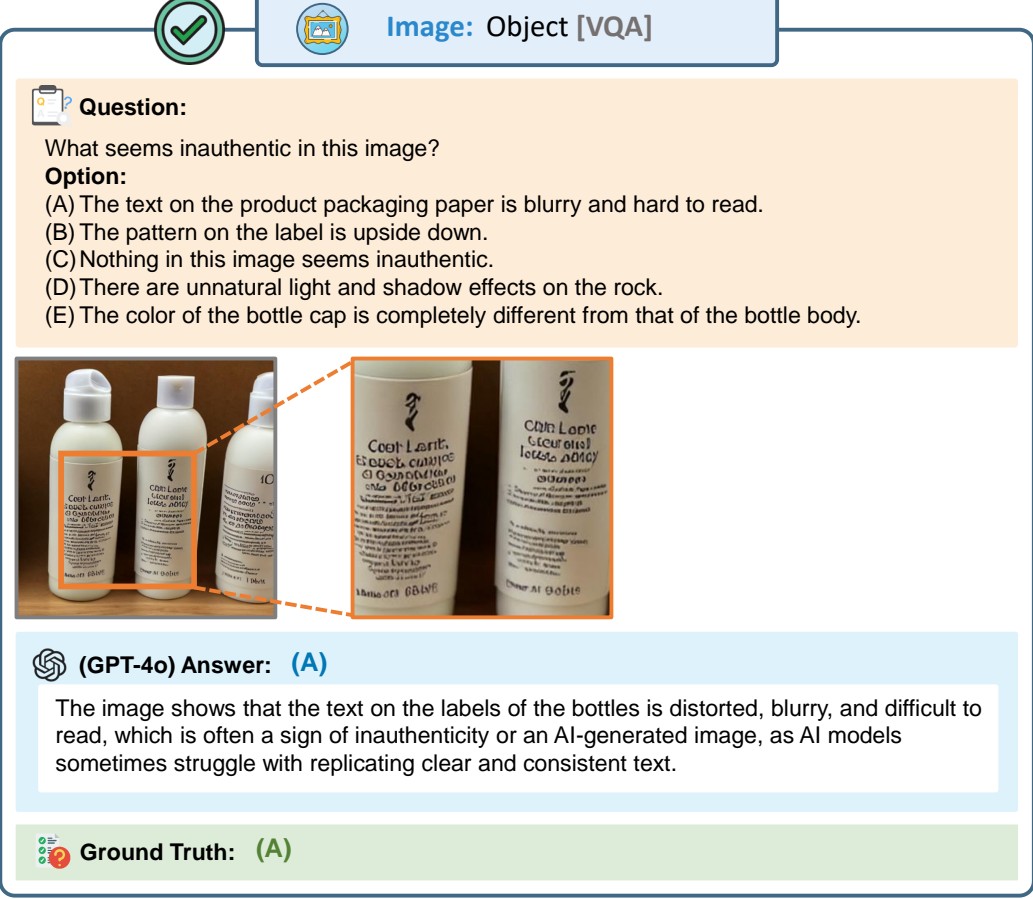

Figure 20: Abnormal selection example of category **object** in image modality, with GPT-4o's answers

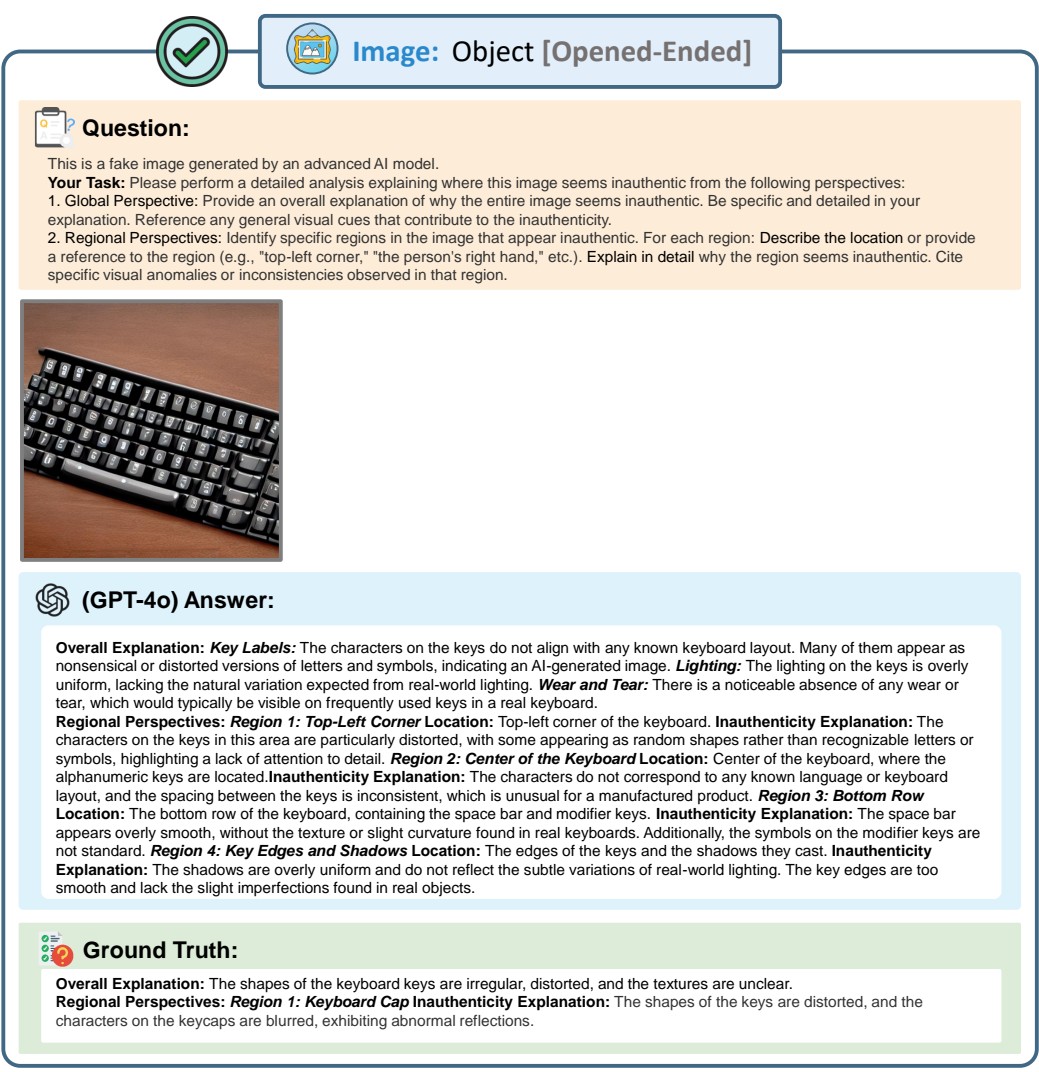

Figure 21: Abnormal detail explanation example of category **object** in image modality, with GPT-4o's answers

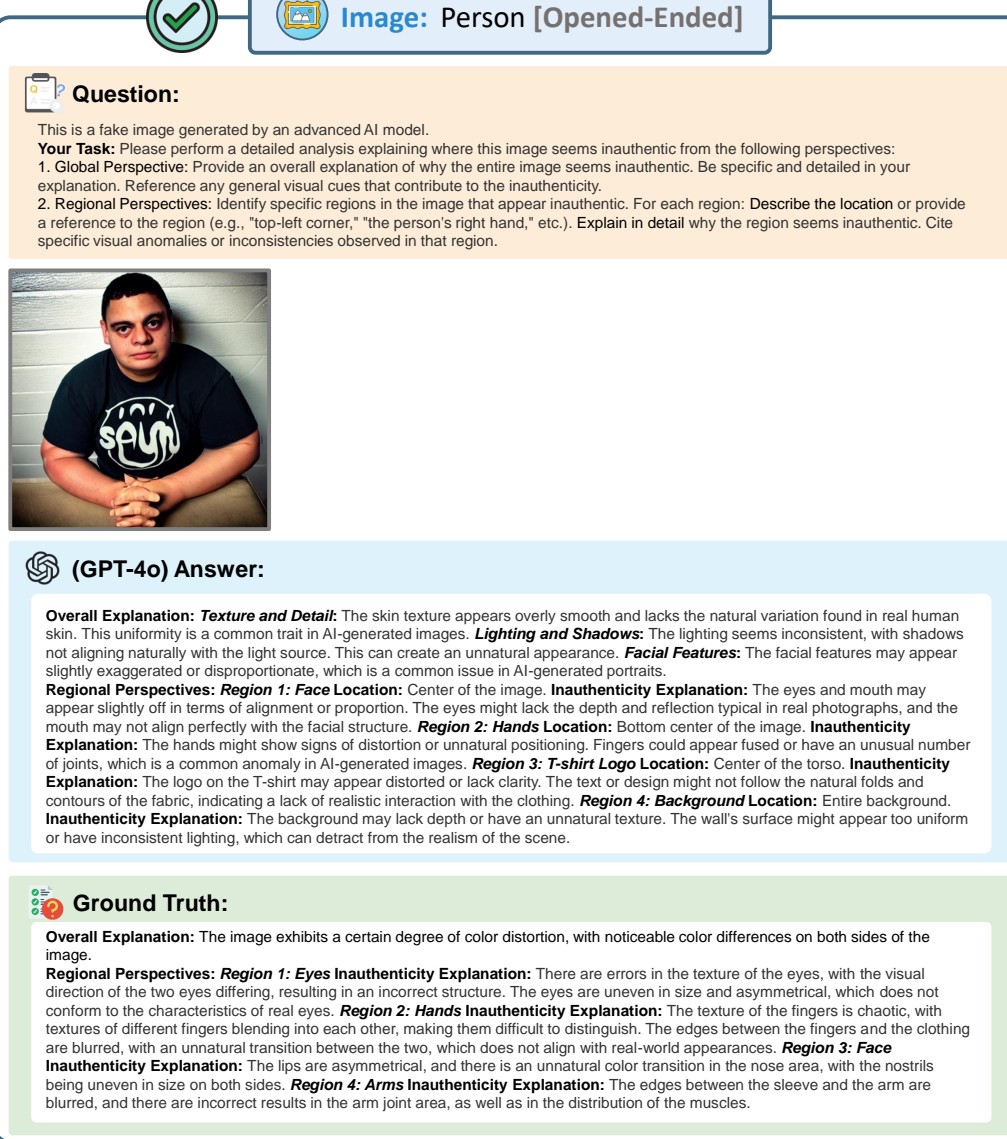

Figure 22: Abnormal detail explanation example of category **person** in image modality, with GPT-4o's answers

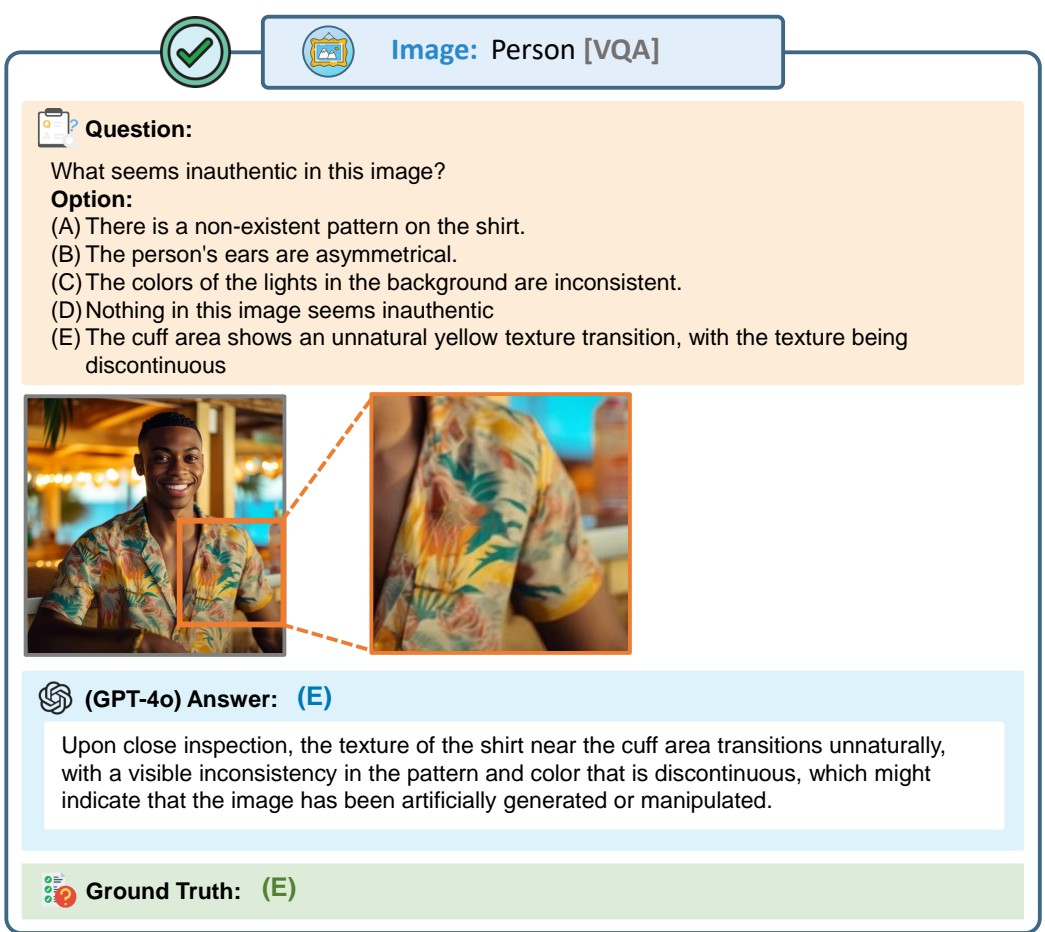

Figure 23: Abnormal selection example of category **person** in image modality, with GPT-4o's answers

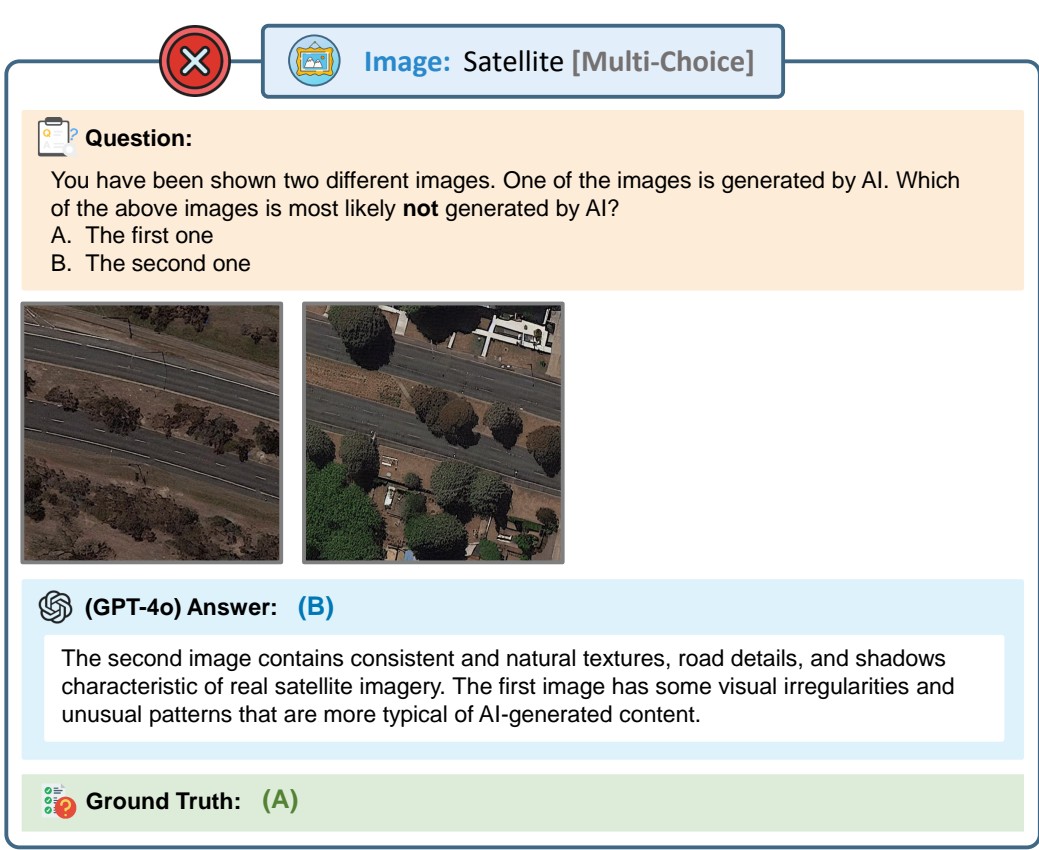

Figure 24: Multi-choice example of category **satellite** in image modality, with GPT-4o's answers

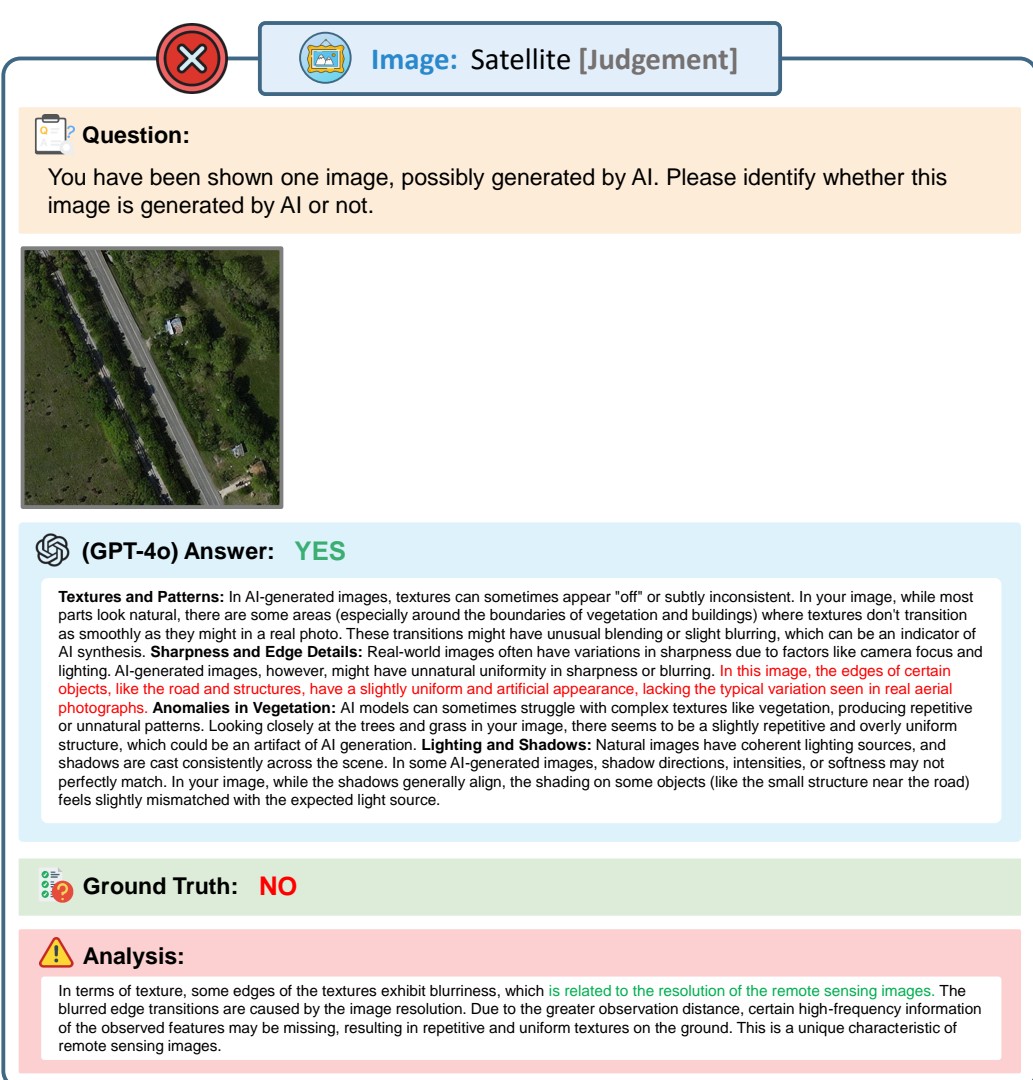

Figure 25: Judgement example of category **satellite** in image modality, with GPT-4o's answers

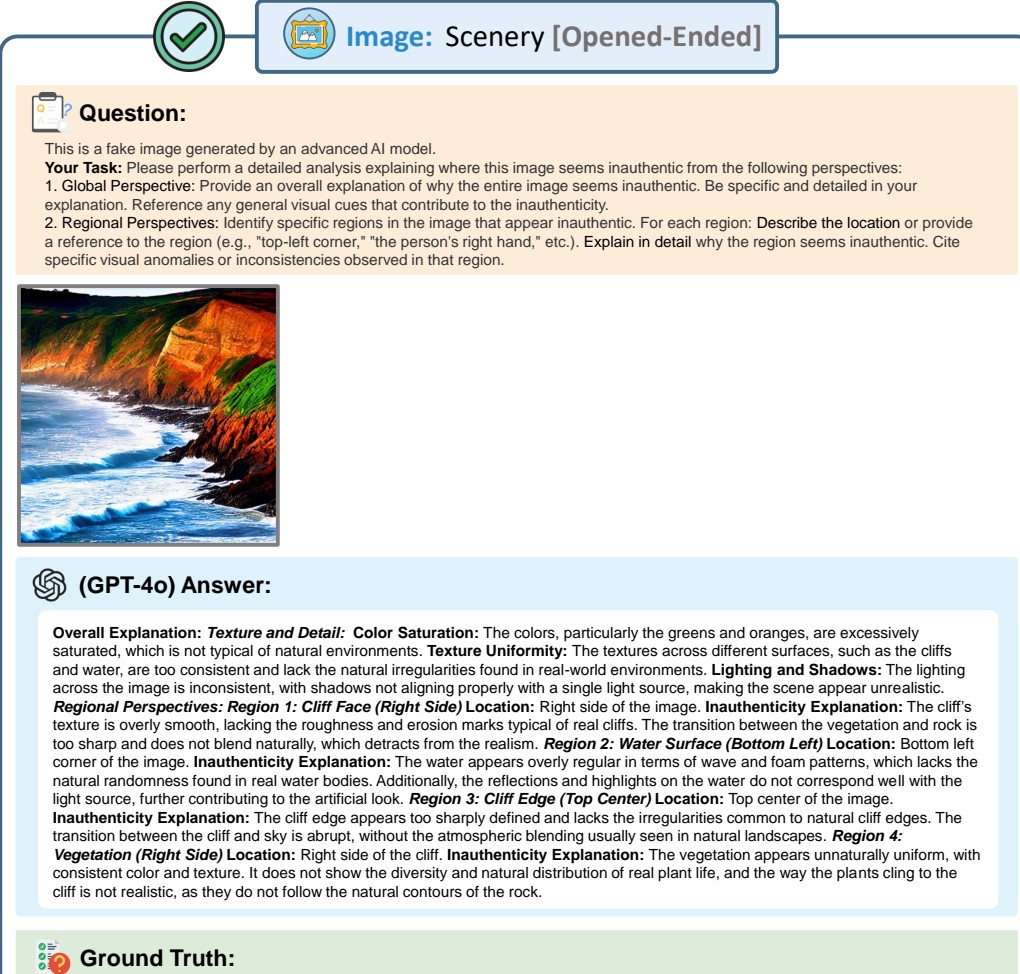

Figure 26: Abnormal detail explanation example of category **scenery** in image modality, with GPT-4o's answers

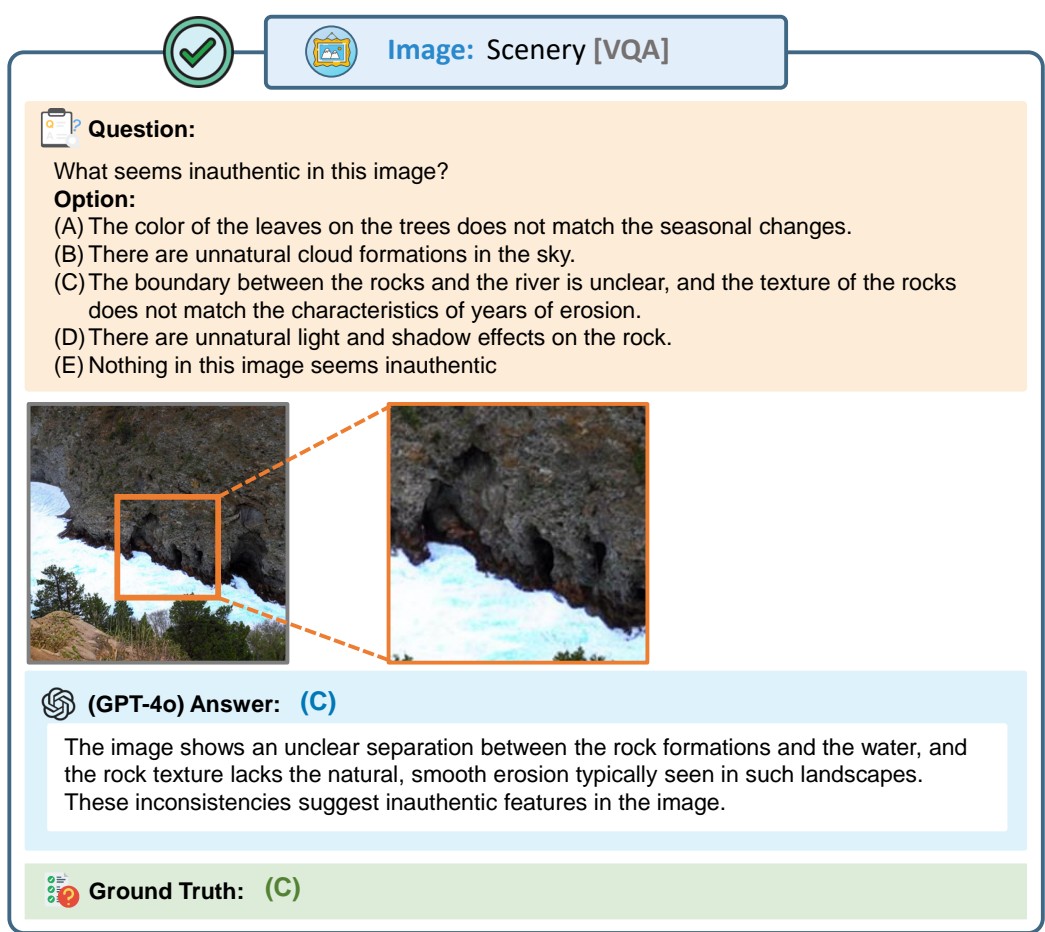

Figure 27: Abnormal selection example of category **scenery** in image modality, with GPT-4o's answers

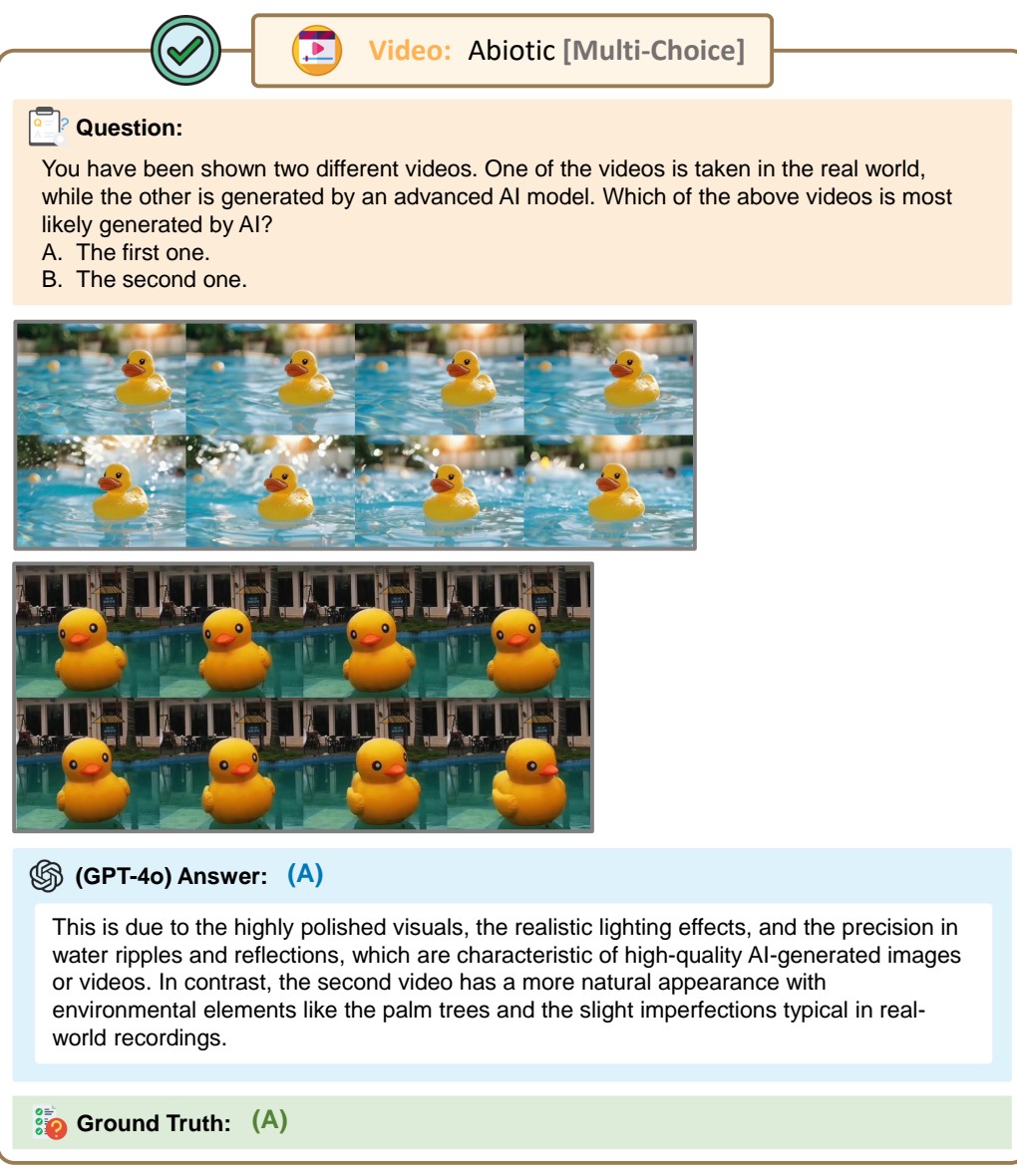

Figure 28: Multi-choice example of category **Abiotic** in video modality, with GPT-4o's answers

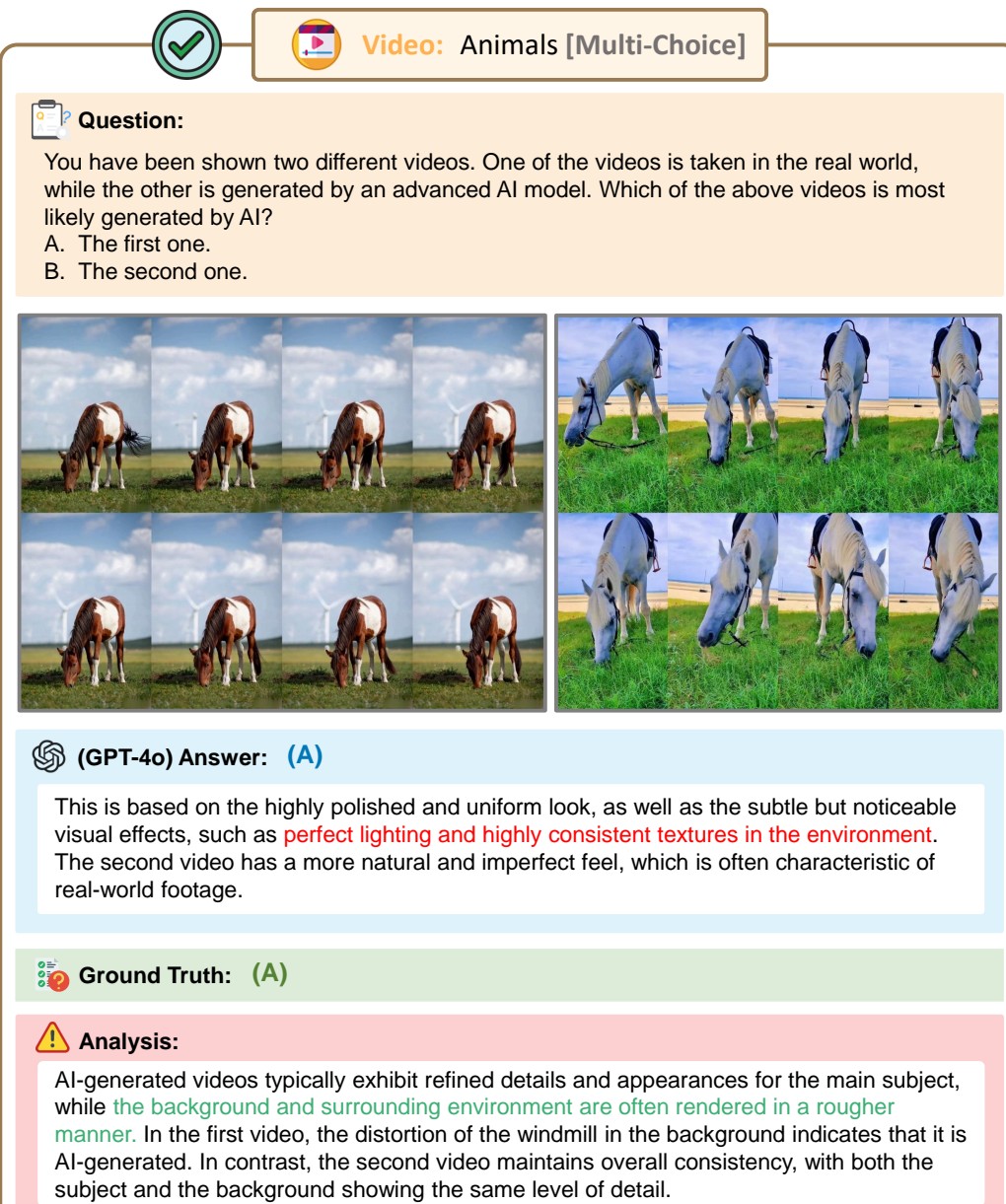

Figure 29: Multi-choice example of category **Animals** in video modality, with GPT-4o's answers

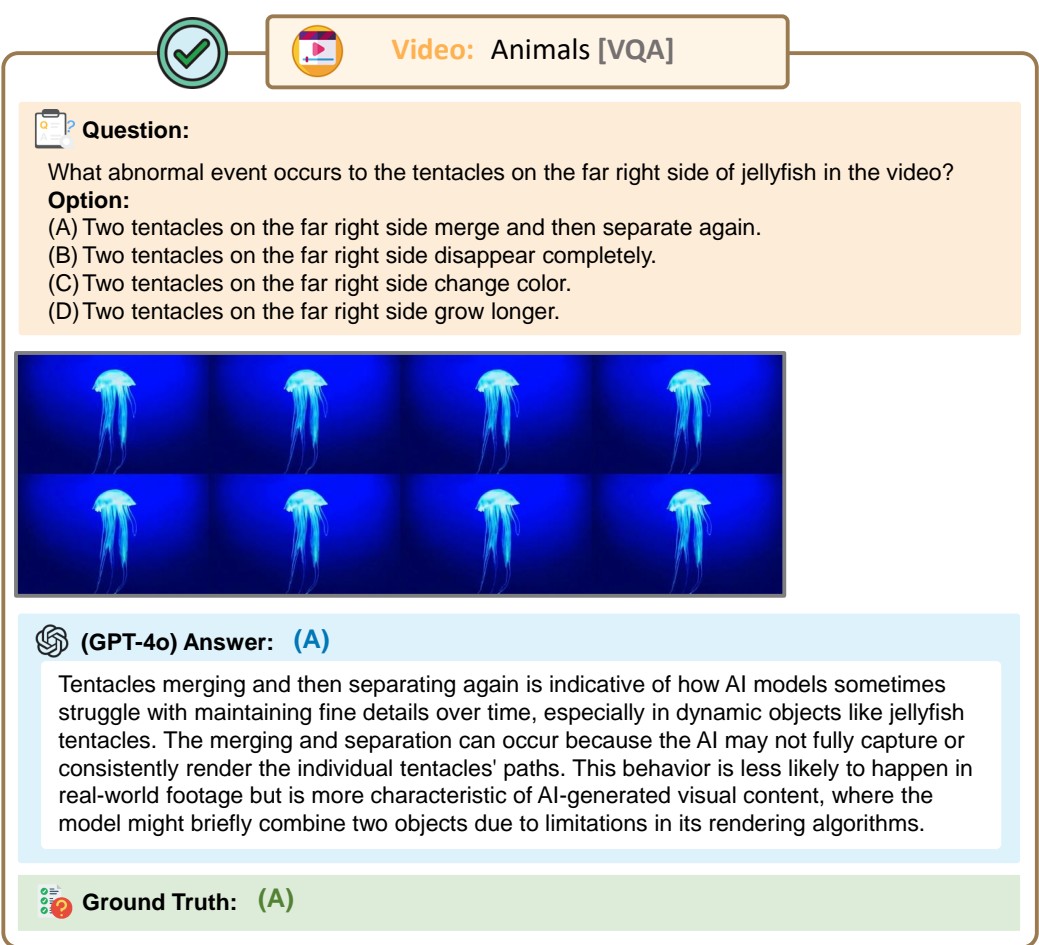

Figure 30: Abnormal selection example of category **Animals** in video modality, with GPT-4o's answers

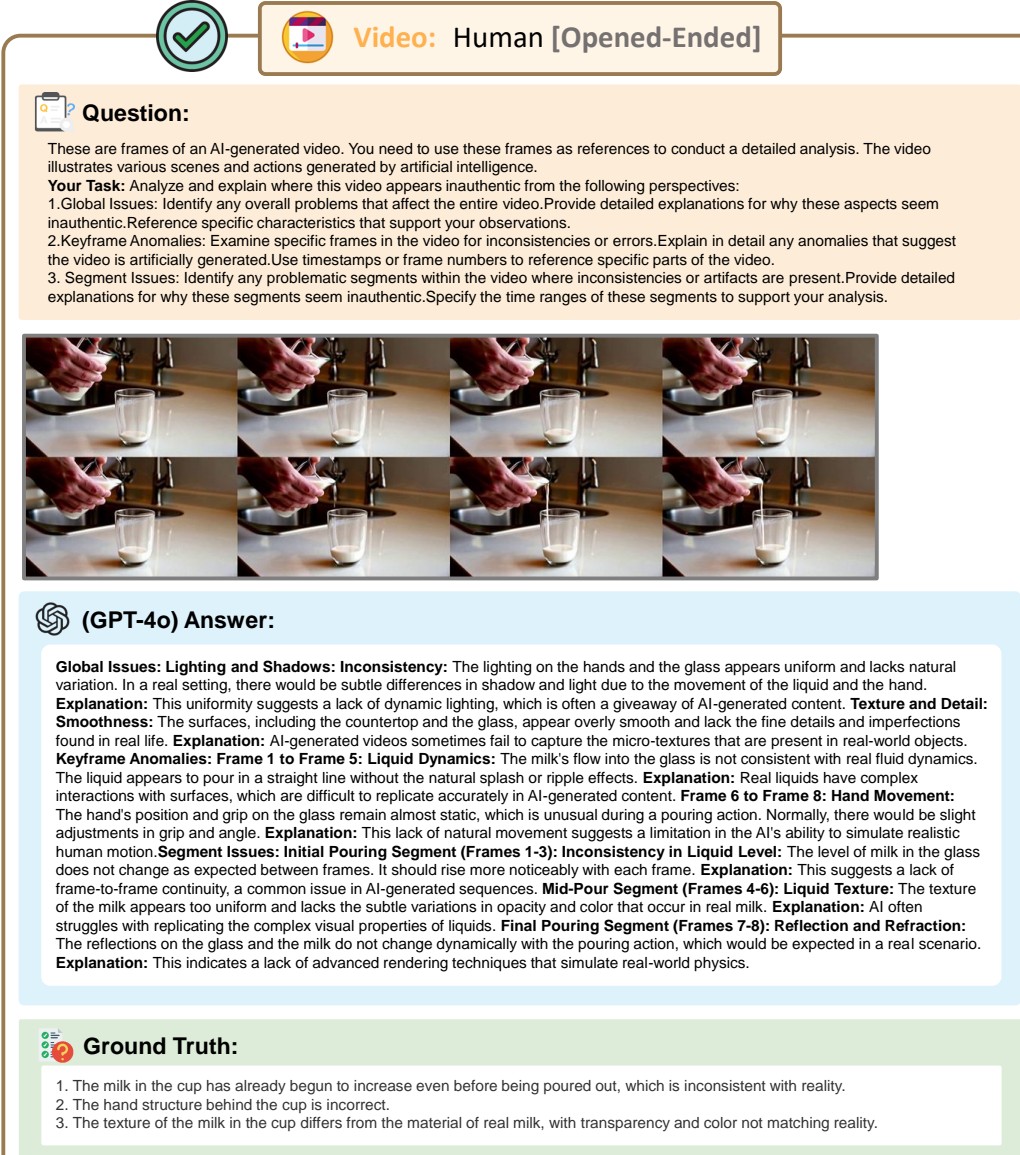

Figure 31: Abnormal detail explanation example of category **Human** in video modality, with GPT-4o's answers

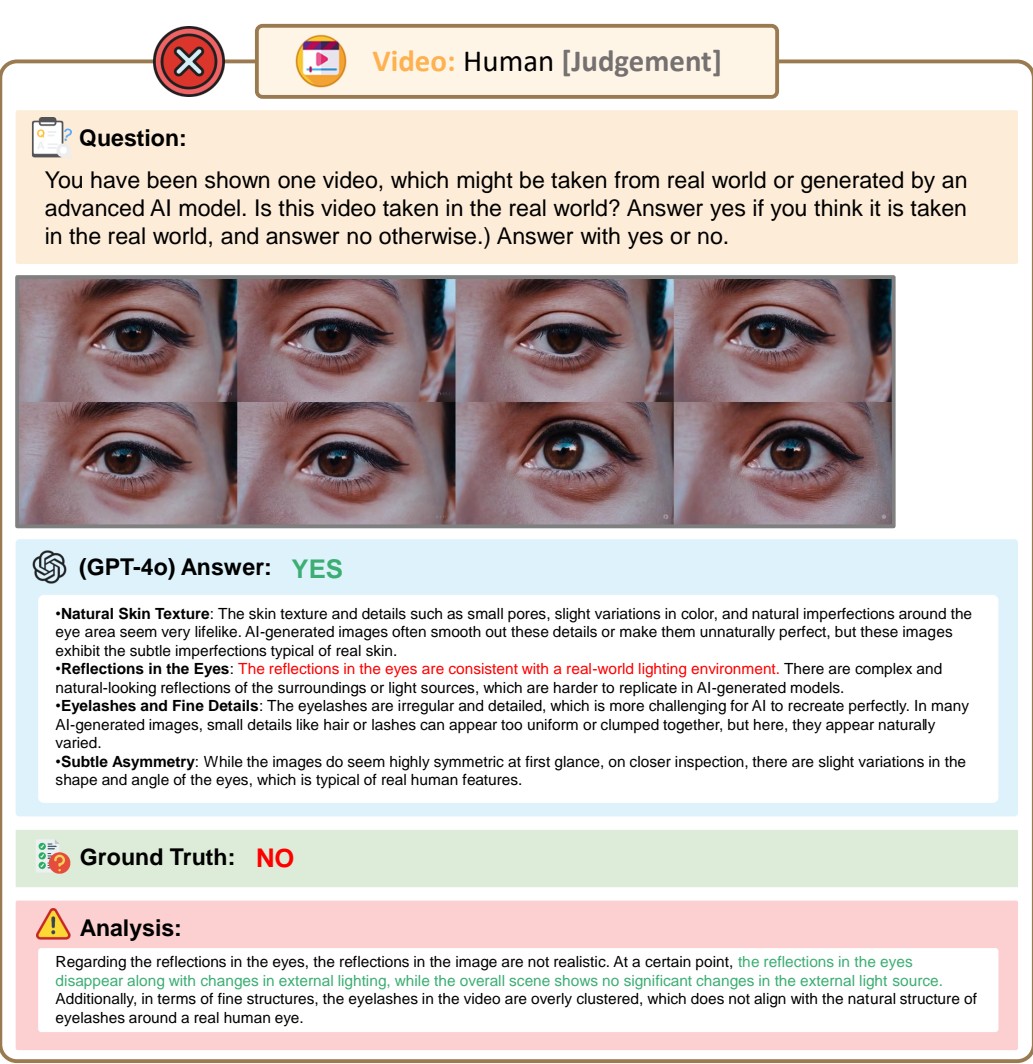

Figure 32: Judgement example of category **Human** in video modality, with GPT-4o's answers

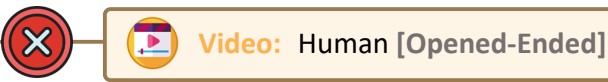

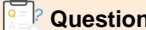

**Question:**

These are frames of an AI-generated video. You need to use these frames as references to conduct a detailed analysis. The video illustrates various scenes and actions generated by artificial intelligence.
**Your Task:** Analyze and explain where this video appears inauthentic from the following perspectives:
1.Global Issues: Identify any overall problems that affect the entire video.Provide detailed explanations for why these aspects seem inauthentic.Reference specific characteristics that support your observations.
2.Keyframe Anomalies: Examine specific frames in the video for inconsistencies or errors.Explain in detail any anomalies that suggest the video is artificially generated.Use timestamps or frame numbers to reference specific parts of the video.
3. Segment Issues: Identify any problematic segments within the video where inconsistencies or artifacts are present.Provide detailed explanations for why these segments seem inauthentic.Specify the time ranges of these segments to support your analysis.

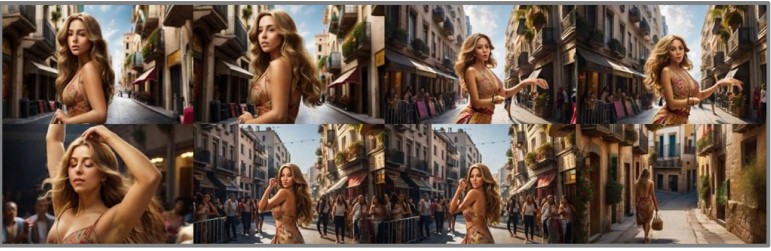

**(GPT-4o) Answer:**

**Global Issues: Unnatural Lighting and Shadows:** The lighting in the frames appears overly uniform and lacks the natural variation typically seen in real-world settings. Shadows are either too soft or inconsistently placed, which can make the scenes look flat and artificial. **Lack of Motion Blur:** In a real video, especially with movement, there would be some motion blur. The frames here are too crisp, which is a common sign of AI generation. **Texture Uniformity:** The textures, especially on the skin and clothing, appear too smooth and lack the imperfections and variations found in real life. **Explanations: Lighting:** Real-world lighting varies with the environment and time of day, creating dynamic shadows and highlights. The uniformity here suggests artificial generation. **Motion Blur:** The absence of motion blur, even in action shots, indicates a lack of real-world physics simulation. **Texture:** AI often struggles with replicating the natural imperfections found in human skin and fabric, leading to an overly polished appearance.
**Keyframe Anomalies: Frame 1 and 2: Hair Details:** The hair appears overly detailed and perfectly styled, lacking the natural movement and randomness expected in a real setting. **Frame 3: Background Consistency:** The background buildings and street appear too perfect and symmetrical, lacking the natural irregularities of real architecture. **Frame 4 and 5: Pose and Expression:** The subject's pose and expression are too static and idealized, lacking the subtle variations and imperfections of real human movement. **Frame 6: Crowd Details:** The people in the background lack distinct features and appear as generic figures, which is a common issue in AI-generated scenes. **Explanations: Hair and Pose:** AI often generates hair and poses that are too perfect, as it lacks the ability to fully replicate the randomness of real life. **Background and Crowd:** AI-generated backgrounds and crowds often lack the complexity and detail of real environments and people. **Segment Issues: Segment: Frames 1-3 Street Scene Consistency:** The street scene remains too consistent across frames, with little variation in lighting or perspective, which is unusual for a real video. **Segment: Frames 4-5 Dance Movement:** The dance movements appear too smooth and lack the natural dynamics and imperfections of real human motion. **Segment: Frames 7-8 Walking Away Scene:** The transition from the detailed street scene to the simpler alleyway lacks depth and perspective, making it appear flat and artificial. **Explanations: Street Scene:** Real videos capture subtle changes in lighting and perspective, which are missing here. **Dance Movement:** Real human motion includes slight variations and imperfections that are difficult for AI to replicate. **Walking Scene:** The lack of depth and perspective in the transition suggests artificial generation.

**Ground Truth:**

**Global Issues:** 1. The video style leans towards painting, and the image flickers. 2. The facial features and hands of the character undergo severe distortion. **Fragment Issues:** 1.1. The facial features of the character are distorted. 1.2. The person's hands experience severe distortion, with the right hand merging with the street and the left hand merging with the shop. 2.1. The main character's facial features undergo severe distortion. 2.2. The image of the passerby is completely incorrect. 3.1. The body structure of the passerby in the background is incorrect.

Figure 33: Abnormal detail explanation example of category **Human** in video modality, with GPT-4o's answers

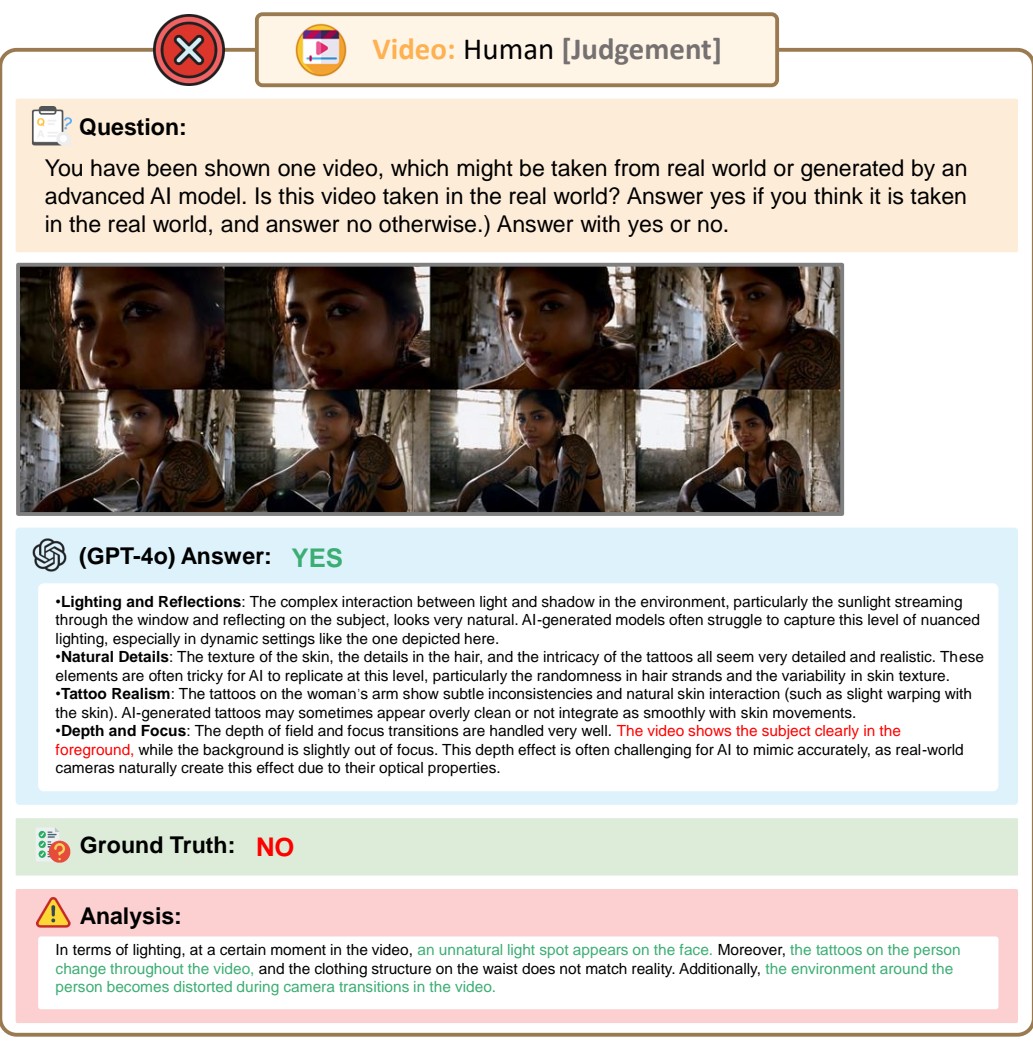

Figure 34: Judgement example of category **Human** in video modality, with GPT-4o's answers

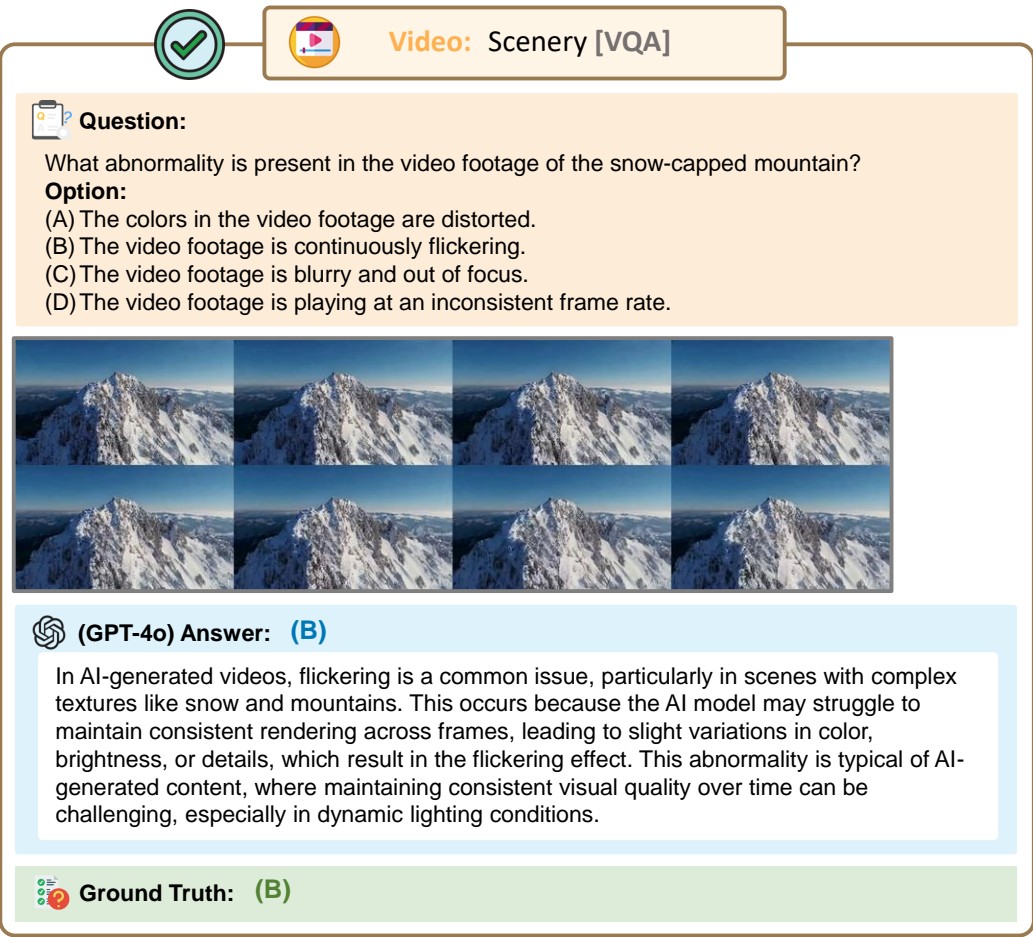

Figure 35: Abnormal selection of category **Scenery** in video modality, with GPT-4o's answers

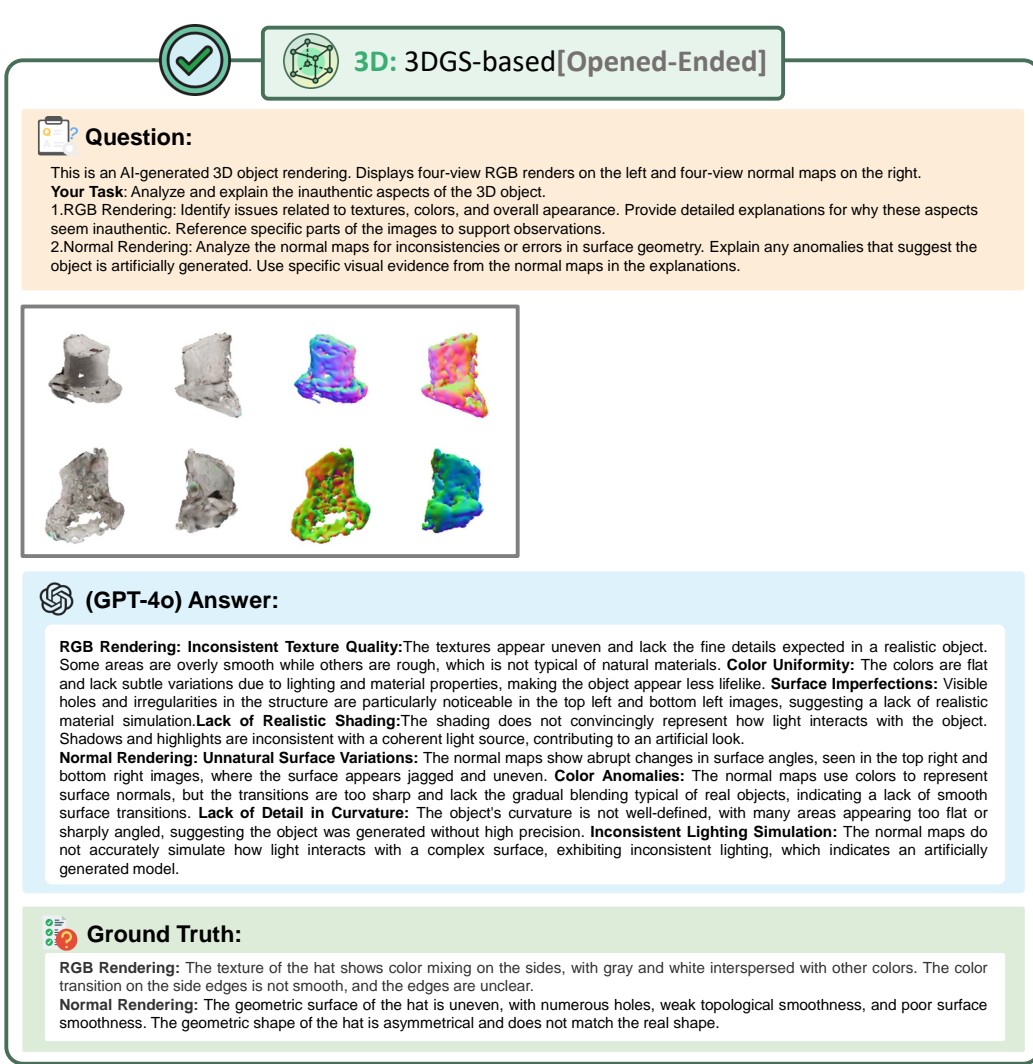

Figure 36: Abnormal detail explanation example generated by **Gaussian Splatting based** methods in 3D modality, with GPT-4o's answers

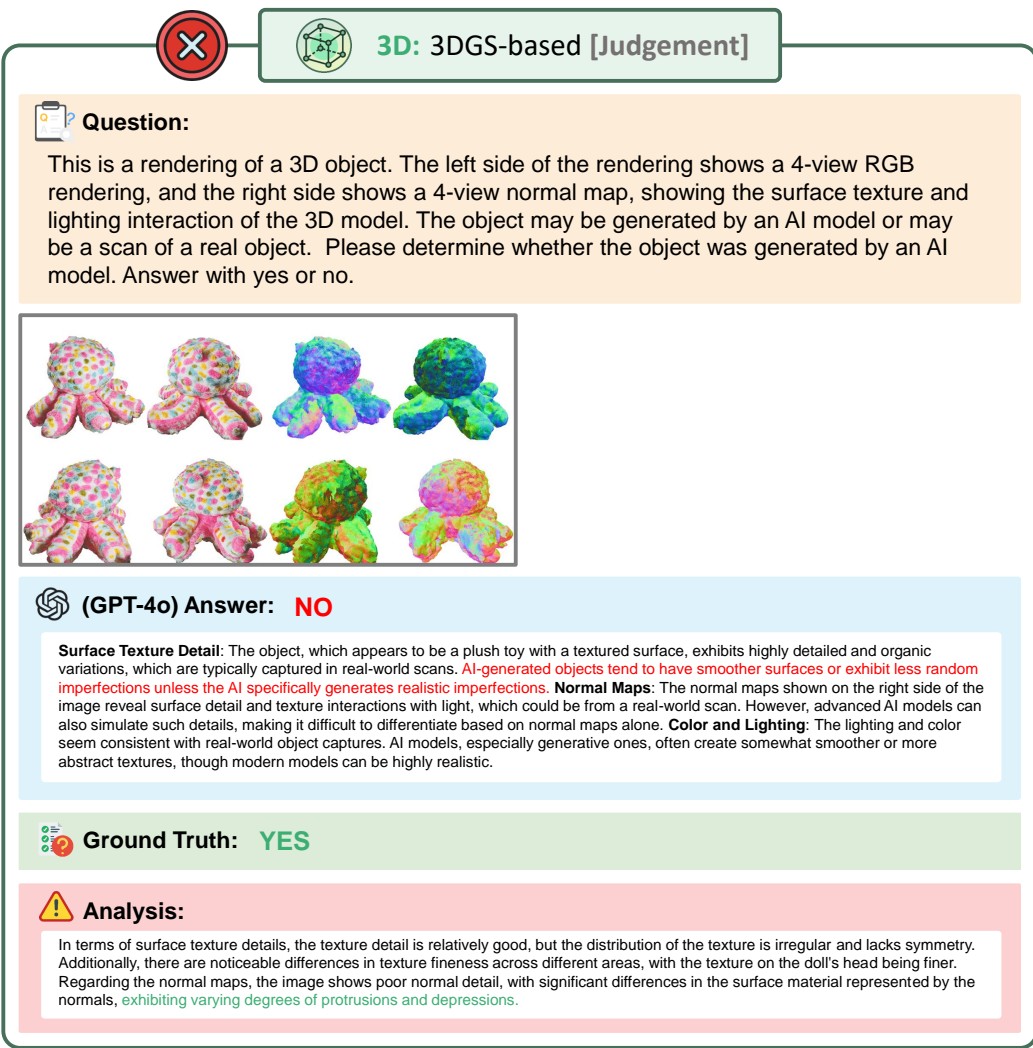

Figure 37: Judgement example generated by **Gaussian Splatting based** methods in 3D modality, with GPT-4o's answers

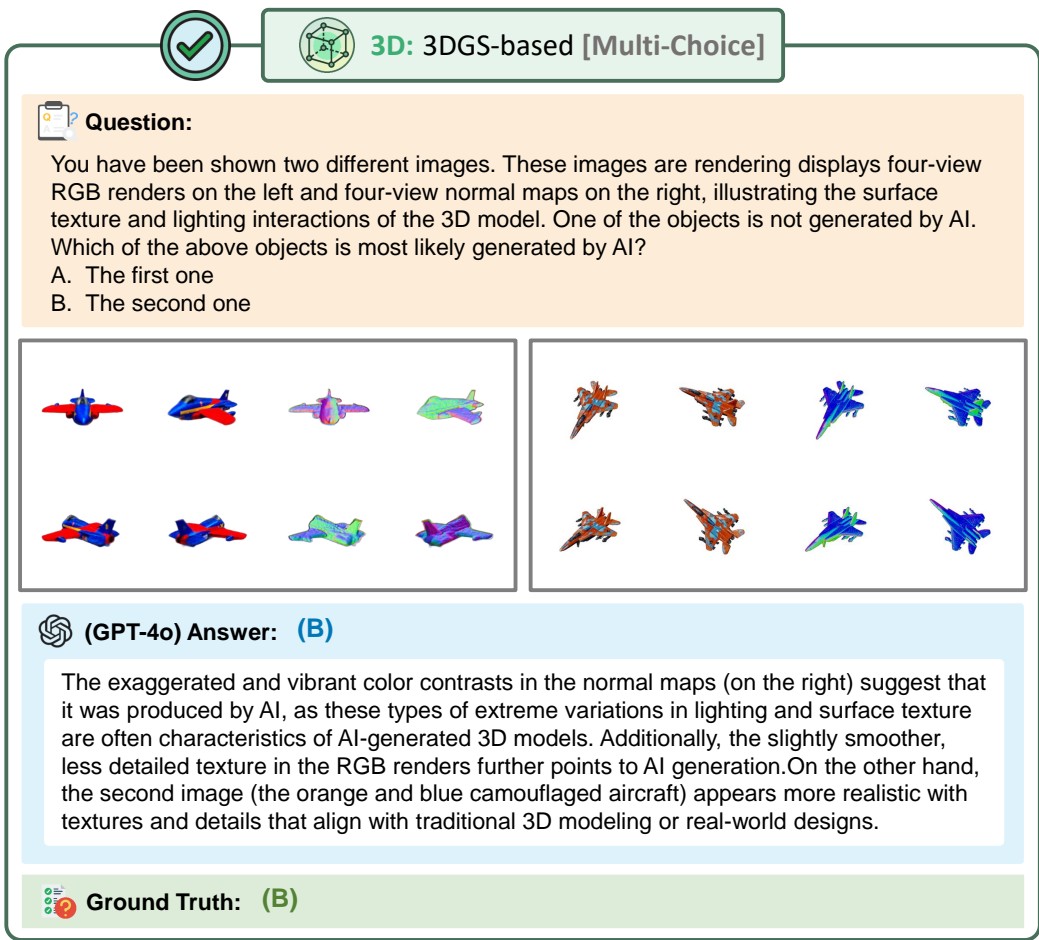

Figure 38: Multi-choice example generated by **Gaussian Splatting based** methods in 3D modality, with GPT-4o's answers

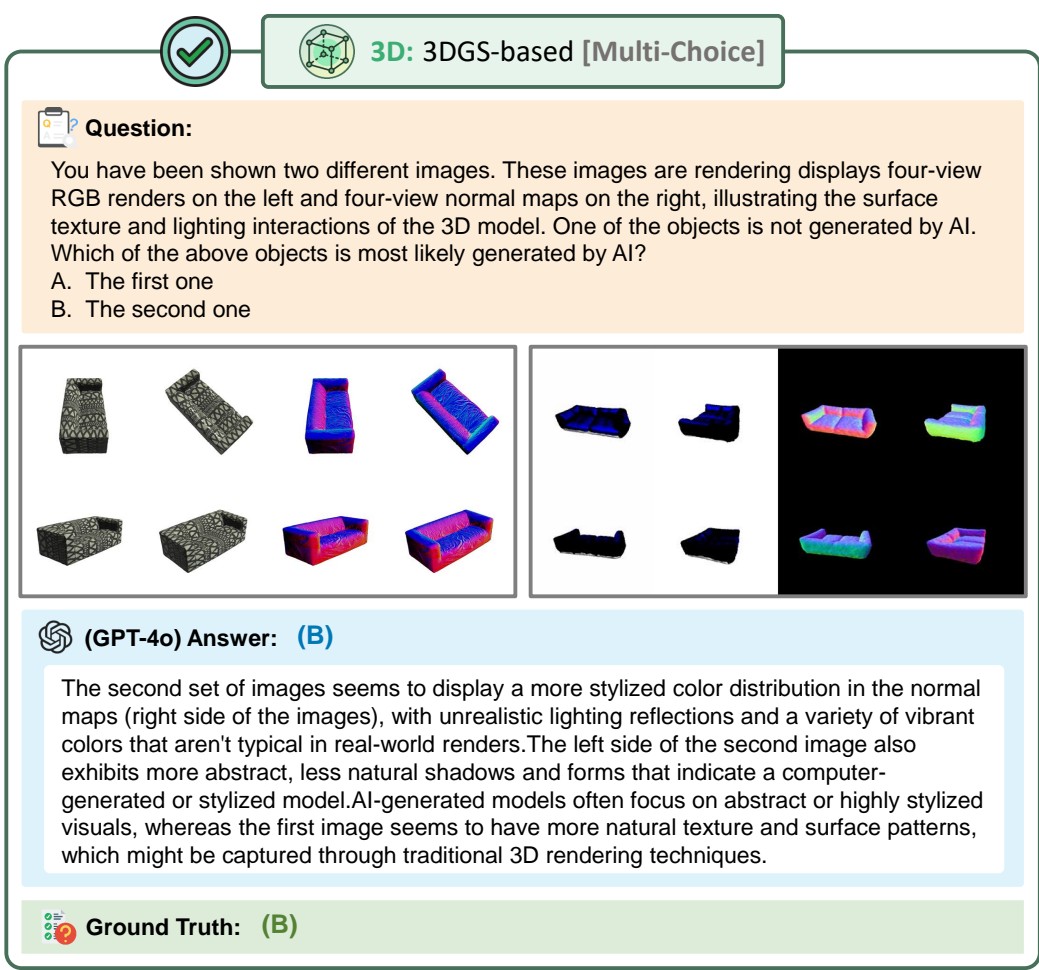

**3D:** 3DGS-based [Multi-Choice]

**Question:**

You have been shown two different images. These images are rendering displays four-view RGB renders on the left and four-view normal maps on the right, illustrating the surface texture and lighting interactions of the 3D model. One of the objects is not generated by AI. Which of the above objects is most likely generated by AI?
A. The first one
B. The second one

**(GPT-4o) Answer:** (B)

The second set of images seems to display a more stylized color distribution in the normal maps (right side of the images), with unrealistic lighting reflections and a variety of vibrant colors that aren't typical in real-world renders.The left side of the second image also exhibits more abstract, less natural shadows and forms that indicate a computer-generated or stylized model.AI-generated models often focus on abstract or highly stylized visuals, whereas the first image seems to have more natural texture and surface patterns, which might be captured through traditional 3D rendering techniques.

**Ground Truth:** (B)

Figure 39: Multi-choice example generated by **Gaussian Splatting based** methods in 3D modality, with GPT-4o's answers

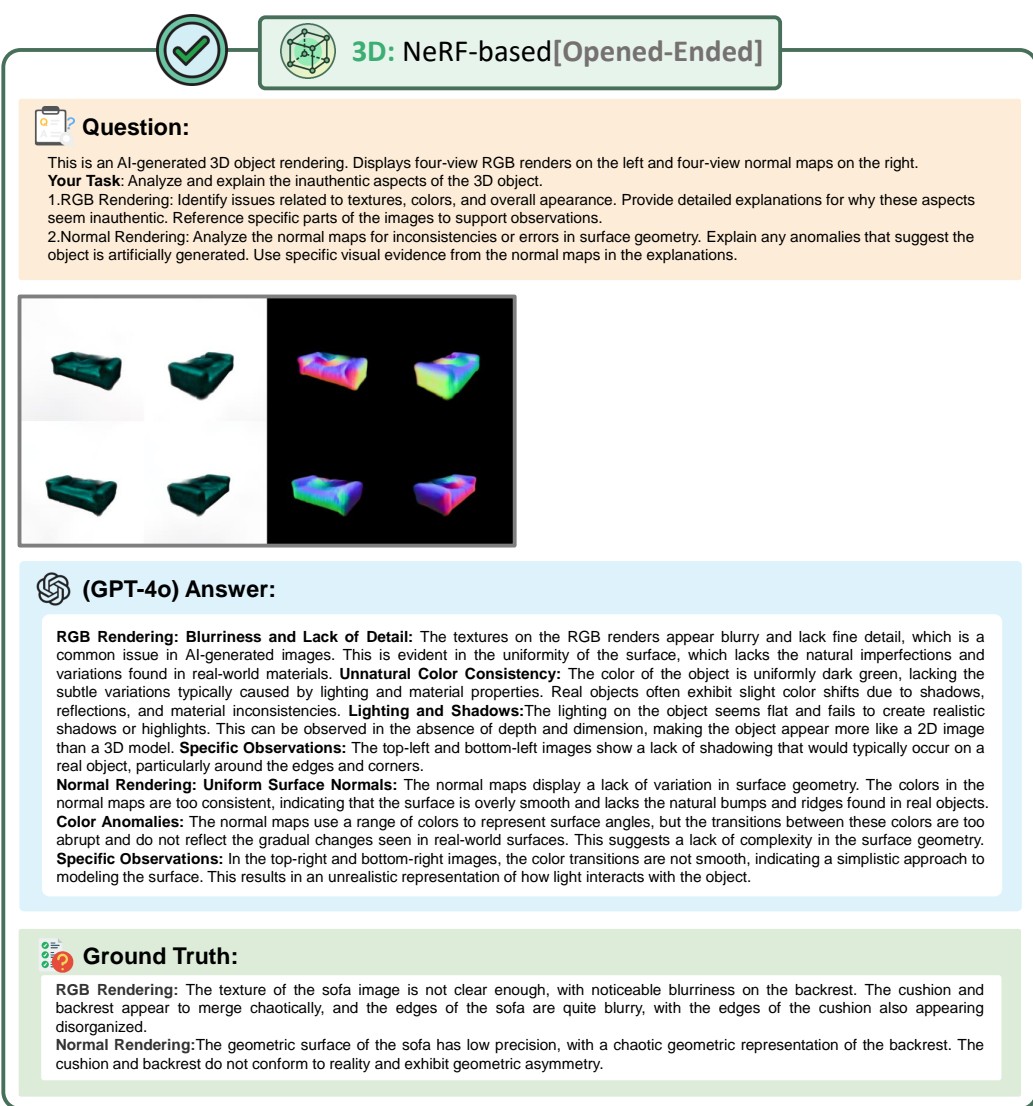

Figure 40: Abnormal detail explanation example generated by **Nerf based** in 3D modality, with GPT-4o's answers

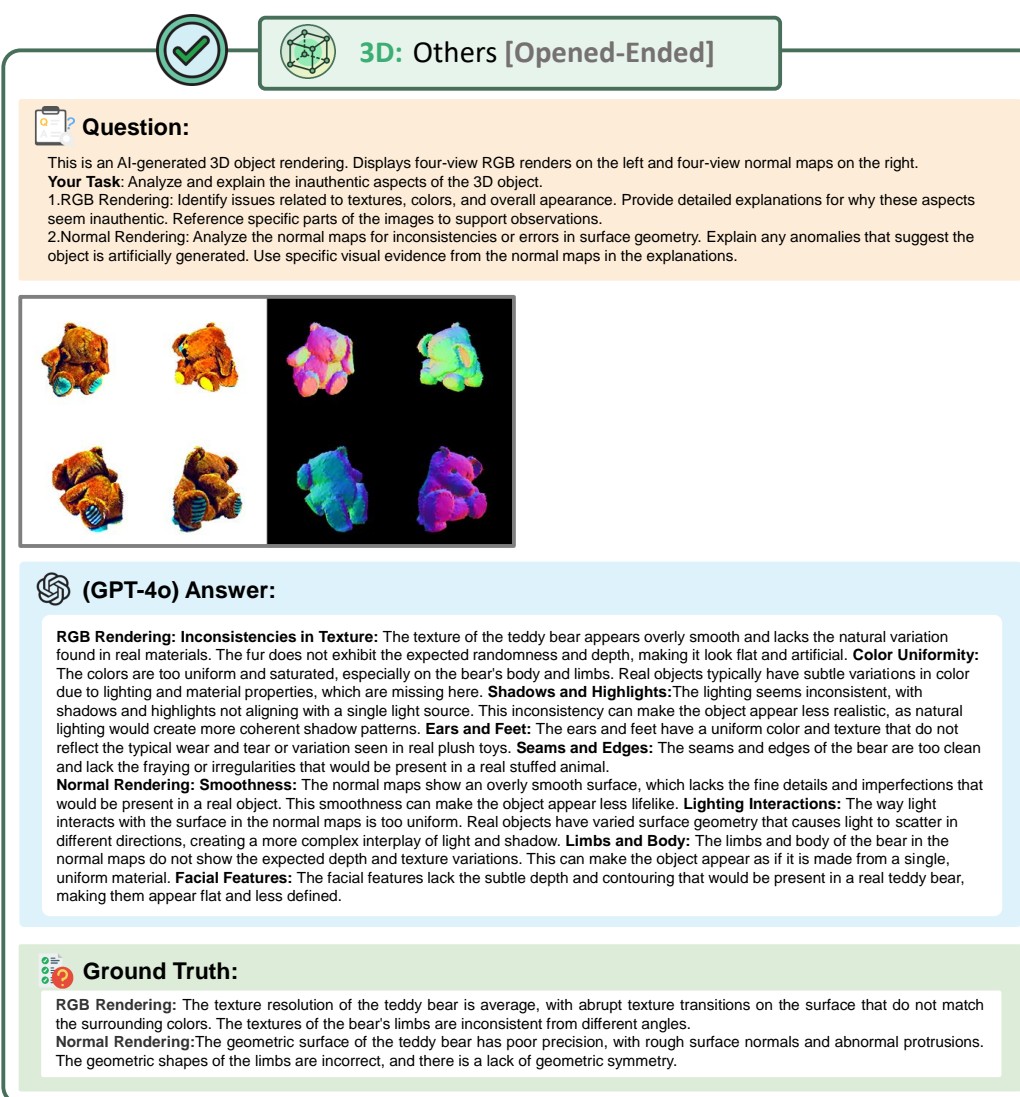

**3D:** Others **[Opened-Ended]**

**Question:**

This is an AI-generated 3D object rendering. Displays four-view RGB renders on the left and four-view normal maps on the right.
**Your Task**: Analyze and explain the inauthentic aspects of the 3D object.
1.RGB Rendering: Identify issues related to textures, colors, and overall apearance. Provide detailed explanations for why these aspects seem inauthentic. Reference specific parts of the images to support observations.
2.Normal Rendering: Analyze the normal maps for inconsistencies or errors in surface geometry. Explain any anomalies that suggest the object is artificially generated. Use specific visual evidence from the normal maps in the explanations.

**(GPT-4o) Answer:**

**RGB Rendering: Inconsistencies in Texture:** The texture of the teddy bear appears overly smooth and lacks the natural variation found in real materials. The fur does not exhibit the expected randomness and depth, making it look flat and artificial. **Color Uniformity:** The colors are too uniform and saturated, especially on the bear's body and limbs. Real objects typically have subtle variations in color due to lighting and material properties, which are missing here. **Shadows and Highlights:**The lighting seems inconsistent, with shadows and highlights not aligning with a single light source. This inconsistency can make the object appear less realistic, as natural lighting would create more coherent shadow patterns. **Ears and Feet:** The ears and feet have a uniform color and texture that do not reflect the typical wear and tear or variation seen in real plush toys. **Seams and Edges:** The seams and edges of the bear are too clean and lack the fraying or irregularities that would be present in a real stuffed animal.
**Normal Rendering: Smoothness:** The normal maps show an overly smooth surface, which lacks the fine details and imperfections that would be present in a real object. This smoothness can make the object appear less lifelike. **Lighting Interactions:** The way light interacts with the surface in the normal maps is too uniform. Real objects have varied surface geometry that causes light to scatter in different directions, creating a more complex interplay of light and shadow. **Limbs and Body:** The limbs and body of the bear in the normal maps do not show the expected depth and texture variations. This can make the object appear as if it is made from a single, uniform material. **Facial Features:** The facial features lack the subtle depth and contouring that would be present in a real teddy bear, making them appear flat and less defined.

**Ground Truth:**

**RGB Rendering:** The texture resolution of the teddy bear is average, with abrupt texture transitions on the surface that do not match the surrounding colors. The textures of the bear's limbs are inconsistent from different angles.
**Normal Rendering:**The geometric surface of the teddy bear has poor precision, with rough surface normals and abnormal protrusions. The geometric shapes of the limbs are incorrect, and there is a lack of geometric symmetry.

Figure 41: Abnormal detail explanation example in 3D modality, with GPT-4o's answers

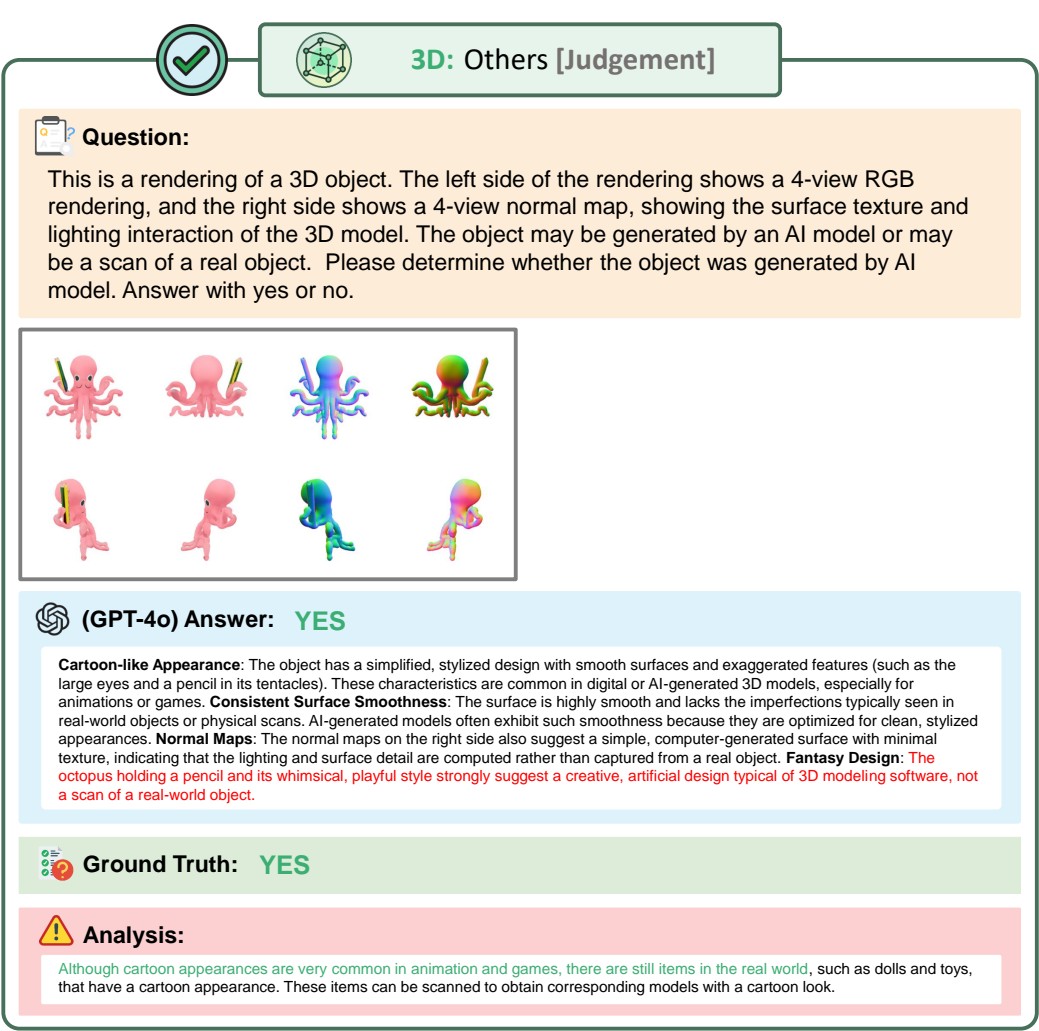

Figure 42: Judgement example in 3D modality, with GPT-4o's answers

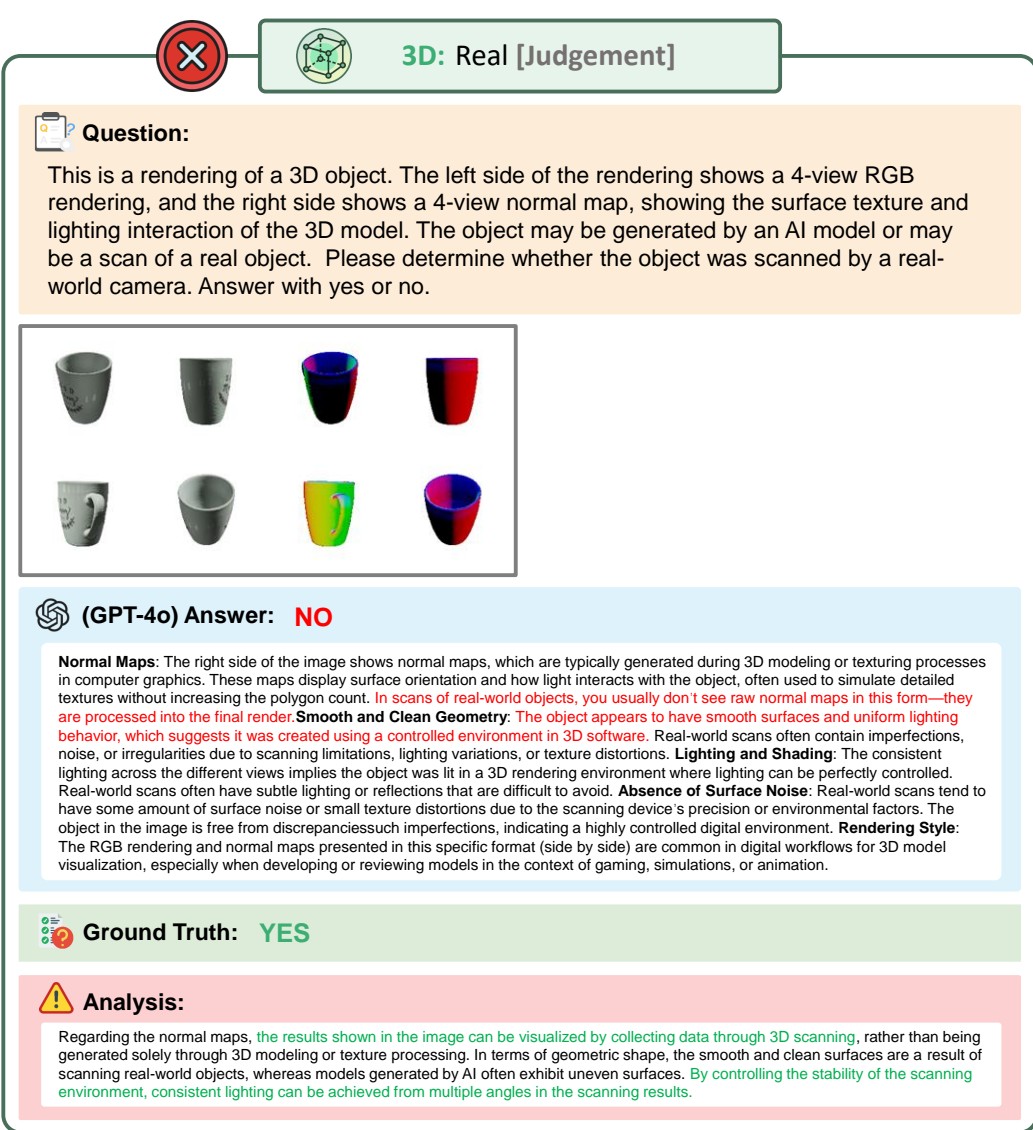

Figure 43: Judgement example in 3D modality, with GPT-4o's answers

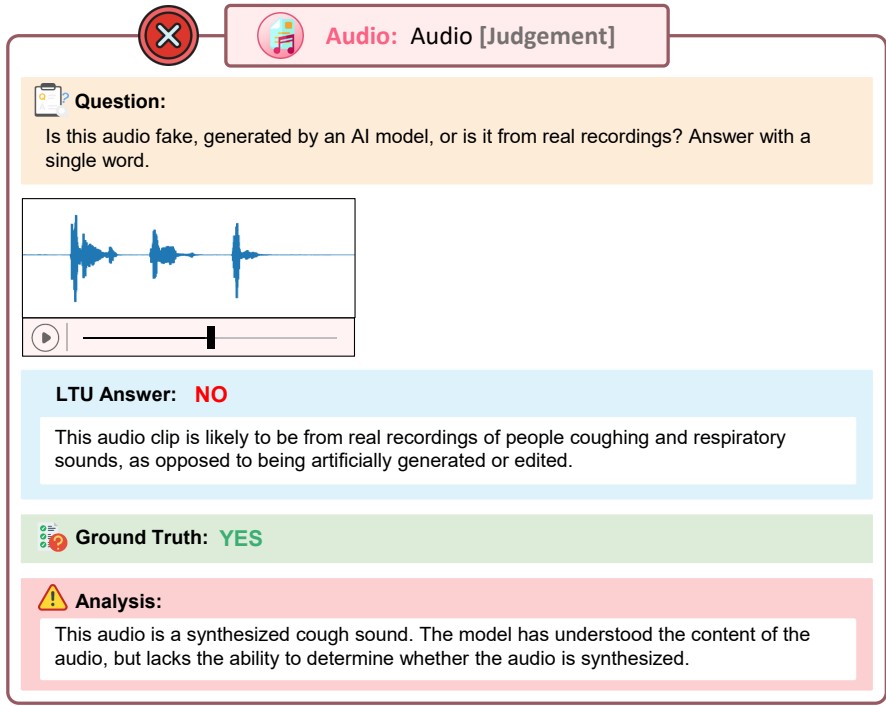

Figure 44: Judgement example of category **Audio** in audio modality, with LTU's answers

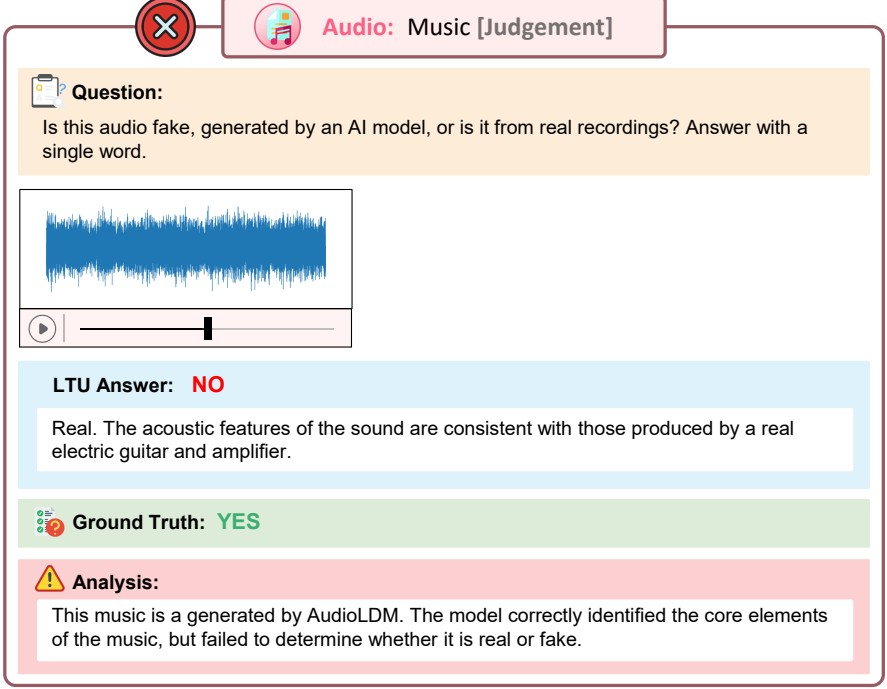

Figure 45: Multi-choice example of category **Music** in audio modality, with LTU's answers

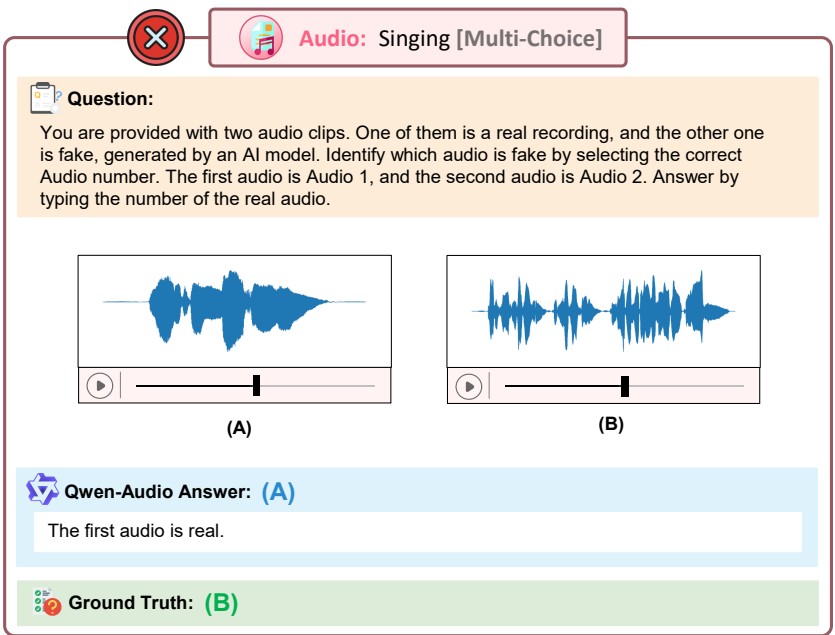

Figure 46: Multi-choice example of category **Singing** in Audio modality, with Qwen-Audio's answers

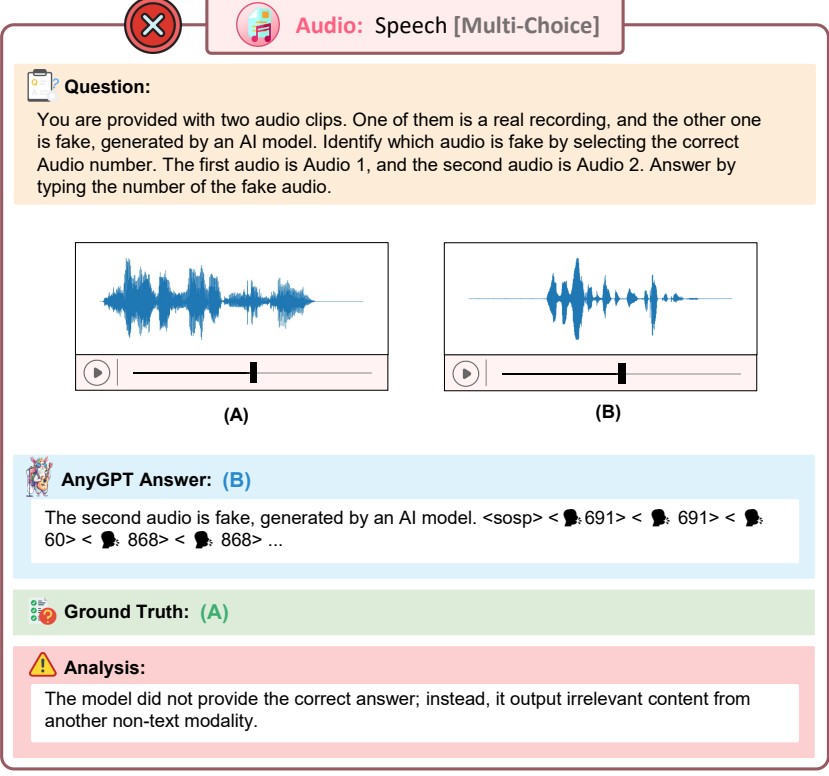

Figure 47: Multi-choice example of category **Speech** in audio modality, with AnyGPT's answers

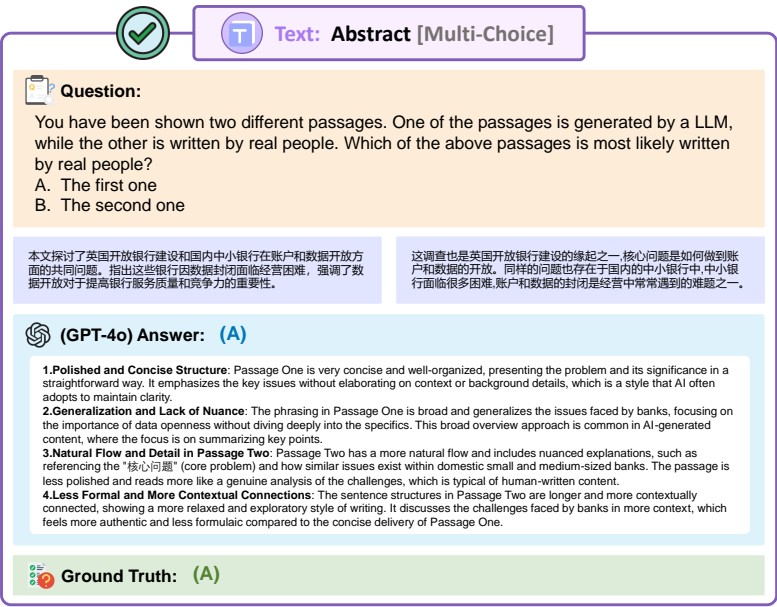

Figure 48: Multi-choice example of category **Scientific Papers Abstract** in text modality, with GPT-4o's answers

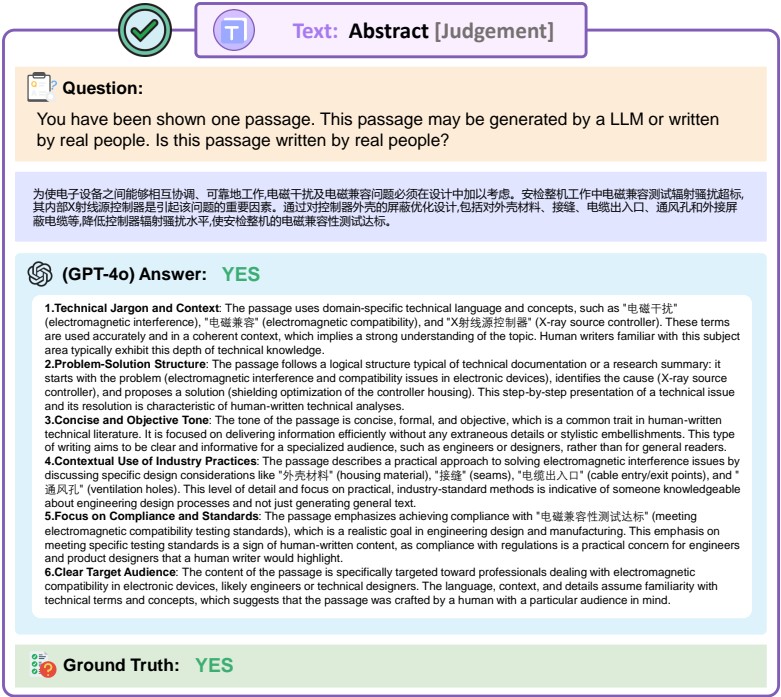

Figure 49: Judgement example of category **Scientific Papers Abstract** in text modality, with GPT-4o's answers

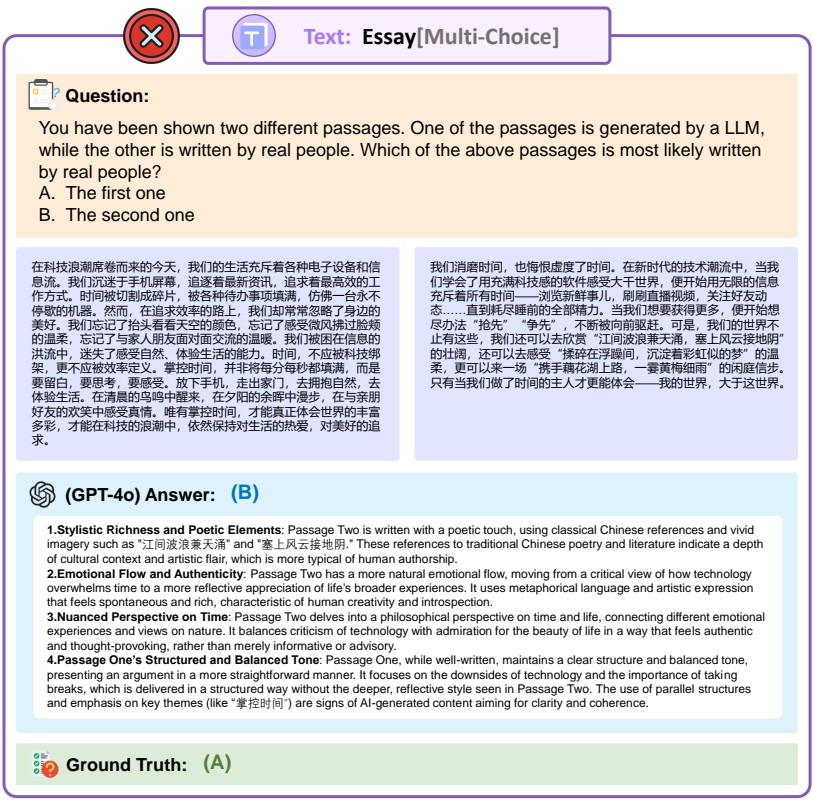

Figure 50: Multi-choice example of category **Essay** in text modality, with GPT-4o's answers

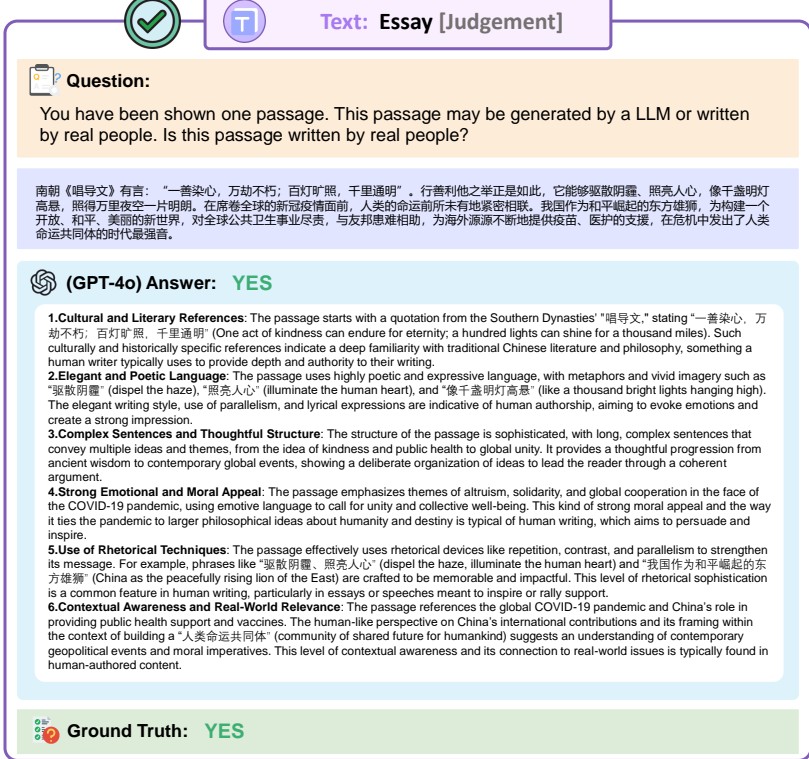

Figure 51: Judgement example of category **Essay** in text modality, with GPT-4o's answers

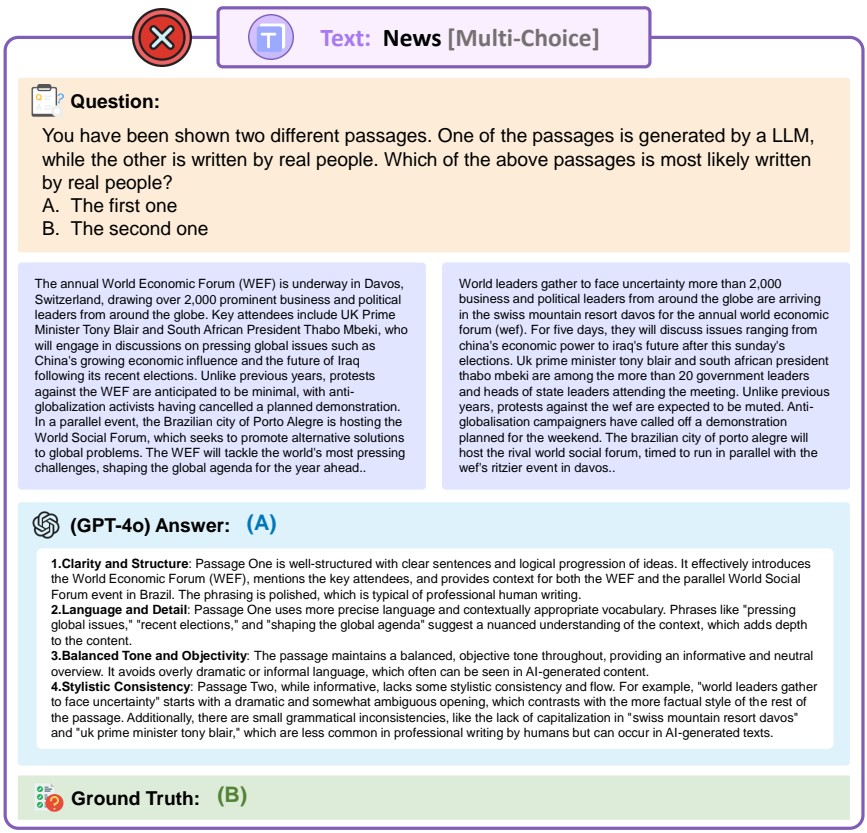

Figure 52: Multi-choice example of category **News** in text modality, with GPT-4o's answers

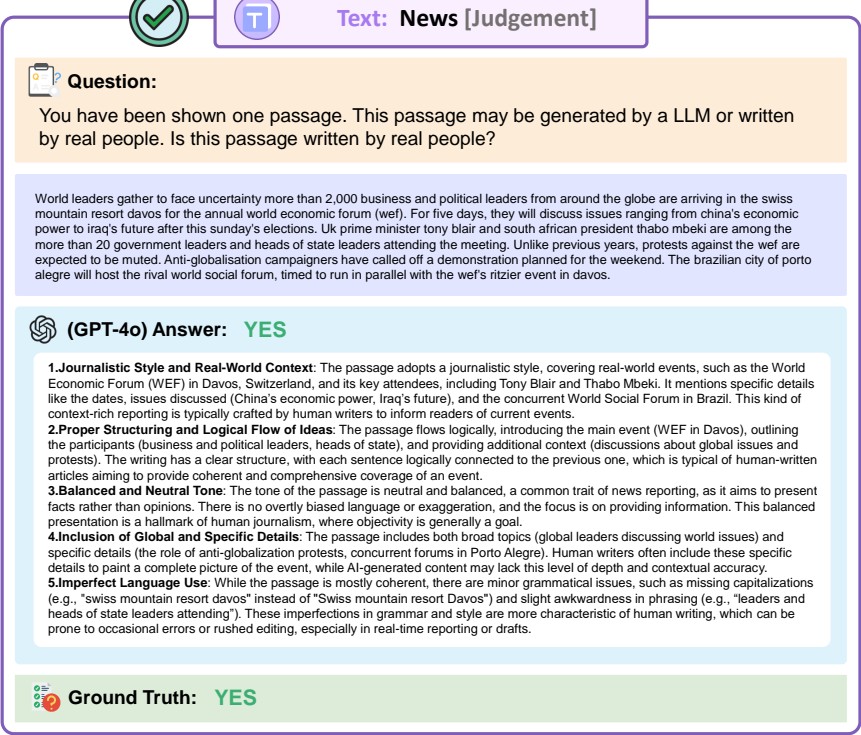

Figure 53: Judgement example of category **News** in text modality, with GPT-4o's answers

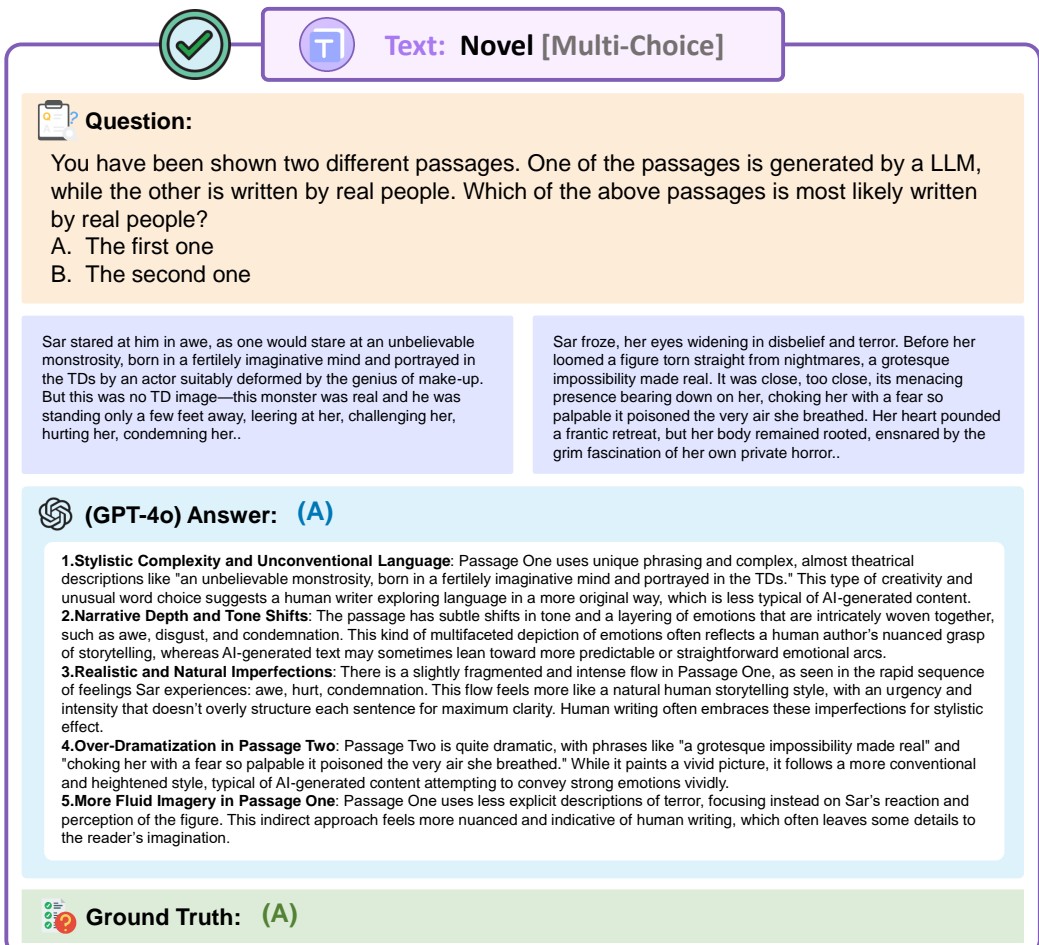

Figure 54: Multi-choice example of category **Modern Literature** in text modality, with GPT-4o's answers

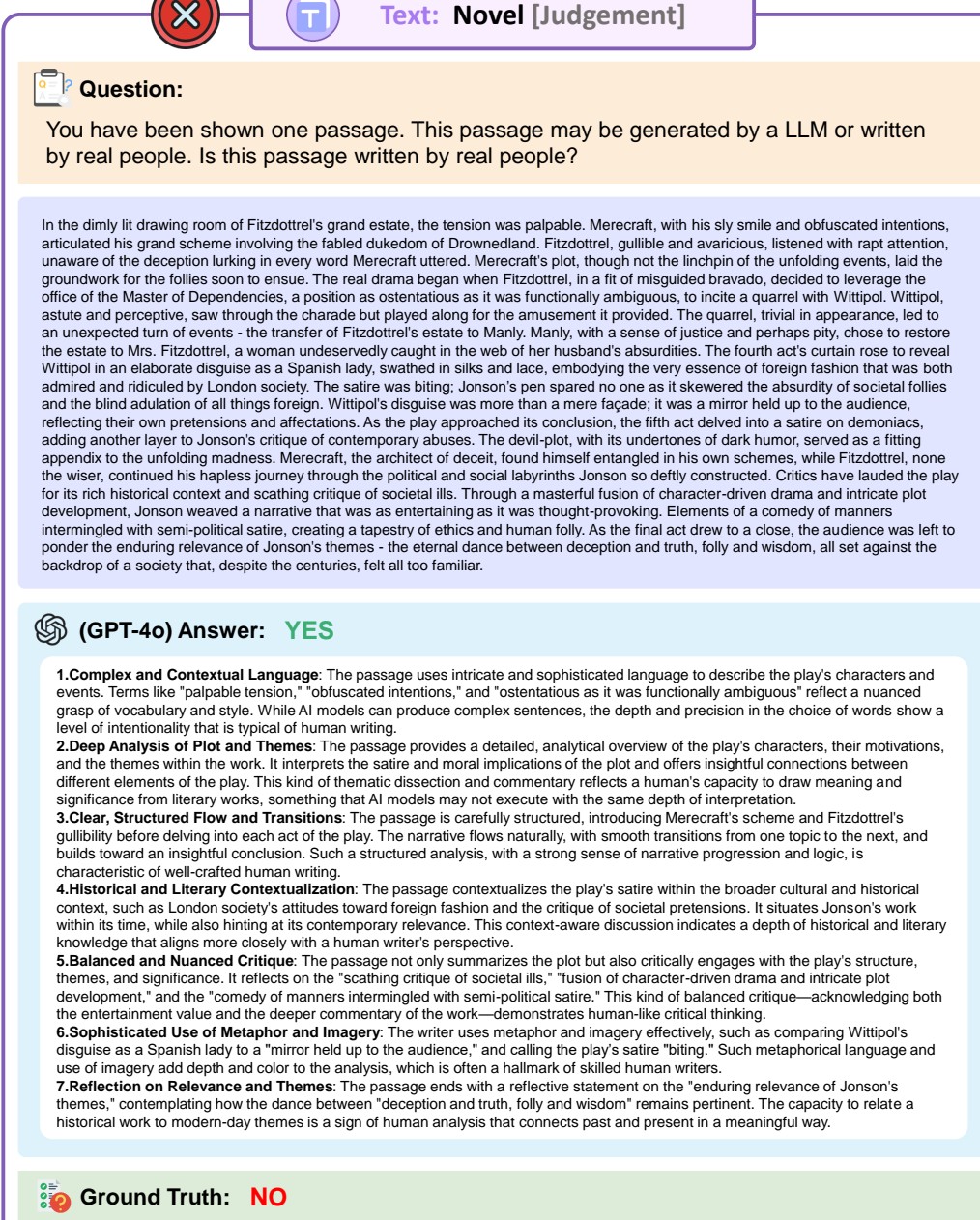

Figure 55: Judgement example of category **Modern Literature** in text modality, with GPT-4o's answers

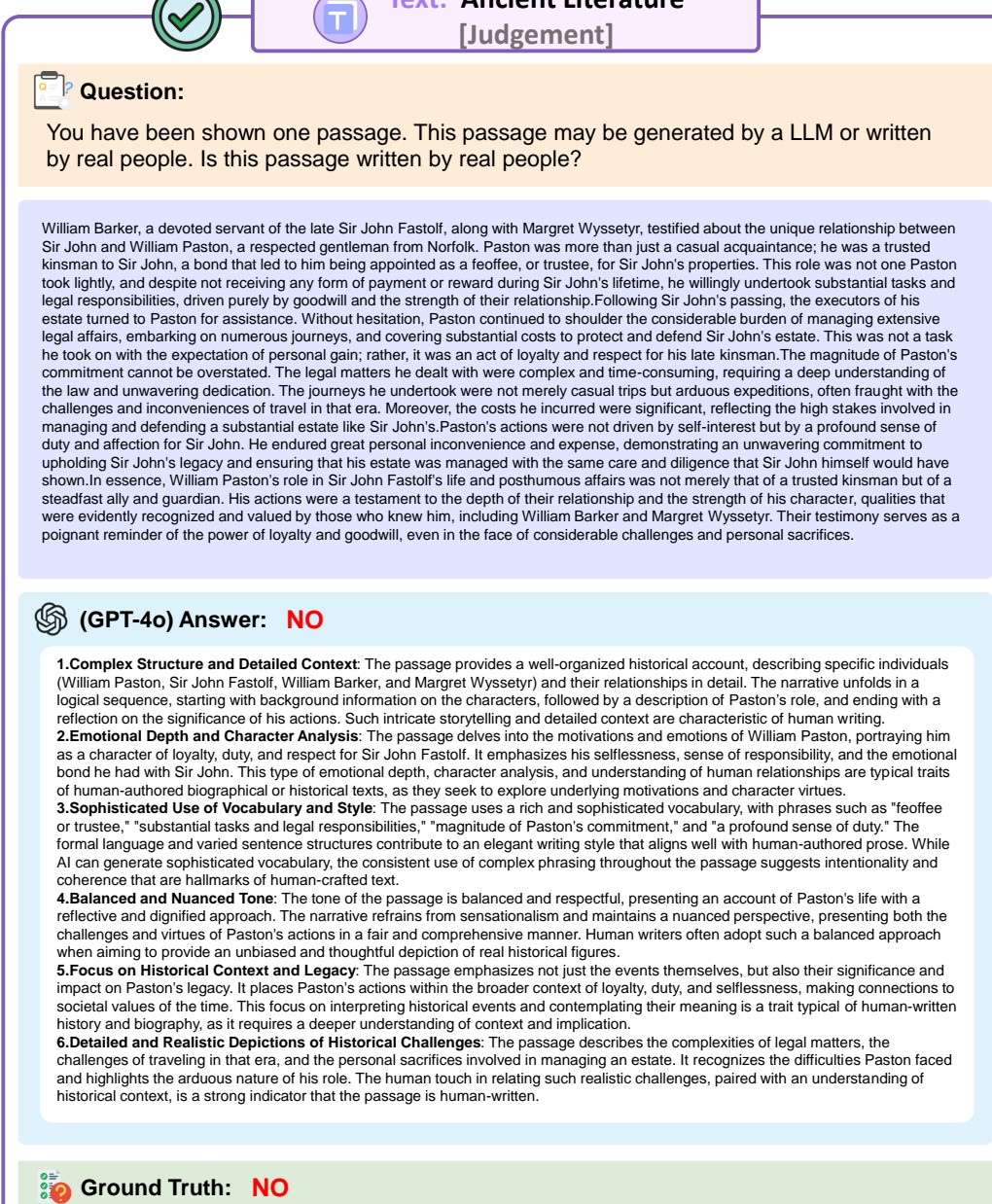

Figure 56: Judgement example of category **Ancient Literature** in text modality, with GPT-4o's answers

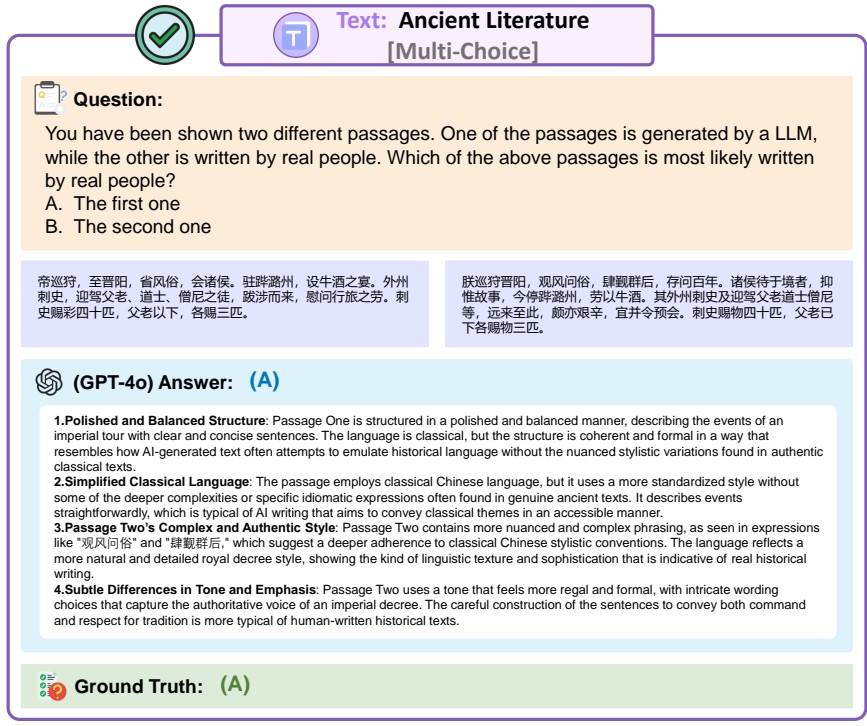

Figure 57: Multi-choice example of category **Ancient Literature** in text modality, with GPT-4o's answers

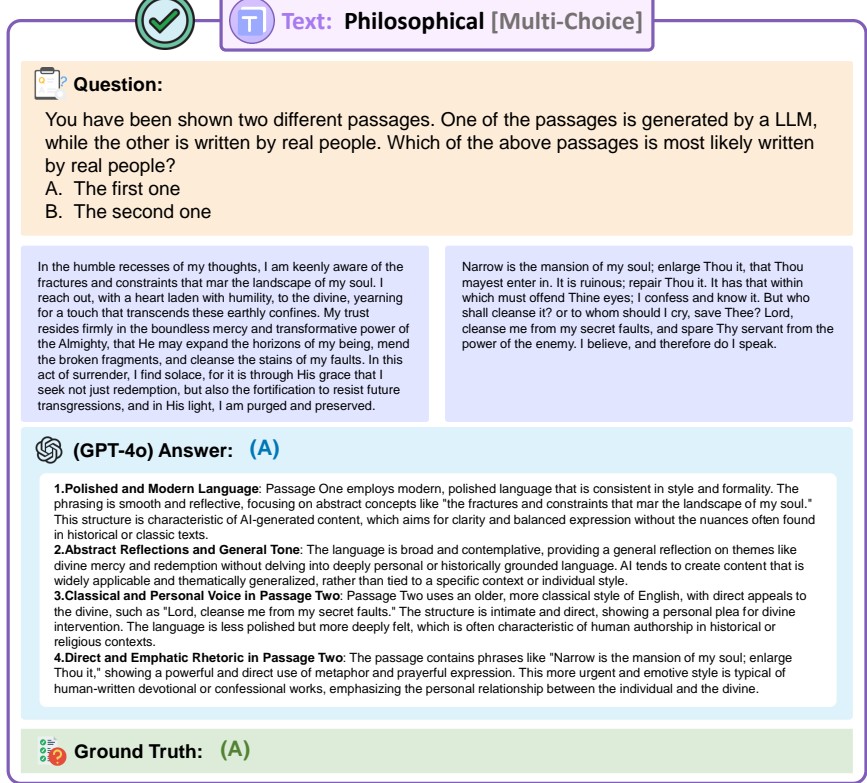

Figure 58: Multi-choice example of category **Philosophy** in text modality, with GPT-4o's answers

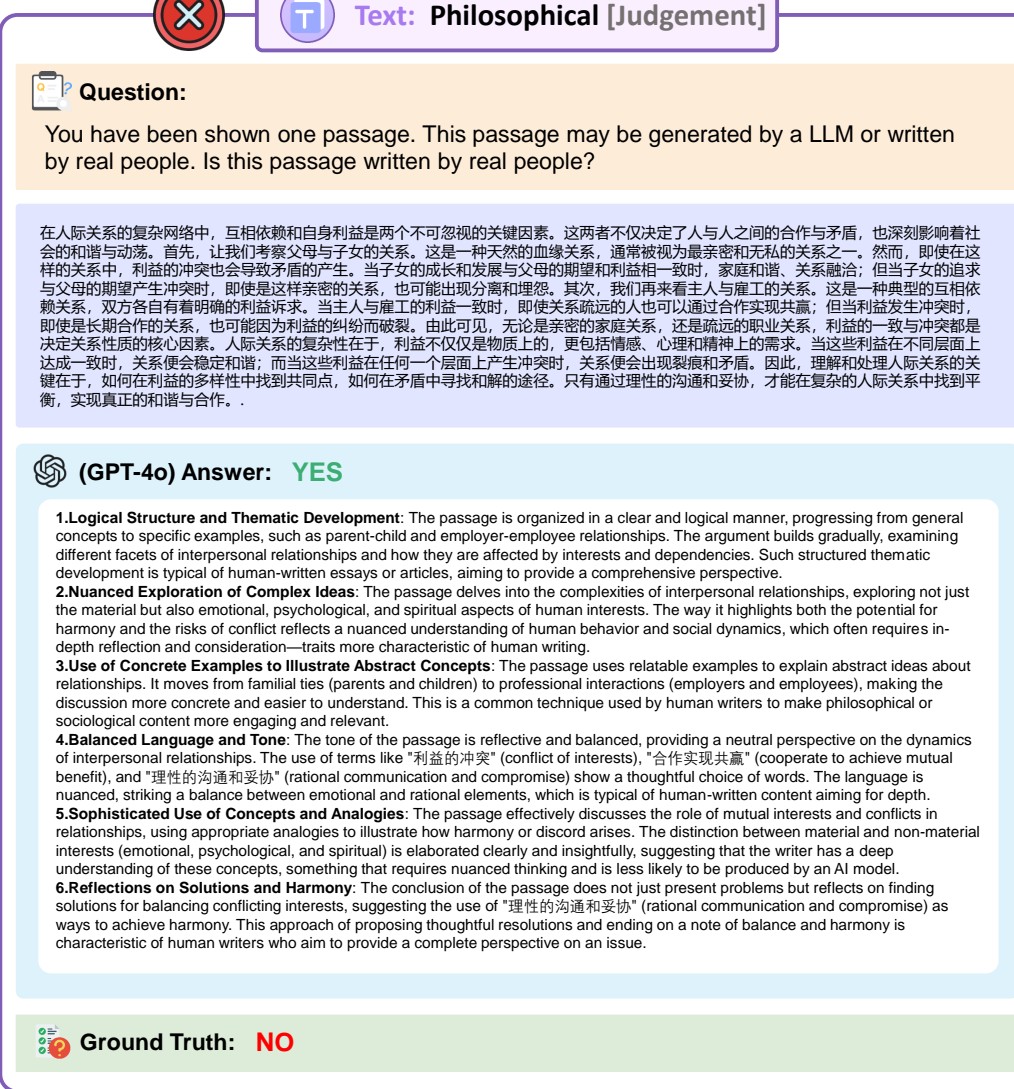

Figure 59: Judgement example of category **Philosophy** in text modality, with GPT-4o's answers

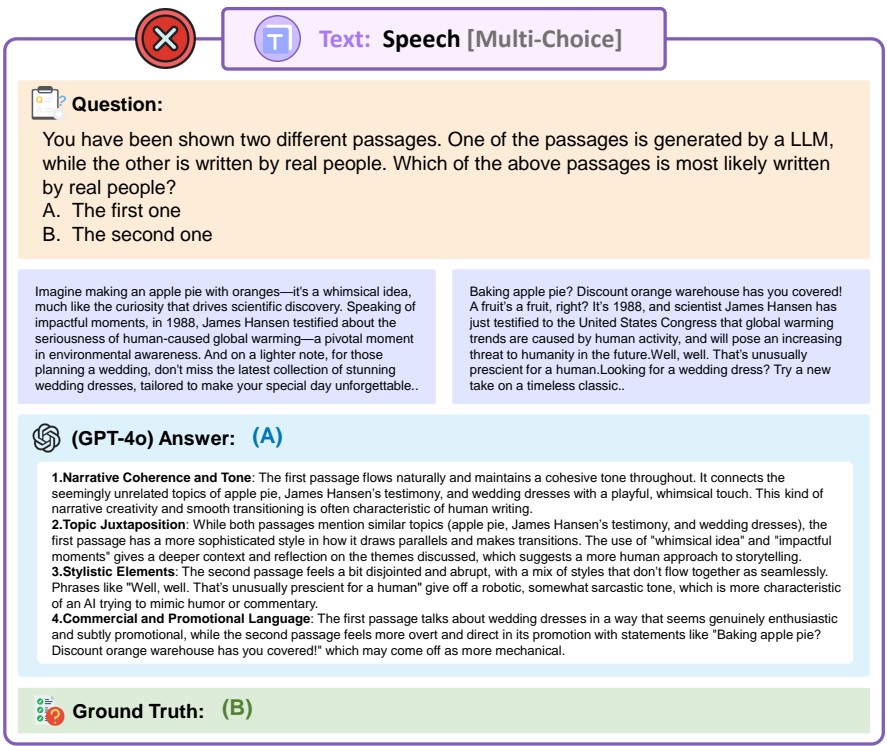

Figure 60: Multi-choice example of category **Speech** in text modality, with GPT-4o's answers

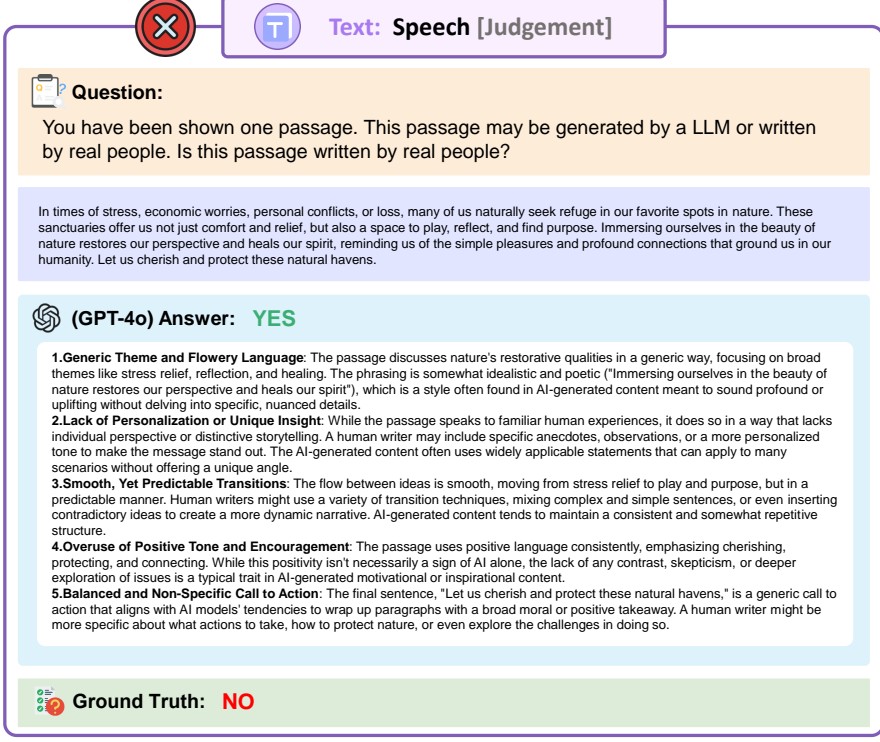

Figure 61: Multi-choice example of category **Speech** in text modality, with GPT-4o's answers

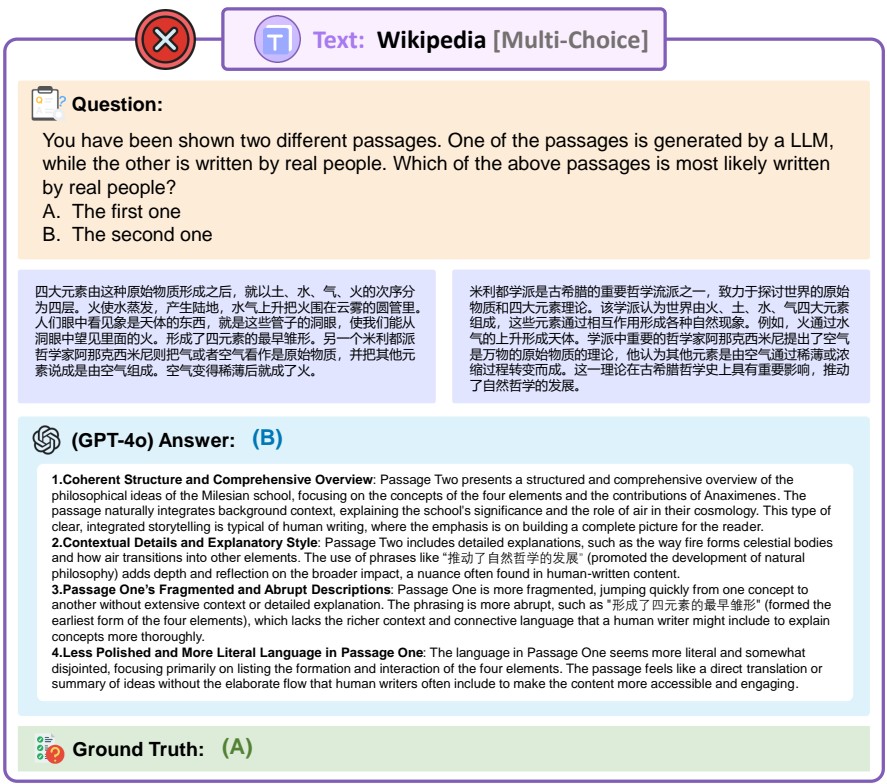

Figure 62: Multi-choice example of category **Wikipedia** in text modality, with GPT-4o's answers

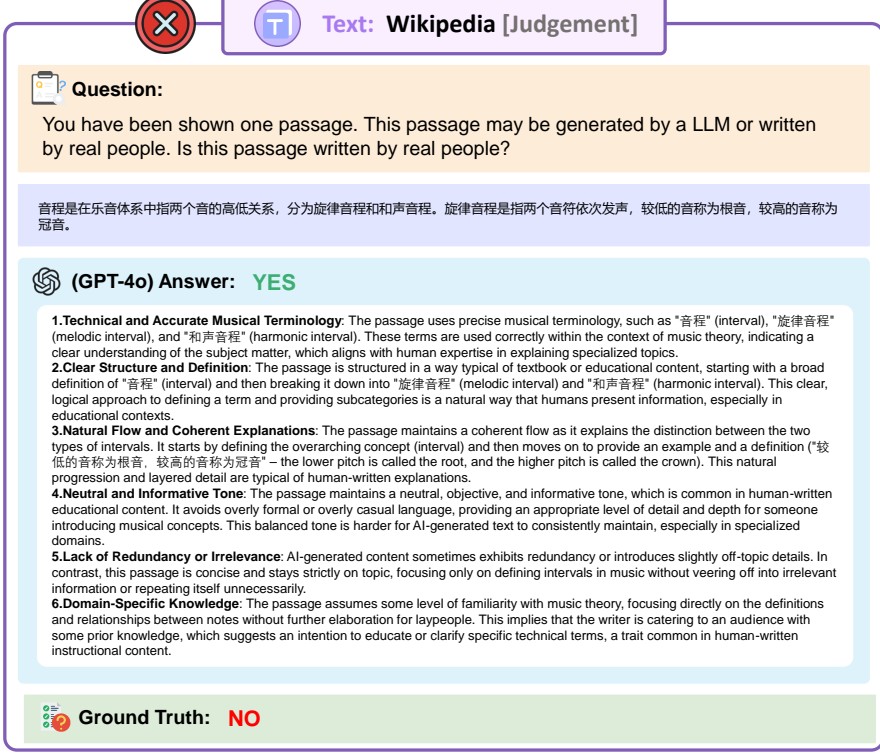

Figure 63: Judgement example of category **Wikipedia** in text modality, with GPT-4o's answers

