# OpenReview forum: "LOKI: A Comprehensive Synthetic Data Detection Benchmark using Large Multimodal Models"
_ICLR.cc/2025/Conference — ICLR 2025 Spotlight_

### Official Review · Reviewer_qo6R · 2024-10-24

**Soundness:** 3
**Presentation:** 3
**Contribution:** 4
**Rating:** 8
**Confidence:** 5

**Summary:**

The paper introduces LOKI, a comprehensive benchmark designed to evaluate the capabilities of Large Multimodal Models in detecting synthetic data across multiple modalities. Recognizing the rapid advancement of AI-generated content and the associated risks of synthetic media proliferation, the authors aim to assess how well LMMs can discern real data from AI-generated counterparts.

LOKI encompasses a diverse set of data modalities, including video, image, 3D models, text, and audio, covering 26 detailed subcategories such as satellite images, medical images, philosophical texts, and various audio types like music and environmental sounds. The benchmark includes over 18,000 carefully curated questions with varying levels of difficulty.

The tasks within LOKI are multi-faceted:

Judgment Tasks: Binary classification to determine if a piece of data is real or AI-generated.
Multiple-Choice Questions: Selecting the AI-generated item from a set of options.
Abnormal Detail Selection: Identifying specific anomalies in synthetic data.
Abnormal Explanation Tasks: Providing explanations for why data is identified as synthetic.
The authors evaluated 22 open-source LMMs and 6 closed-source models (including GPT-4 and Gemini) using LOKI. Their findings highlight that while LMMs show promise in synthetic data detection and offer interpretability advantages over traditional expert models, they also exhibit significant limitations. Models tend to have biases, lack domain-specific knowledge, and display unbalanced performance across different modalities.

**Strengths:**

- Comprehensive Multimodal Benchmark: LOKI covers an extensive range of data modalities and subcategories.
- Inclusion of specialized domains like satellite imagery, medical images, and philosophical texts pushes the boundaries of traditional datasets and tests models in less-explored areas.
- Multi-Level Task Design: The benchmark doesn't just focus on binary classification but also includes tasks that assess models' abilities to explain their reasoning, promoting the development of interpretable AI systems.
- Highlighting the Importance of Explainability
- By testing perception, knowledge, and reasoning across modalities, LOKI contributes to the broader goal of advancing towards AGI.

**Weaknesses:**

- Limited Performance in Certain Modalities: The benchmark reveals that LMMs perform poorly in modalities like 3D and audio, which may be due to the lack of available models or training data in these areas.
- Insufficient Details on Data Generation Methods: The paper could provide more in-depth information on how synthetic data was generated for each modality, which is crucial for reproducibility and understanding potential biases in the dataset.
- Evaluation of Few-Shot and Chain-of-Thought Prompting: The analysis of prompting strategies is somewhat limited.

**Questions:**

- Quality and Realism of Synthetic Data: How does the synthetic data in LOKI reflect the latest advancements in generative models? Are there measures taken to ensure that the synthetic data poses a realistic challenge for detection models?
- Prompting Strategies Implementation: Could you elaborate on how chain-of-thought and few-shot prompting were implemented across different models? Specifically, how did these strategies impact models that are less capable of handling long contexts or reasoning tasks?

---

> ### Author Response · Authors · 2024-11-23
>
> **W1:** *Limited Performance in Certain Modalities.*
>
> Thank you for your detailed feedback. The limited performance of models in certain modalities aligns with the conclusions we reached in **Section 4.2** of the paper, particularly regarding performance in 3D and audio modalities. In **Appendix D.3** and **D.4**, we provide detailed analysis of such phenomenon:
>
> - Compared to image and text modalities, existing LMMs exhibit significantly poorer performance on 3D and audio modality tasks. A key reason for this disparity is the relative scarcity of 3D and audio data in the training datasets of current LMMs.
> - Currently, there are very few LMMs that support audio or 3D modalities, and most perform poorly on synthetic detection tasks. As noted in Appendix D.3, previous studies (Liu et al., 2024b) have shown that acoustic features are more critical than the audio content itself for detecting deepfake audio. However, the architecture and training data of current audio-language models are primarily focused on content understanding rather than capturing acoustic features, which is likely a key reason for their subpar performance.
>
> **Our perspective aligns closely with your views, which is also one of our key findings. Furthermore, we would like to clarify that the poor performance of LMMs on these modalities reflects the limitations of the existing models, not a weakness of the LOKI benchmark.**
>
> ---
>
> **W2:** *Insufficient Details on Data Generation Methods.*
>
> We appreciate the reviewer’s attention to the details of the data synthesis methods. Our data collection process is described in detail in **Appendix B.1**. Our data primarily originates from online internet collections, reused from public datasets, and self-synthesized into new composite data. [**In this Table 6**](https://postimg.cc/sMtkTgz3) of the Appendix, we provide detailed descriptions of the data sources for each modality and the models used for generating the data. To ensure diversity in synthetic data, each modality incorporates more than five different synthesis methods. **Figures 9** and **10** also showcase additional examples of synthetic data. Importantly, we ensured the inclusion of the latest or SOTA models, which demonstrate outstanding performance in their respective fields, such as Video-Sora, 3D-CLAY, Text-GPT-4o, Image-FLUX, and Audio-Suno.  In **Appendix B.4**, we provide additional explanations regarding the SPECIAL DATA synthesis methods, including document images, remote sensing imagery, and medical images. We believe the detailed descriptions in the supplementary materials will facilitate a better understanding of the data synthesis methods.
>
> ---
>
> **Q1:** *Quality and Realism of Synthetic Data.*
>
> We fully understand the reviewers' concerns regarding the quality of synthetic data. Below, we analyze this aspect from four perspectives: data synthesis methods, data categories and annotations, diverse task design, and the real-world relevance of the data.
>
> **Latest Multimodal Synthetic Data Generation Methods.**
>
> As outlined in Weakness 2, we ensure the inclusion of the latest or SOTA models, all of which demonstrate exceptional performance in their respective domains. These advanced synthetic data generation methods effectively reflect the latest progress in generative models, ensuring the production of high-quality synthetic data.
>
> **Heterogeneous Categories and Multi-level Annotations.**
>
> LOKI integrates a wide range of recently popular synthetic modalities, including video, image, 3D data, text, and audio, ensuring comprehensive data coverage. The dataset spans 26 detailed categories across these modalities. Moreover, LOKI provides both basic "synthetic or real" labels and fine-grained annotations of anomalous regions.
>
> **Diverse task design to evaluate the model's capabilities.**
>
> Our task design is diverse and challenging (as detailed in **Section 3.3** of the paper), including judgment, multiple-choice questions, anomaly detection, and anomaly explanation. The LOKI dataset’s task design extends beyond traditional binary classification problems to include complex tasks requiring deep understanding and reasoning. Models must not only classify synthetic data but also identify global or local anomaly locations and generate reasonable natural language explanations.
>
> **Real-world relevance of the data.**
>
> LOKI includes heterogeneous data covering various application domains such as medical imaging, remote sensing, and natural scenes (as shown in Figure 2). This design ensures that the dataset can simulate complex real-world scenarios, making it closely aligned with practical application needs. This fine-grained design ensures that the dataset is suitable for various real-world scenarios and provides researchers with a highly realistic and challenging evaluation environment.

---

> > ### Author Response · Authors · 2024-11-23
> >
> > **W3:** *Evaluation of Few-Shot and Chain-of-Thought Prompting: The analysis of prompting strategies is somewhat limited.*
> >
> > We appreciate your reminder regarding the limited analysis of our prompting strategies. Based on our analysis, we find that the negative impact of FS prompting of LMMs may be related to the inherent difficulty of synthetic detection while CoT generally increases (neutral or better) performance for most LMMs.
> >
> > On one hand, synthetic detection requires fine-grained perception and complex reasoning to recognize artifacts, which might be out of the training scope of current LMMs. Therefore, in FS prompting, simply providing in-context examples without reasoning steps might be insufficient for most models to learn how to detect artifacts. Furthermore, adding them to their context window might interfere with their original reasoning paths, resulting in degraded performances. Therefore, in the difficult setup of synthetic detection, the effects of FS prompting on model performance are limited. We note that the strong performance of GPT-4o models after FS prompting might stem from its inherently strong reasoning abilities.
> >
> > On the other hand, consistent with findings in other CoT prompting methods ([1], [2], [3]), we find CoT generally improves model performance.  To further demonstrate the effectiveness of CoT prompting, we have included more experiments in our response to **Reviewer AhiU W3** with more modalities and task types, which also demonstrates CoT's positive impacts on most LMMs. It is worth mentioning that we have also provided additional analysis of prompting strategies in response to **Reviewer rrG5's** concerns.
> >
> > We will include these analyses and further details in the supplementary material to provide a clearer understanding of the varying effects of prompting strategies on different models.
> >
> > ---
> >
> > **Q2:** *Prompting Strategies Implementation: Could you elaborate on how chain-of-thought and few-shot prompting were implemented across different models? Specifically, how did these strategies impact models that are less capable of handling long contexts or reasoning tasks?*
> >
> > We appreciate your interest in Few-Shot and Chain-of-Thought Prompting. In the final paragraph of the experimental section in Chapter 4 of our paper, we have described the implementation of CoT and FS prompting, where we mainly follow previous works to implement our FS([1], [2]) and CoT([3], [4], [5]) prompting. To further clarify our methods, we note:
> > - **For FS Prompting**, we selected two samples (one about real data and one about AI-generated data)  that are from the same data types of the prompt question and concatenate them before it.
> > - **For CoT Prompting**, to better elicit reasoning steps, we include our artifact annotations in the prompt and prompt the model to reason step by step.
> >
> > As noted in Line 522 of our paper, we attribute LLaVA-OV's irregular behaviors to its inability to handle longer contexts. By analyzing the outputs of LLaVA-OV, we found that its performance decline under CoT strategies might be due to insufficient training on long-context scenarios. When we increased the context using CoT (providing both answers and reasoning chains in the examples), LLaVA-OV often produced irregular sentences, including irrelevant responses, premature sentence endings (directly outputting `<eos>`), and unrelated replies. This behavior indicates that the model struggles to handle extended context effectively.
> >
> > *References:*
> >
> > [1] Brown, T. B. (2020). Language models are few-shot learners. arXiv preprint arXiv:2005.14165.
> >
> > [2] Alayrac, J. B., Donahue, J., Luc, P., Miech, A., Barr, I., Hasson, Y., ... & Simonyan, K. (2022). Flamingo: a visual language model for few-shot learning. Advances in neural information processing systems, 35, 23716-23736.
> >
> > [3] Wei, J., Wang, X., Schuurmans, D., Bosma, M., Xia, F., Chi, E., ... & Zhou, D. (2022). Chain-of-thought prompting elicits reasoning in large language models. Advances in neural information processing systems, 35, 24824-24837.
> >
> > [4] Wang, X., Wei, J., Schuurmans, D., Le, Q., Chi, E., Narang, S., ... & Zhou, D. (2022). Self-consistency improves chain of thought reasoning in language models. arXiv preprint arXiv:2203.11171.
> >
> > [5] Zhang, Z., Zhang, A., Li, M., Zhao, H., Karypis, G., & Smola, A. (2023). Multimodal chain-of-thought reasoning in language models. arXiv preprint arXiv:2302.00923.

---

> > > ### Author Response · Authors · 2024-12-02
> > >
> > > Dear Reviewer qo6R,
> > >
> > > We would like to once again express our sincere gratitude for your valuable comments and feedback. As the Rebuttal period is approaching its end, we kindly remind you that if you have any further questions or concerns, please do not hesitate to reach out to us at your convenience.
> > >
> > > Thank you once again for your time and consideration.

---

### Official Review · Reviewer_AhiU · 2024-10-28

**Soundness:** 3
**Presentation:** 3
**Contribution:** 3
**Rating:** 8
**Confidence:** 4

**Summary:**

This paper introduces LOKI, a novel benchmark for evaluating large multimodal models (LMMs) in detecting synthetic data across multiple modalities, such as video, image, text, and audio.This benchmark features broad-level assessments, including judgment and multiple-choice tasks, alongside detailed anomaly selection and explanation exercises, providing an in-depth evaluation of large multimodal models (LMMs).

**Strengths:**

1.	The paper is well written and easy to follow. The authors provide sufficient technical details for readers to understand their work.
2.	The benchmark designed by the authors encompasses a rich variety of modalities and diverse question types, enabling a comprehensive evaluation of LMM performance.
3.	The authors introduce a metric called the Normalized Bias Index (NBI) to quantify the performance differences of the model on natural and AI-generated data across different modalities, which is an innovative way to assess model bias.

**Weaknesses:**

1.	The current evaluation mainly relies on accuracy and NBI; however, at low recall rates, NBI may not adequately reflect model bias. Additionally, the design of NBI may be insufficient to comprehensively capture various types of bias exhibited by the model.
2.	The paper mentions that the model exhibits "bias" across different modalities. However, the specific causes of this bias are not thoroughly explored through experiments or comparative analysis. This conclusion may be based on surface-level observations without further investigation into whether the bias arises from data, model architecture, or task design.
3.	The paper mentions that the Chain-of-Thought (CoT) approach can impact model performance in image and 3D reasoning tasks.  However, it does not provide sufficient experimental details to clarify whether CoT significantly enhances performance across all types of tasks or if it is only effective for the specific tasks currently evaluated.
4.	It is suggested to discuss and compare more related works such as [1,2] in this paper.

[1] Detecting and Grounding Multi-Modal Media Manipulation and Beyond. TPAMI 2024.

[2] Detecting and Grounding Multi-Modal Media Manipulation. CVPR 2023.

**Questions:**

1.	In the fine-grained anomaly selection task, all samples were manually reviewed to ensure question quality. However, how can the robustness of the task design be ensured in future large-scale applications?
2.	Could you provide additional insights into why Claude-3.5-Sonnet tends to misclassify synthetic images as real? For instance, does this result from limitations in the model's ability to recognize fine-grained abnormalities, or are there certain types of synthetic image features that are particularly challenging for the model?

---

> ### Author Response · Authors · 2024-11-23
>
> **W1:** *The current evaluation mainly relies on accuracy and NBI; however, at low recall rates, NBI may not adequately reflect model bias. Additionally, the design of NBI may be insufficient to comprehensively capture various types of bias exhibited by the model.*
>
> We appreciate your concerns about the reliability and comprehensiveness of the current NBI design under low recall rates. After a thorough review, we have critically reflected on the rationale behind the NBI formula proposed in Appendix C.2 of the paper, particularly its accuracy in scenarios with low recall rates.
>
> **1. Clarification of the Reasonableness of the Current NBI Formula**
>
> We would like to clarify that the current NBI formula is inherently reasonable across different recall rates, though there is room for improvement. Specifically, we provide the following mathematical discussions:
> - When $R{\text{natural}}$ and $R{\text{generated}}$ both approach 0, it indicates that the model has completely reversed the real and fake classifications. We consider this scenario highly unlikely to occur.
> - When either $R{\text{natural}}$ or $R{\text{generated}}$ approaches 0, the existing NBI formula will always result in a significant bias toward the other class, which is quite reasonable.
> - In other cases, the original NBI formula can be rewritten as:
> $$
> \text{NBI} = \frac{R_{\text{natural}} - R_{\text{generated}}}{R_{\text{natural}} + R_{\text{generated}}} = \frac{1 - \frac{R_{\text{generated}}}{R_{\text{natural}}}}{1 + \frac{R_{\text{generated}}}{R_{\text{natural}}}} \in [-1, 1]
> $$
>
> This demonstrates that the current NBI formula is not inherently inadequate in reflecting model bias under low recall rates. Instead, it fundamentally depends on the ratio of recall rates between real and generated data. We plotted the graph of the existing NBI function [**in this figure**](https://postimg.cc/bsfWKMcF).
>
> **2. Ensuring Unbiased Evaluations of LMMs**
>
> We also wish to emphasize that in evaluating LMMs, we took rigorous precautions at multiple levels (e.g., data and task design) to prevent the introduction of extraneous biases, ensuring that the observed bias originates from the LMMs themselves. Detailed clarifications are as follows:
>
> - Data Level:
>   - We clarify that the LOKI benchmark proposed in this work minimizes prior biases. As outlined in Appendix B.1 Table 6, LOKI contains diverse sources of both real and synthetic data (e.g., multiple generative models and public datasets), ensuring data diversity and mitigating the risk of single-distribution dominance.
>   - Additionally, for each modality, we ensure that real and synthetic data originated from identical domains to prevent any apparent content biases. This guarantees consistency and ensures that real and synthetic data share a comparable feature space.
>   - Finally, we aimed for balanced quantities of real and synthetic data within each modality (e.g., for images). For modalities with imbalanced data quantities (e.g., 3D or video), we manually adjusted evaluation metrics to ensure balance by separately computing results for real and synthetic data and averaging the two, thereby eliminating biases caused by data imbalance.
> - Task Design Level:
>   - We acknowledge the potential influence of different prompts on LMM responses. To eliminate biases introduced by task design, we implemented positive and negative questioning during LOKI evaluation. For every input, we posed a positive query (e.g., asking whether the input is real data) and a negative query (e.g., asking whether the input is synthetic). The final result was derived as the mean of the two queries, effectively eliminating biases arising from differences in task design.
>
> **3. Future Directions**
>
> We sincerely appreciate your valuable feedback regarding the NBI design. As you suggested, biases exhibited by models may manifest at multiple granularities. In future work, we plan to design more targeted metrics to evaluate LMMs' biases at various fine-grained levels. Additionally, we aim to conduct further instance-based analyses to explore the underlying causes of biases across different granularities.

---

> > ### Author Response · Authors · 2024-11-23
> >
> > **W2:** *The paper mentions that the model exhibits "bias" across different modalities. However, the specific causes of this bias are not thoroughly explored through experiments or comparative analysis. This conclusion may be based on surface-level observations without further investigation into whether the bias arises from data, model architecture, or task design.*
> >
> > We appreciate your observation regarding the lack of explicit analysis of the potential causes for bias exhibited by current LMMs in evaluations. While we have conducted some bias analysis for audio modality models in Appendix D.4 Line 1625, we acknowledge the need for further elaboration. Specifically, we provide additional analysis from the following three perspectives:
> >
> > **1. Data Level.**
> > Although the LOKI benchmark itself has a relatively minimal prior bias during the evaluation phase, we suspect that one potential cause of the observed bias lies in the pretraining data used by different LMMs. The pretraining datasets may inherently exhibit biases. As highlighted in some recent studies [1], the imbalance in pretraining data distribution can result in significant cultural biases. Similarly, the imbalance between real and synthetic data distributions during pretraining could be a key factor contributing to the performance difference between natural samples and generated samples during synthetic detection tasks.
> >
> > **2. Model Level.**
> > At the model level, we hypothesize that the observed biases may stem from a mismatch between current models and their mainstream training objectives, as well as the lack of targeted pretraining alignment for distinguishing real and synthetic data sources. Taking the audio modality as an example, as noted in Appendix D.4 Line 1625, existing models predominantly focus on content comprehension rather than the analysis of acoustic features. While this focus may be sufficient for identifying real data, it presents significant challenges for detecting synthetic data, where flaws are often subtle and primarily embedded in the fine details of acoustic features. This limitation underscores a broader issue across modalities: current LMMs tend to prioritize content understanding during pretraining while overlooking aspects related to data provenance and authenticity, which likely contributes to the biases observed in tasks involving real and synthetic data.
> >
> > **3. Future Work.**
> > We sincerely appreciate your valuable suggestions regarding the causes of bias in current models. In response, we will expand and refine the aforementioned analysis in the supplementary materials. Moving forward, we plan to design relevant experiments, such as leveraging open-source LMMs to validate the impact of pretraining data distribution or specific alignment tasks. These efforts aim to further investigate the origins of model bias and deepen our understanding of its underlying causes.
> >
> > *Reference:*
> >
> > [1] Bhatia, M., Ravi, S., Chinchure, A., Hwang, E. and Shwartz, V., 2024. From local concepts to universals: Evaluating the multicultural understanding of vision-language models. arXiv preprint arXiv:2407.00263.

---

> ### Author Response · Authors · 2024-11-23
>
> **W3:** *The paper mentions that the Chain-of-Thought (CoT) approach can impact model performance in image and 3D reasoning tasks. However, it does not provide sufficient experimental details to clarify whether CoT significantly enhances performance across all types of tasks or if it is only effective for the specific tasks currently evaluated.*
>
> In LOKI, we primarily focused on 3D and image modalities when exploring the impact of different prompting strategies on model performance. This focus was chosen because these modalities provide sufficiently annotated data and diverse question types. However, due to the large sample size in the video modality and the high cost of conducting comprehensive prompting tests on closed-source models, we did not include video modality in our initial experiments.
>
> To address your concern and further validate the effectiveness of CoT across various question types and modalities, we conducted additional experiments. These include testing on multiple question types within the image modality (e.g., multiple-choice and anomaly selection) and extending the evaluation to video and text modalities (e.g., binary classification tasks). The results are summarized [**in this table**](https://postimg.cc/87tQ40rV).
>
> From the results, we observe that for most models, CoT significantly improves performance across different question types and modalities. While LLaVA-OV-7B struggles under CoT settings, other models, such as InternVL2-8B, Qwen2-VL, and GPT-4o, consistently benefit from CoT prompting, with noticeable performance gains across both tasks (e.g., anomaly detection, binary classification) and modalities (e.g., image, video, text).
>
> These additional experiments strongly support the effectiveness of CoT prompting across various task types and modalities. We have included more details about the CoT implementation and additional results in Appendix E.3 (highlighted in blue).
>
> ---
>
> **W4:** *It is suggested to discuss and compare more related works such as [1,2] in this paper.*
>
> We appreciate the reviewer’s suggestion and acknowledge the important contributions of [1] and [2] in advancing the detection and grounding of multi-modal media manipulation. These works tackle a critical problem by not only detecting manipulated media but also localizing manipulated content, which provides valuable insights for reasoning across visual and textual modalities. Similar to [1] and [2], our work also aims to push the boundaries of model performance in detecting artificial data and emphasize the importance of reasoning in multi-modal tasks.
>
> However, our work differs in its broader scope. While [1] and [2] concentrate on manipulation detection and grounding, LOKI focuses on synthetic data detection across a wider range of modalities, including video, audio, and 3D. Additionally, LOKI also introduces tasks such as anomaly selection and natural language explanations, allowing for a comprehensive evaluation of LMMs' reasoning and explainability. Once again, we would like to thank the reviewer for the reminder. We have included the citations in the revised version to confirm the relevance of [1] and [2] (highlighted in blue).
>
> ---
>
> **Q2:** *Could you provide additional insights into why Claude-3.5-Sonnet tends to misclassify synthetic images as real? For instance, does this result from limitations in the model's ability to recognize fine-grained abnormalities, or are there certain types of synthetic image features that are particularly challenging for the model?*
>
> The anomalous performance of Claude-3.5-Sonnet has indeed also drawn our attention. In the Abnormal Explanation experiment, we indeed observed a particular behavior from Claude. Specifically, in the image modality, when the model was prompted to explain anomalous phenomena (as illustrated in Figure 12), Claude frequently (98.3% of the time) outright rejected the premise of the task. A typical response would be: *"Apologies, but I must disagree with your premise. Upon careful examination, this image does not exhibit any features indicative of being artificially generated or forged."*
>
> It is worth noting that this issue was independent of the type of input image or the nature of the anomaly. We suspect that this behavior arises from biases introduced during Claude’s post-training for safety alignment [1]. To prioritize "safety," Claude appears to avoid making definitive judgments in tasks involving AI-generated content, opting instead to decline participation as a risk-avoidance mechanism. While this approach ensures "safe" responses, it significantly undermines the model’s performance in this task.  However, this limitation does not appear to be triggered in other modalities. We would like to note that, since we cannot bypass Claude's safety constraints, the performance of Claude on such tasks is likely underestimated in this context.
>
> *Reference:*
>
> [1] Anthropic. Claude 3.5 Sonnet URL: https://www.anthropic.com/news/claude-3-5-sonnet

---

> > ### Author Response · Authors · 2024-11-23
> >
> > **Q1:** *In the fine-grained anomaly selection task, all samples were manually reviewed to ensure question quality. However, how can the robustness of the task design be ensured in future large-scale applications?*
> >
> > We appreciate the reviewers’ attention to the robustness of fine-grained anomaly selection task design for future large-scale applications. Ensuring annotation quality in large-scale applications indeed presents significant challenges. As highlighted in a related study [1], maintaining high-quality and consistent annotations in fine-grained tasks is a complex and time-consuming process. In the LOKI dataset, we prioritized annotation quality over scale by adopting a fully manual review process to ensure data quality. However, we also recognize that solely relying on manual annotations may face efficiency and cost constraints in large-scale future applications.
> >
> > To address this challenge, we propose several potential strategies to improve robustness while scaling up:
> > 1. **Large-model-assisted semi-automatic annotation:** Large language models can directly generate preliminary annotation results, such as feature descriptions of anomalous samples or suggestions for region delineation. These annotations can then be reviewed and refined by human annotators. Feedback from annotators during this process can also be used to continuously optimize the model’s prompts, thereby improving its performance in annotation tasks.This generative-assisted approach not only significantly reduces the workload of manual annotation but also lays the foundation for handling more complex annotation tasks and ensuring the robustness in the future.
> > 2. **Expert model-based annotation:** Training expert models based on manually annotated data can effectively enhance annotation efficiency. High-quality initial data can be generated through manual annotation and then used to train expert models (e.g., SAM [2] and Florence [3]). These models can automatically annotate anomaly regions or generate multimodal anomaly descriptions. The model-generated annotations can be reviewed and refined by human annotators, and the improved results can be used to further train the model. This iterative process gradually enhances the model's ability for specific annotation tasks, reducing costs and increasing efficiency.
> > 3. **Crowdsourcing for enhanced annotation efficiency:** For large-scale tasks, crowdsourcing can efficiently execute complex annotation tasks through reasonable task decomposition. Annotation tasks can be divided into smaller, fine-grained tasks and distributed to crowdsourcing participants for preliminary annotation. Reliability checks, such as embedding validation samples or performing consistency checks, can then filter out high-quality results. Low-quality or inconsistent annotations can be redistributed for further refinement. For critical samples, annotations can be reviewed by domain experts to ensure the final results meet high-quality and consistency standards.
> >
> > With these strategies, we aim to further optimize task design robustness, reduce manual costs, and maintain the high-quality annotation standards of the LOKI dataset in future large-scale applications. We look forward to evaluating the practical effects of these strategies in future iterations.
> >
> > *Reference:*
> >
> > [1]Liang, Y., He, J., Li, G., Li, P., Klimovskiy, A., Carolan, N., ... & Navalpakkam, V. (2024). Rich human feedback for text-to-image generation. Proceedings of the IEEE/CVF Conference on Computer Vision and Pattern Recognition (CVPR), 19401–19411
> >
> > [2]Kirillov, A., Mintun, E., Ravi, N., Mao, H., Rolland, C., Gustafson, L., Xiao, T., Whitehead, S., Berg, A. C., Lo, W.-Y., Dollar, P., & Girshick, R. (2023). Segment anything. In Proceedings of the IEEE/CVF International Conference on Computer Vision (ICCV) (pp. 4015–4026).
> >
> > [3]Xiao, B., Wu, H., Xu, W., Dai, X., Hu, H., Lu, Y., Zeng, M., Liu, C., & Yuan, L. (2024). Florence-2: Advancing a unified representation for a variety of vision tasks. Proceedings of the IEEE/CVF Conference on Computer Vision and Pattern Recognition (CVPR), 4818–4829.

---

> ### Comment · Reviewer_AhiU · 2024-11-26
>
> The author has addressed most of the concerns. The reviewer increases the score and is inclined towards acceptance of the paper.

---

> > ### Author Response · Authors · 2024-11-26
> >
> > Great! Thank you for your valuable suggestions. Your feedback has helped us further improve our work. We are sincerely grateful for the time and energy you dedicated to reviewing our manuscript.

---

### Official Review · Reviewer_rrG5 · 2024-11-04

**Soundness:** 3
**Presentation:** 4
**Contribution:** 3
**Rating:** 8
**Confidence:** 4

**Summary:**

The authors introduce  LOKI, a novel benchmark designed to evaluate the ability of LMMs to detect synthetic data across multiple modalities. With the concurrent preponderance of synthetic data and the rise of powerful LLMs, the authors aim to address the emerging research topic of LMM evaluation for synthetic data detection. LLMs can provide reasoning behind authenticity judgments, benefitting explainability. The focus of LOKI is to evaluate the performance of LLMs on synthetic data detection tasks.

In particular, LOKI is aimed at addressing several shortcomings present in extant synthetic data evaluation datasets, including emphasizing human interpretability and multimodality. Overall, LOKI provides key improvements to synthetic data detection, including: diverse modalities, heterogenous categories, multi-level annotations and multimodal synthetic evaluation. framework.

**Strengths:**

The paper is well-written and easy to follow.

This work addresses a glaring and emergent need in a topic field: synthetic data detection for LLMs; I agree with the authors that there doesn't currently exist a comprehensive, multi-modal, nuanced dataset including explainability assessment for this domain area.

Extensive examples and case studies provided in appendices.

Many data domains are covered in this benchmark, including several categories and characteristics that are often underrepresented in synthetic data detection (e.g., satellite images, "abnormal details").

**Weaknesses:**

While the differentiated modalities, categories and annotation levels are beneficial, the overall size of the dataset actually seems relatively small vis-a-vis related datasets (Table 1).

It is unclear to me, how a user can methodically compare scores for different models across tasks/categories (e.g., in Table 2); perhaps the authors can address this, given the heterogenous and imbalanced nature of the data modalities and tasks, as well as the problem/domain "difficulty".

As deepfake detection is one of the most prominent synthetic data detection categories today, I believe the benchmark would benefit from its inclusion.

**Questions:**

Do you have a sense as to why the prompting strategies (FS, CoT) in general had a poor/neutral impact for most of the model/data pairs (Table 5)?

---

> ### Author Response · Authors · 2024-11-23
>
> **W1:** *While the differentiated modalities, categories and annotation levels are beneficial, the overall size of the dataset actually seems relatively small vis-a-vis related datasets (Table 1).*
>
> Thank you for your concern regarding the scale of our dataset. Table 1 provides a detailed comparison of LOKI with existing datasets, including traditional synthetic detection benchmarks and those tailored for evaluating LMMs.
>
> 1. Traditional synthetic detection datasets, such as Fake2M (Lu et al., 2023b) and HC3 (Guo et al., 2023), are primarily designed for training expert detection models. As such, their reported sizes include both training and testing data.  In contrast, LOKI is specifically designed to test the capabilities of multimodal large models (LMMs), and its reported size exclusively represents the test set. Furthermore, these traditional datasets often use binary ground truth labels , eliminating the need for fine-grained annotation of anomalous regions as required in our dataset, thus facilitating the construction of larger-scale datasets.
> 2. Compared to other LMMs' synthetic detection benchmarks like FakeBench (Li et al., 2024b)(6K) and VANE (Bharadwaj et al., 2024)(0.9K) in the middle section of Table 1, LOKI has a clear quantitative advantage in size (18k) , which further demonstrates our advantages as a robust benchmark for evaluating synthetic data detection across a variety of scenarios.
>
> In addition, the table below further provides a comparison between LOKI and other widely-used benchmarks for evaluating LMMs, including MMMU (Yue et al., 2024) (11.5K), MME[1] (2.37K), and MMBench[2] (6.67K).
>
> | Dataset  | MMMU  | MME   | MMbench | Fakebench | VANE  | Ours  |
> |----------|-------|-------|---------|-----------|-------|-------|
> | Size     | 11.5k | 2.37k | 6.67k   | 6k        | 0.9k  | 18k   |
>
> In summary, we believe LOKI’s scale is sufficient to reveal model strengths and limitations, particularly in detecting multimodal synthetic data. Thank you for raising your concerns.
>
> *References:*
>
> [1] C. Fu et al., "MME: A Comprehensive Evaluation Benchmark for Multimodal Large Language Models," arXiv preprint arXiv:2306.13394, 2024. [Online]. Available: https://arxiv.org/abs/2306.13394.
>
> [2] Liu, Y., et al. (2025). MMBench: Is Your Multi-modal Model an All-Around Player? In Leonardis, A., Ricci, E., Roth, S., Russakovsky, O., Sattler, T., & Varol, G. (Eds.), Computer Vision – ECCV 2024. Lecture Notes in Computer Science (Vol. 15064).

---

> ### Author Response · Authors · 2024-11-23
>
> **W2:** *It is unclear to me, how a user can methodically compare scores for different models across tasks/categories (e.g., in Table 2); perhaps the authors can address this, given the heterogenous and imbalanced nature of the data modalities and tasks, as well as the problem/domain "difficulty".*
>
> Since LOKI involves a large number of modalities and diverse task types, with each modality further encompassing heterogeneous data, methodically comparing scores for different models across tasks/categories is indeed a significant challenge. We sincerely appreciate the reviewers' concerns on this issue. Below, we provide additional elaboration based on existing metrics, potential future improvements, and detailed supplementary results.
>
> 1. As detailed in Appendix C.2, average accuracy is the primary metric for judgment, multiple-choice, and abnormal detail selection tasks. Table 2 in the main text evaluates the performance of large models across multiple modalities and tasks using average accuracy as the unified metric. This evaluation approach follows the metric design used in previous comprehensive benchmarks, such as MMMU and MMT-Bench. For instance, MMMU assesses capabilities across diverse disciplines and tasks, while MMT-Bench evaluates over 150 different subtasks. Both benchmarks calculate accuracy based on whether model responses align with the ground truths. This unified metric design ensures consistent evaluation across tasks, avoids task-specific metrics, and provides a standardized basis for comparison, which is a common strategy for assessing the multi-task and general capabilities of large multimodal models.
> 2. The reviewers’ suggestion to consider the “difficulty” of different problems or modalities as a factor for balancing model score comparisons is highly insightful. In Table 4, we have highlighted the evaluation performance across different modalities at varying difficulty levels. Additionally, we have supplemented the results with more tasks across finer-grained difficulty levels, as shown [**in this Table**](https://postimg.cc/hJsFC3zB) , and incorporated these results into the revised Appendix E.4 (highlighted in blue). As part of our future plans, we also intend to explore difficulty-weighted scoring for models to further improve the evaluation of comprehensive model capabilities and provide a more detailed summary of their strengths and weaknesses.
> 3. It is worth noting that, in addition to the more general overall evaluation presented in Table 2, we have provided more fine-grained evaluation results in supplementary materials. These results detail model performance under different tasks and modality divisions, enabling a more comprehensive assessment of model capabilities:
>   - Video Modality: Tables 7 and 8 respectively present the model's performance in judgment tasks and multiple-choice tasks under different video synthesis methods.
>   - Image Modality: Tables 10 and 11 illustrate the model's performance in selection and judgment tasks across different types of images.
>   - 3D Modality: Tables 13 and 15 evaluate the model's performance under different 3D synthesis methods (e.g., NeRF and GS) and point cloud input formats.
>   - Audio Modality: Table 16 demonstrates the model's performance across various audio types.
>   - Text Modality: Tables 17 through 19 assess the model's capabilities across different topics, text lengths, and both Chinese and English scenarios.

---

> ### Author Response · Authors · 2024-11-23
>
> **W3:** *As deepfake detection is one of the most prominent synthetic data detection categories today, I believe the benchmark would benefit from its inclusion.*
>
> We appreciate the reviewer's constructive suggestion regarding the inclusion of the deepfake detection category[1][2]. We would like to clarify that our benchmark LOKI already includes partial deepfake datasets from audio and image modalities. However, we did not perform a standalone analysis for the Deepfake category. The existing data for the Deepfake category includes:
> - **Audio Modality**: In audio tasks, some samples of speech and singing are sourced from their respective audio deepfake datasets ASVSpoof2019 and DCASE2023 Track 7，as mentioned in Table 6. These datasets are built on state-of-the-art voice cloning techniques, such as Text-to-Speech[3], Voice Conversion[4], and Singing Voice Conversion [5]. These technologies address major challenges in deepfake detection, including vulnerabilities in voice authentication systems and the misuse of synthetic voice technology to generate counterfeit content, such as fake artists.
> - **Image Modality**: As shown in Table 6, our image data includes a portion of facial deepfake images, comprising partial datasets from FFHQ and Deepfakeface. These data are directly aligned with the core challenges in image-based deepfake detection, ensuring that our benchmark comprehensively evaluates models on this critical aspect of synthetic data detection.
>
> In response to your suggestion, we have independently evaluated the deepfake-related components of our current dataset and presented the results below. From the results, it is evident that most models perform significantly worse on deepfake data compared to their overall accuracy. This discrepancy likely arises because the overall evaluation task includes simpler synthetic samples (e.g., basic synthetic data), which are easier for the models to distinguish, thereby inflating their overall accuracy. In contrast, deepfake data often involves more nuanced and fine-grained forgery features that are particularly challenging for general-purpose multimodal pretrained models to capture effectively. Interestingly, GPT-4o achieves the best performance on deepfake data among all evaluated models, demonstrating its relatively stronger ability to handle such complex forgeries.
>
> | Model Name                         | Deepfake Accuracy | Overall Accuracy |
> |------------------------------------|-------------------|------------------|
> | Claude-3-5-sonnet         | 0.14              | 0.54             |
> | InternVL2-8B                       | 0.13              | 0.50             |
> | VILA1.5-40B                       | 0.29              | 0.49             |
> | Llava-OV-7B        | 0.13              | 0.50             |
> | Mantis-8B            | 0.53              | 0.55             |
> | Qwen2-VL-72B             | 0.34              | 0.53             |
> | Llama3-8B      | 0.47              | 0.50             |
> | GPT-4o               | 0.59              | 0.63             |
>
> To emphasize the importance of deepfake data, we have established it as a separate detection category and included detailed model results in Appendix E.2 (highlighted in blue). We sincerely thank the reviewers once again for their constructive feedback and reminders.
>
> *References:*
>
> [1] Wang, T., Liao, X., Chow, K. P., Lin, X., & Wang, Y. (2024). Deepfake Detection: A Comprehensive Survey from the Reliability Perspective. ACM Computing Surveys, 57(3), 1–35. https://doi.org/10.1145/3699710
>
> [2] Pei, G., Zhang, J., Hu, M., Zhang, Z., Wang, C., Wu, Y., Zhai, G., Yang, J., Shen, C., & Tao, D. (2024). Deepfake Generation and Detection: A Benchmark and Survey. arXiv. https://arxiv.org/abs/2403.17881
>
> [3] Ren, Y., Ruan, Y., Tan, X., Qin, T., Zhao, S., Zhao, Z., & Liu, T. Y. (2019). Fastspeech: Fast, robust and controllable text to speech. Advances in xinkneural information processing systems, 32.
>
> [4] Sisman, B., Yamagishi, J., King, S., & Li, H. (2020). An overview of voice conversion and its challenges: From statistical modeling to deep learning. IEEE/ACM Transactions on Audio, Speech, and Language Processing, 29, 132-157.
>
> [5] Nachmani, E., & Wolf, L. (2019). Unsupervised singing voice conversion. arXiv preprint arXiv:1904.06590.

---

> > ### Author Response · Authors · 2024-11-23
> >
> > **Q1:** *Do you have a sense as to why the prompting strategies (FS, CoT) in general had a poor/neutral impact for most of the model/data pairs (Table 5)?*
> >
> > Thank you for raising this important question. We acknowledge that the impact of prompting strategies (FS[4][5], CoT[1][2]) on synthetic detection tasks varies significantly across models and scenarios, and we believe this is a critical aspect to analyze further. Based on our findings, the effects are closely tied to the complexity of synthetic detection tasks and the specific capabilities of individual models.
> >
> > **Few-shot Prompting**
> >
> > Few-shot (FS) prompting generally shows a negative impact on most models (e.g., InternVL2-8B: 49.6 → 46.1, Gemini-1.5-pro: 47.9 → 41.2). We believe this is due to:
> > 1. **Task Complexity**: Synthetic detection often requires fine-grained perception and reasoning to identify subtle artifacts. FS prompting, which provides in-context examples without reasoning steps, may be insufficient for models to learn these intricate patterns.
> > 2. **Context Interference**: Adding in-context examples can increase the context window and potentially interfere with the model's reasoning pathways. For instance, InternVL2-8B experiences a performance drop under FS prompting, which may reflect this challenge.
> >
> > **Chain-of-Thought Prompting**
> >
> > Recent studies ([1], [2], [3]) have shown that Chain-of-Thought (CoT) prompting enhances model performance by promoting step-by-step reasoning, which is especially beneficial for tasks requiring complex reasoning. Our findings are consistent with these observations. For example:
> >
> > Qwen2-VL-7B improves from 56.8 (Baseline) to 59.5 (CoT), demonstrating CoT's utility in refining reasoning.
> >
> > GPT-4o maintains robust performance under CoT (74.2), showing its capability to leverage extended reasoning pathways effectively.
> >
> > To further substantiate CoT’s effectiveness, we have included additional experiments in our response to **Reviewer AhiU**, which explore CoT prompting across more modalities and task types. These results consistently confirm its positive impact on most LMMs. However, not all models benefit from CoT. For example, LLaVA-OV-7B shows a significant performance drop (56.6 → 18.8). Upon closer examination, we found that LLaVA-OV-7B struggles with long-context reasoning, as evidenced by irregular outputs such as incomplete sentences, irrelevant responses, or prematurely ending sentences with `<eos>`. This suggests limitations in its training for handling extended contexts, which may hinder its ability to utilize CoT effectively.
> >
> > We will include these analyses and further details in the supplementary material to provide a clearer understanding of the varying effects of prompting strategies on different models.
> >
> > *References:*
> >
> > [1] Wei, J., Wang, X., Schuurmans, D., Bosma, M., Ichter, B., Xia, F., Chi, E., Le, Q. V., & Zhou, D. (2022). Chain-of-Thought Prompting Elicits Reasoning in Large Language Models.
> >
> > [2] Besta, M., Blach, N., Kubicek, A., Gerstenberger, R., Podstawski, M., Gianinazzi, L., Gajda, J., Lehmann, T., Niewiadomski, H., Nyczyk, P., & Hoefler, T. (2024). Graph of Thoughts: Solving Elaborate Problems with Large Language Models.
> >
> > [3] Chen, W., Ma, X., Wang, X., & Cohen, W. W. (2023). Program of Thoughts Prompting: Disentangling Computation from Reasoning for Numerical Reasoning Tasks.
> >
> > [4] Brown, T. B. (2020). Language models are few-shot learners. arXiv preprint arXiv:2005.14165.
> >
> > [5] Alayrac, J. B., Donahue, J., Luc, P., Miech, A., Barr, I., Hasson, Y., ... & Simonyan, K. (2022). Flamingo: a visual language model for few-shot learning. Advances in neural information processing systems, 35, 23716-23736.

---

> > > ### Author Response · Authors · 2024-12-02
> > >
> > > Dear Reviewer rrG5,
> > >
> > > We would like to once again express our sincere gratitude for your valuable comments and feedback. As the Rebuttal period is approaching its end, we kindly remind you that if you have any further questions or concerns, please do not hesitate to reach out to us at your convenience.
> > >
> > > Thank you once again for your time and consideration.

---

### Official Review · Reviewer_xU39 · 2024-11-04

**Soundness:** 3
**Presentation:** 3
**Contribution:** 3
**Rating:** 8
**Confidence:** 5

**Summary:**

The paper introduces a novel benchmark designed to evaluate the capability of LMMs in detecting synthetic data across multiple modalities, including video, image, 3D, text, and audio. LOKI is structured to provide diverse modalities, cover 26 detailed categories, and offer multi-level annotations, enabling tasks that range from basic authenticity judgments to fine-grained anomaly selection and explanation. The benchmark consists of 18,000 questions across various levels of difficulty, allowing for a comprehensive analysis of LMMs in detecting synthetic content. Additionally, the paper includes evaluations of 22 open-source and 6 closed-source LMMs, revealing both the potential and current limitations of these models in terms of detection accuracy and interpretability.

**Strengths:**

+ LOKI is a novel, multimodal dataset.

+ The paper is easy to read and well-organised.

+ Comprehensive Evaluation and Validation.

+ Curates a diverse dataset with 18,000 questions across five modalities and 26 categories, providing a solid foundation for synthetic data detection evaluation.

+ Detailed Annotations.

+ Directly addresses the challenges of synthetic data proliferation, impacting security, misinformation, and content authenticity.

+ LOKI’s findings on LMM strengths and weaknesses have the potential to drive advancements in synthetic data detection and multimodal model development.

**Weaknesses:**

-  The benchmark lacks a robustness test against common real-world conditions like compression artifacts. To enhance real-world applicability, the authors could include performance evaluations on compressed data.

**Questions:**

No follow-up questions.

---

> ### Author Response · Authors · 2024-11-23
>
> **W:** *The benchmark lacks a robustness test against common real-world conditions like compression artifacts. To enhance real-world applicability, the authors could include performance evaluations on compressed data.*
>
> Thank you for your valuable suggestion. We completely agree that incorporating robustness tests under real-world scenarios, such as compression artifacts, is essential for better evaluating the applicability of multimodal large models (LMMs) in practical settings. Following previous studies[1][2]，we conducted JPEG compression tests on judgment tasks under the image modality, based on three representative LMMs and an expert model AIDE，as previously introduced in the main text.
>
> |          | 100%  | 95%   | 90%   | 80%   | 60%   |
> |----------|--------|-------|-------|-------|-------|
> | InternVL2-8B | 56.6% | 56.1% | 56.1% | 56.4% | 56.6% |
> | Qwen2-VL-7B  | 54.5% | 54.0% | 54.2% | 54.6% | 54.1% |
> | GPT-4o       | 63.4% | 63.2% | 63.1% | 62.8% | 63.1% |
> | AIDE         | 62.8% | 46.4% | 45.0% | 44.2% | 45.2% |
>
> From these results, we observe that expert models AIDE[3] are more significantly impacted by compression artifacts. In contrast, LMMs demonstrate remarkable robustness, with compression levels having little to no effect on their detection capabilities. This distinction highlights the inherent advantages of LMMs over traditional expert models. There are several possible factors that contribute to the robustness of LMMs to compressed images.
>
> 1. **Extensive Pretraining on Diverse Data**: The vast amount of data available on the internet includes images with varying levels of compression. During pretraining, LMMs inherently learn features that are robust to such artifacts by including the internet-scale data, as demonstrated in works like UniversalFakeDetect[2] and CLIPping[4] .
> 2. **Model Scale Advantage**: Unlike traditional expert modules, which typically operate with parameter counts in the millions, LMMs are equipped with parameters in the billions. This scale significantly enhances their generalization and robustness, consistent with the emergent properties observed in large-scale models.
>
> This finding further underscores the potential of LMMs for real-world applications, particularly in scenarios involving compression artifacts. We have included these experimental results and compression robustness analysis (highlighted in blue) in Appendix E.1, to provide a comprehensive understanding of the reliability and applicability of LMM.
> If you have additional questions, we welcome further discussion.
>
>
> *References:*
>
> [1] Wang, S. Y., Wang, O., Zhang, R., Owens, A., & Efros, A. A. (2020). CNN-generated images are surprisingly easy to spot... for now. In Proceedings of the IEEE/CVF conference on computer vision and pattern recognition (pp. 8695-8704).
>
> [2] Ojha, U., Li, Y., & Lee, Y. J. (2023). Towards universal fake image detectors that generalize across generative models. In Proceedings of the IEEE/CVF Conference on Computer Vision and Pattern Recognition (pp. 24480-24489).
>
> [3] Yan, S., Li, O., Cai, J., Hao, Y., Jiang, X., Hu, Y., & Xie, W. (2024). A sanity check for ai-generated image detection. arXiv preprint arXiv:2406.19435.
>
> [4] Khan, S. A., & Dang-Nguyen, D.-T. (2024). CLIPping the deception: Adapting vision-language models for universal deepfake detection. In Proceedings of the 2024 International Conference on Multimedia Retrieval (pp. 1006–1015).

---

> ### Author Response · Authors · 2024-12-02
>
> Dear Reviewer xU39,
>
> We would like to once again express our sincere gratitude for your valuable comments and feedback. As the Rebuttal period is approaching its end, we kindly remind you that if you have any further questions or concerns, please do not hesitate to reach out to us at your convenience.
>
> Thank you once again for your time and consideration.

---

### Meta-Review · Area_Chair_eTLS · 2024-12-21

**Metareview:**

4x accept. This paper introduces a multimodal benchmark aimed at evaluating LMMs’ abilities to detect synthetic data across video, image, 3D, text, and audio modalities. The reviewers agree on the (1) comprehensive coverage of multiple data domains and tasks, (2) clear and well-structured presentation, (3) innovative metrics like NBI for bias assessment, and (4) detailed evaluations including explainability. However, they note (1) insufficient robustness tests under real-world conditions and compression artifacts, (2) limited analysis of deeper bias causes, (3) lack of detail on data generation methods, and (4) partial prompting strategy evaluations. The authors’ rebuttal includes new compression tests, clarifications on synthetic data generation, further bias analyses, and expanded prompting experiments, so the AC leans to accept this submission.

**Additional Comments On Reviewer Discussion:**

N/A

---

### Decision · Program_Chairs · 2025-01-22

Accept (Spotlight)